# *Peptostreptococcus anaerobius* mediates anti-PD1 therapy resistance and exacerbates colorectal cancer via myeloid-derived suppressor cells in mice

Yali Liu[1], Chi Chun Wong[1], Yanqiang Ding[1], Mengxue Gao[2], Jun Wen [1], Harry Cheuk-Hay Lau [1], Alvin Ho-Kwan Cheung[3], Dan Huang [4], He Huang [2] & Jun Yu [1] ✉

Bacteria such as the oral microbiome member *Peptostreptococcus anaerobius* can exacerbate colorectal cancer (CRC) development. Little is known regarding whether these immunomodulatory bacteria also affect antitumour immune checkpoint blockade therapy. Here we show that administration of *P. anaerobius* abolished the efficacy of anti-PD1 therapy in mouse models of CRC. *P. anaerobius* both induced intratumoral myeloid-derived suppressor cells (MDSCs) and stimulated their immunosuppressive activities to impair effective T cell responses. Mechanistically, *P. anaerobius* administration activated integrin $\alpha_2\beta_1$–NF-κB signalling in CRC cells to induce secretion of CXCL1 and recruit CXCR2$^+$ MDSCs into tumours. The bacterium also directly activated immunosuppressive activity of intratumoral MDSCs by secreting lytC_22, a protein that bound to the Slamf4 receptor on MDSCs and promoted ARG1 and iNOS expression. Finally, therapeutic targeting of either integrin $\alpha_2\beta_1$ or the Slamf4 receptor were revealed as promising strategies to overcome *P. anaerobius*-mediated resistance to anti-PD1 therapy in CRC.

Accumulating evidence has suggested the existence of an intratumoral microbiome in multiple cancers[1]. For instance, colorectal cancer (CRC)-associated microbes play important roles in CRC development by modulating tumour cell proliferation and the tumour immune microenvironment[1,2]. Spatial profiling has demonstrated that pathogen-infected tumour cells from patients with CRC display gene signatures of enhanced cell migration and interferon response[3]. Studies of individual intratumoral pathogens also confirmed their ability to foster an immunosuppressive niche and drive colorectal tumorigenesis[4,5]. For instance, *Fusobacterium nucleatum* infection in CRC is associated with compromised antitumour immunity due to the recruitment of tumour-infiltrating myeloid cells[6]. Our investigations into pathogenic oral bacteria demonstrated that *Peptostreptococcus anaerobius*[7] and *Parvimonas micra*[8] enriched in CRC modulate the tumour immune microenvironment to facilitate CRC development and progression. Nevertheless, the molecular basis by which the intratumoral bacteria–immune cell interplay modulates colorectal tumorigenesis remains largely unknown.

[1]Institute of Digestive Disease and Department of Medicine and Therapeutics, State Key Laboratory of Digestive Disease, Li Ka Shing Institute of Health Sciences, CUHK Shenzhen Research Institute, The Chinese University of Hong Kong, Hong Kong, China. [2]Department of Biochemical Engineering, School of Chemical Engineering and Technology, Tianjin University, Tianjin, China. [3]Department of Anatomical and Cellular Pathology, State Key Laboratory of Translational Oncology, Prince of Wales Hospital, The Chinese University of Hong Kong, Hong Kong, China. [4]Department of Biology, School of Life Sciences, Southern University of Science and Technology, Shenzhen, China. ✉e-mail: junyu@cuhk.edu.hk

Immune checkpoint blockade (ICB) therapy is a breakthrough in the treatment of many cancers[9,10]; however, its efficacy is typically poor in most patients with CRC. Accumulating evidence has highlighted the important role of gut microbiota in modulating the response to ICB therapy. Gut commensals have been shown to be crucial in mediating effective immunotherapeutic responses[11–15], and fecal microbiota transplantation and probiotic cocktails[16] have been proposed as potential adjuvants for ICB therapy. On the other hand, it is unclear whether pathogenic intratumoral bacteria could function to suppress the efficacy of ICB therapy by virtue of their immunomodulatory effects. It is a particularly relevant issue in CRC given that pathobionts could directly infiltrate tumours to directly interact with and potentially impair or activate tumour-infiltrating immune cells.

Here we highlight the role of *P. anaerobius* in driving resistance to anti-PD1 therapy via the induction of immunosuppressive myeloid-derived suppressor cells (MDSCs) in tumours of mice with CRC. Mechanistically, *P. anaerobius* has a dual mode of action on MDSCs. *P. anaerobius* engages integrin $\alpha_2\beta_1$ on CRC cells to activate NF-κB and CXCL1, thereby promoting intratumoral infiltration of MDSCs. In addition, *P. anaerobius* secretes the lytC_22 protein into the tumour microenvironment, which directly interacts with the Slamf4 receptor on MDSCs to activate their immunosuppressive function. Importantly, the blockade of integrin $\alpha_2\beta_1$ or the Slamf4 receptor overcomes *P. anaerobius*-driven resistance to anti-PD1, representing potential druggable targets in CRC with *P. anaerobius* enrichment.

## Results

### *P. anaerobius* promotes an immunosuppressive tumour immune microenvironment in CRC

Transgenic $Apc^{min/+}$ mice were used to examine the role of *P. anaerobius* in colorectal tumorigenesis. Antibiotic-treated $Apc^{min/+}$ mice were orally gavaged with $1 \times 10^8$ colony-forming units (c.f.u.) of *P. anaerobius*, *Escherichia coli* (negative control) or PBS (Fig. 1a). Both tumour incidence and tumour load were increased in the *P. anaerobius*-treated mice compared with controls (Fig. 1a,b). The immune cell profile was evaluated by flow cytometry. We detected a higher number of MDSCs ($CD45^+CD11b^+Gr-1^{high}$) in the colonic lamina propria of the *P. anaerobius*-treated mice compared with the *E. coli* and the PBS control groups (both $P < 0.05$; Fig. 1c). Functional $CD8^+$ T cells (interferon-γ $(IFN-γ)^+CD8^+$) and T helper 2 cells ($T_H2$ cells; $IFN-γ^+CD4^+$) were significantly decreased in the lamina propria of *P. anaerobius*-treated mice compared with the controls (both $P < 0.05$; Fig. 1d), whereas no differences were observed in total $CD8^+$ and $CD4^+$ T cells (Extended Data Fig. 1a). Corroborating these data, immunofluorescence staining ($CD11b^+Gr-1^+$) validated that MDSCs were markedly induced in both tumour ($P < 0.0001$; Fig. 1e) and colon ($P = 0.0004$; Extended Data Fig. 1b) sites of *P. anaerobius*-treated mice.

We next validated the immunosuppressive role of *P. anaerobius* in an azoxymethane (AOM)-induced CRC mouse model (Extended Data Fig. 1c). Mice gavaged with *P. anaerobius* developed significantly more tumours and heavier tumour load compared with the PBS group ($P < 0.05$; Extended Data Fig. 1d). Consistent with the observations in $Apc^{min/+}$ mice, *P. anaerobius* administration in AOM-induced CRC resulted in an increase in MDSCs ($P < 0.05$; Extended Data Fig. 1e) along with a decrease in functional T cells ($IFN-γ^+CD8^+$ and $IFN-γ^+CD4^+$; both $P < 0.05$; Extended Data Fig. 1f) compared with the control mice, as determined by flow cytometry analysis. No change was found in the infiltration of total $CD8^+$ and $CD4^+$ T cells (Extended Data Fig. 1g). Immunofluorescence staining of tumours (Extended Data Fig. 1h) and colon tissues (Extended Data Fig. 1i) confirmed significantly higher numbers of MDSCs in the *P. anaerobius*-treated mice (both $P < 0.0001$). These findings indicate that *P. anaerobius* promotes an immunosuppressive microenvironment in CRC by enhancing MDSCs and reducing functional T cell infiltration.

We validated these observations in a mouse model in which AOM and dextran sodium sulfate (DSS) are used to induce CRC. *P. anaerobius* promoted tumour growth (Extended Data Fig. 1j), increased the number of intratumoral MDSCs and decreased $IFN-γ^+CD8^+$ T cells compared with the PBS and *E. coli* groups (Extended Data Fig. 1k,l). However, no significant changes in the abundance of lymphoid cell types 1, 2 and 3, plasmacytoid dendritic cells, myeloid dendritic cells or γδ T cells in colonic lamina propria were observed (Extended Data Fig. 1m). To further confirm the modulation of the tumour immune microenvironment by *P. anaerobius*, we profiled tumour-infiltrating leukocytes in the AOM and DSS mouse model using single-cell RNA sequencing (scRNA-seq). Tumours were isolated from mice treated with *P. anaerobius* or PBS, and $CD45^+$ cells from dissociated tumours were sorted, pooled and sequenced (Extended Data Fig. 2a). We identified and annotated eight distinct clusters based on the expression of known genetic markers (Fig. 1f and Extended Data Fig. 2b). Tumours of mice gavaged with *P. anaerobius* showed a significant increase in the proportion of intratumoral MDSCs (Fig. 1f), thereby confirming the flow cytometry findings. We also observed a decrease in neutrophils expressing the canonical neutrophil markers MMP8 and Retnlg as well as macrophages after *P. anaerobius* treatment (Fig. 1f). To visualize shifts in different T lymphocyte populations, we re-clustered T lymphocytes into subsets and identified seven subclusters (Fig. 1g and Extended Data Fig. 2c) that all expressed Trbc1. We observed a significant decrease in cytotoxic T cell populations characterized by Prf1, Nkg7 and Ifng expression, identified by flow cytometry, after *P. anaerobius* administration (Fig. 1g), which further confirmed the impaired cytotoxic T cell function following *P. anaerobius* infection. In line with our results, the intratumoral *P. anaerobius* abundance in The Cancer Genome Atlas (TCGA) CRC cohort was found to correlate positively with MDSCs ($P = 0.01$) but correlate negatively with effector T cells ($P = 0.02$; Extended Data Fig. 2d).

To assess whether *P. anaerobius*-induced MDSC recruitment is tumour specific, we gavaged naive mice with *P. anaerobius*. Flow cytometry showed that *P. anaerobius* also increased MDSCs in naive mice

**Fig. 1 | *P. anaerobius* promotes an immunosuppressive tumour microenvironment and induced anti-PD1 immunotherapy resistance in CRC. a**, Schematic of the experimental design for the *P. anaerobius* treatment of transgenic $Apc^{min/+}$ mice (left). Representative images of mouse colonoscopies at week 9 (right). **b**, Representative images of the colon following euthanasia (left). Tumour load in PBS-, *E. coli*- and *P. anaerobius*-treated $Apc^{min/+}$ mice (right); $n = 13$ (PBS) and 11 (*E. coli* and *P. anaerobius*) mice. **c**, Absolute number of MDSCs in the colonic lamina propria of the $Apc^{min/+}$ mice; $n = 10$ (PBS, *E. coli*) and 9 (*P. anaerobius*) mice. **d**, Absolute number of $IFN-γ^+CD8^+$ and $IFN-γ^+CD4^+$ T cells in the colonic lamina propria of $Apc^{min/+}$ mice; $n = 7$ (PBS and *P. anaerobius*) and 6 (*E. coli*) mice. **e**, Immunofluorescence quantification of $CD11b^+Gr-1^+$ cells in tumours from $Apc^{min/+}$ mice; $n = 6$ mice per group. Scale bars, 25 μm. **f**, T-distributed stochastic neighbour embedding (tSNE) plot of immune cell clusters (left) and the proportion of immune cell types in control- and *P. anaerobius*-treated groups (right). **g**, Re-clustering of T lymphocytes. A tSNE plot for T cell subclusters (middle) and the proportion of T lymphocyte subtypes in the different groups (right) are provided. **h**, Tumour growth curve (left) and quantification of tumour weight (right) of MC38 allograft model mice following anti-PD1 (a-PD1) therapy. **i**, The percentage of MDSCs (left) and $IFN-γ^+CD8^+$ T cells (right) in MC38 allografts were determined by flow cytometry analysis. **j**, MDSC depletion using anti-Ly6G in the MC38 allograft model. Tumour growth curve (left) and tumour weight (right) for each group. **h–j**, $n = 5$ mice were used in each group. **b–e,h–j**, Data are the mean ± s.e.m. **b–j**, $P$ values were calculated using a one-way analysis of variance (ANOVA), followed by Tukey's post-hoc test (**b–e,h–j**), Fisher's exact test (**f,g**) or two-way ANOVA, followed by Bonferroni's post-hoc test (**h,j**); *$P < 0.05$, **$P < 0.01$ and ***$P < 0.001$. EC, *E. coli*; PA, *P. anaerobius*; DCs, dendritic cells.

(Extended Data Fig. 2e), suggesting that *P. anaerobius* recruits MDSCs independently from tumorigenesis. Nevertheless, the presence of CRC increased the abundance of *P. anaerobius* in stool and its colonization of the colon (Extended Data Fig. 2f,g).

## *P. anaerobius* reduces anti-PD1 efficacy by increasing MDSCs

The expansion of immunosuppressive MDSCs is associated with acquired resistance to PD1 blockade[17]. Given the enrichment of MDSCs by *P. anaerobius*, we explored whether intratumoral *P. anaerobius* could

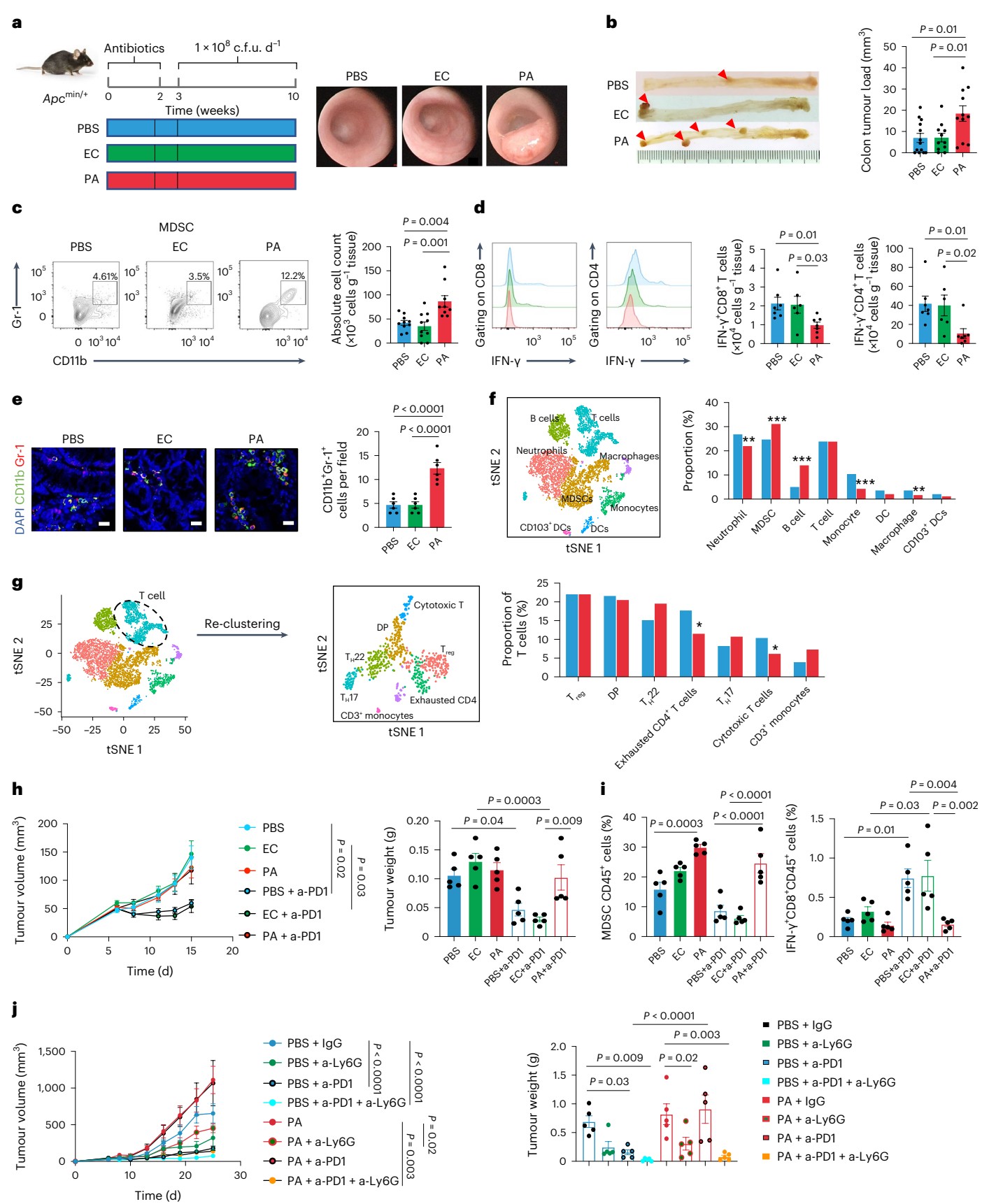

affect the efficacy of anti-PD1 therapy. We established a syngeneic mouse model with the MC38 CRC cell line (MSI-H) that is responsive to anti-PD1 treatment (Extended Data Fig. 3a). We found that both the tumour volume and weight (Fig. 1h and Extended Data Fig. 3a) in the PBS and *E. coli* groups were significantly reduced after anti-PD1 treatment. In contrast, the antitumor efficacy of anti-PD1 was blunted in the mice that were intratumorally injected with *P. anaerobius* (Fig. 1h). Higher levels of intratumoral MDSCs (Fig. 1i), specifically granulocyte-like MDSCs (Extended Data Fig. 3b), were observed (by flow cytometry) in tumours from *P. anaerobius*-injected mice regardless of anti-PD1 treatment. Although anti-PD1 activated IFN-$\gamma^+$CD8$^+$ T cells in the PBS and *E. coli* groups (Fig. 1i), this effect was abrogated by *P. anaerobius*. Anti-PD1 also suppressed PD1$^+$CD8$^+$ T cells, whereas *P. anaerobius* had no effect (Extended Data Fig. 3b). Hence, treatment with intratumorally injected *P. anaerobius* attenuates anti-PD1 efficacy in CRC allografts.

We sought to validate the effect of *P. anaerobius* on anti-PD1 efficacy in CRC tumorigenesis in mice induced by AOM and DSS. After confirming colon tumour formation by colonoscopy, monoclonal anti-PD1 or IgG isotype control were administered to mice by intraperitoneal injection together with daily gavage of *P. anaerobius* or PBS (Extended Data Fig. 3c). The anti-PD1 treatment significantly inhibited tumour growth compared with the IgG control ($P < 0.05$; Extended Data Fig. 3c). However, mice that were co-treated with anti-PD1 and *P. anaerobius* showed significantly increased tumour burdens compared with the mice treated with anti-PD1 alone, suggesting that *P. anaerobius* administration attenuated the efficacy of anti-PD1 ($P < 0.01$; Extended Data Fig. 3c). Consistent with these data, flow cytometry (Extended Data Fig. 3d) and immunofluorescence staining (Extended Data Fig. 3e) of tumours indicated significantly higher levels of MDSC infiltration in the *P. anaerobius*-treated mice, irrespective of whether they had been treated with IgG or anti-PD1. Accordingly, *P. anaerobius* administration abolished the induction of IFN-$\gamma^+$CD8$^+$ T cells by anti-PD1 (Extended Data Fig. 3f). Moreover, *P. anaerobius* reversed the effect of anti-PD1 on decreasing PD1$^+$CD8$^+$ T cells (Extended Data Fig. 3g). We then depleted MDSCs in MC38 tumour-bearing mice by injecting them with anti-Ly6G (Fig. 1j and Extended Data Fig. 3h). Anti-Ly6G dramatically increased the antitumour effect of anti-PD1 therapy in the context of *P. anaerobius* (Fig. 1j and Extended Data Fig. 3i,j), demonstrating the importance of MDSCs in anti-PD1 resistance induced by *P. anaerobius*. Together, *P. anaerobius* compromises the efficacy of anti-PD1 therapy in CRC tumorigenesis by recruiting MDSCs to antagonize CD8$^+$ T cell function.

### *P. anaerobius* induces CXCL1 secretion to recruit MDSCs

Chemotaxis of MDSCs is primarily driven by tumour cell-derived chemokines[18]. To investigate whether *P. anaerobius* modulates CRC cell-mediated MDSC recruitment, we first collected conditioned medium from CRC cells that were pre-incubated with or without *P. anaerobius* and then evaluated their ability to induce MDSC migration in a transwell assay (Extended Data Fig. 4a). Conditioned medium derived from CRC cells (HCT116, Caco-2 and MC38) pre-incubated with *P. anaerobius* exhibited an enhanced ability to induce MDSC chemotaxis compared with conditioned medium from CRC cells that were pre-incubated with PBS or blank control (DMEM medium + *P. anaerobius* with antibiotics; Fig. 2a and Extended Data Fig. 4b). To evaluate *P. anaerobius*-induced secretary factors in CRC cells, we performed a chemokine antibody array of conditioned medium from CRC cells with or without *P. anaerobius* treatment (Fig. 2b). *P. anaerobius* significantly altered the levels of secreted cytokines in Caco-2 cells, among which CXCL1 was the most significantly upregulated. We validated the induction of CXCL1 secretion by *P. anaerobius* in a panel of CRC cell lines using enzyme-linked immunoassays (ELISAs; Fig. 2c and Extended Data Fig. 4c). Elevated CXCL1 levels were detected in the peripheral blood (protein) and colonic tissues (messenger RNA) of *P. anaerobius*-treated *Apc*$^{min/+}$ mice as well as *P. anaerobius*-treated AOM CRC mice (Fig. 2c and Extended Data Fig. 4c).

To test whether CRC cell-derived CXCL1 is involved in *P. anaerobius*-induced MDSC migration, we silenced CXCL1, using small interfering RNA (siRNA) targeting *CXCL1*, in CRC cell lines (HCT116, Caco-2 and MC38) co-cultured with *P. anaerobius* (Extended Data Fig. 4d). Conditioned medium from *CXCL1*-knockdown cell lines pre-incubated with *P. anaerobius* failed to enhance MDSC migration (Extended Data Fig. 4e). Similarly, the addition of an antagonist of CXCR2 (SB225002), the cognate receptor of CXCL1 (ref. 19), abrogated the induction of MDSC migration in conditioned medium from all three CRC cell lines pre-incubated with *P. anaerobius* ($P < 0.01$; Extended Data Fig. 4e). CXCL3, another CXCR2 ligand[19], was also elevated by *P. anaerobius* co-culture, albeit to a lower extent compared with CXCL1 (Extended Data Fig. 4f). Moreover, *CXCL3* knockdown using siRNA had no impact on *P. anaerobius*-mediated MDSC migration (Extended Data Fig. 4g). These data suggest that *P. anaerobius* conveys immunomodulation by inducing the tumour cell-derived chemoattractant CXCL1 and the latter interacts with CXCR2 on MDSCs to promote MDSC migration.

### *P. anaerobius* promotes the integrin $\alpha_2\beta_1$–NF-κB–CXCL1 axis in CRC

Next, we investigated the mechanism through which *P. anaerobius* promotes CXCL1 secretion. We previously showed that *P. anaerobius* could directly interact with the integrin $\alpha_2\beta_1$ receptor on CRC cells[7], which drives its oncogenic effect. To investigate whether *P. anaerobius*-mediated recruitment of MDSCs involves its interaction with the integrin $\alpha_2\beta_1$ receptor, we performed integrin $\alpha_2\beta_1$ knockdown using siRNA in the CRC cells HCT116 and Caco-2. Knockdown of integrin $\alpha_2\beta_1$ in CRC cells (Extended Data Fig. 5a) abrogated

**Fig. 2 | *P. anaerobius* drives MDSC migration by activating the integrin $\alpha_2\beta_1$–NF-κB–CXCL1 pathway in CRC cells. a**, Representative images of MDSCs that migrated to the lower chamber and quantification of MDSC migration towards CRC-cell-conditioned medium pre-incubated with vehicle or *P. anaerobius* ($n = 3$ biologically independent samples). Scale bars, 100 μm. **b**, Cytokine array analysis (36 cytokines) of conditioned medium from Caco-2 cells treated with *P. anaerobius* or vehicle (left). Fold change in cytokine levels after *P. anaerobius* treatment (right). Red box, significantly changed cytokines; NC, negative control. **c**, Levels of CXCL1 secreted from Caco-2 and MC38 cells treated with vehicle or *P. anaerobius* (left). Levels of serum CXCL1 and mRNA expression of *CXCL1* in the colon of *Apc*$^{min/+}$ mice (right). **d**, Integrin $\alpha_2\beta_1$ knockdown in Caco-2 cells abrogated *P. anaerobius*-induced CXCL1 secretion (left) and migration of MDSCs (right). **e**, RGDS peptide treatment of MC38 cells abrogated *P. anaerobius*-induced CXCL1 secretion (left) and migration of MDSCs (right). **f**, Treatment with JSH-23 abolished *P. anaerobius*-induced CXCL1 secretion (left) and MDSCs migration (right). **c–f**, $n = 3$ biologically independent samples. **g–j**, AOM-induced CRC mice were treated with RGDS.

**g**, Tumour load of the mice; $n = 8$ (PBS), 11 (PA), 7 (PBS + RGDS) and 10 (PA + RGDS) mice. **h**, Serum levels of CXCL1; $n = 5$ (PBS and PA) and 4 (PBS + RGDS and *PA* + RGDS). **i**, Percentage of MDSCs in the colonic lamina propria of the mice (left). Immunofluorescence images (middle) and quantification of CD11b$^+$Gr-1$^+$ cells in mouse tumours (right). Scale bars, 25 μm. **j**, Mean fluorescence intensity (MFI) of IFN-$\gamma^+$CD8$^+$ and IFN-$\gamma^+$CD4$^+$ T cells in colonic lamina propria. **i,j**, $n = 7$ (PBS and PBS + RGDS) and 9 (PA and PA + RGDS) mice. **k–m**, MC38 allograft model mice were treated with RGDS peptide and anti-PD1. **k**, Tumour growth curves (left) and tumour weight (right); $n = 7$ (PBS + anti-PD1) and 8 (all other groups) mice. **l**, CXCL1 levels in tumour tissue lysates; $n = 5$ (PBS + IgG, PBS + anti-PD1 and PBS + anti-PD1 + RGDS), 8 (PA + IgG), 7 (PA + anti-PD1) and 3 (PA + anti-PD1 + RGDS) mice. **m**, Percentage of MDSCs (left) and IFN-$\gamma^+$CD8$^+$ T cells (right) in tumours; $n = 7$ (PBS + anti-PD1) and 8 (all other groups) mice. **a,c–m**, Data are the mean ± s.e.m. *P* values were calculated using a one-way ANOVA, followed by Tukey's post-hoc test (**a**,**d–j**); an unpaired two-tailed Student's *t*-test (**c**); a one-way ANOVA, followed by Fisher's least significant difference test (**k–m**) or a two-way ANOVA, followed by Bonferroni's post-hoc test (**k**).

*P. anaerobius*-induced CXCL1 secretion and MDSC migration (Fig. 2d and Extended Data Fig. 5a). Similarly, the RGDS peptide, a blocker of integrin $\alpha_2\beta_1$ receptor, inhibited the induction of CXCL1 and MDSC migration in conditioned medium of CRC cells treated with

*P. anaerobius* (Fig. 2e and Extended Data Fig. 5b). *P. anaerobius*-driven MDSC recruitment thus depends on its direct interplay with CRC cells. We further demonstrated that *P. anaerobius* promoted the activation of NF-κB p65 in CRC cells (Extended Data Fig. 5c), which was abolished

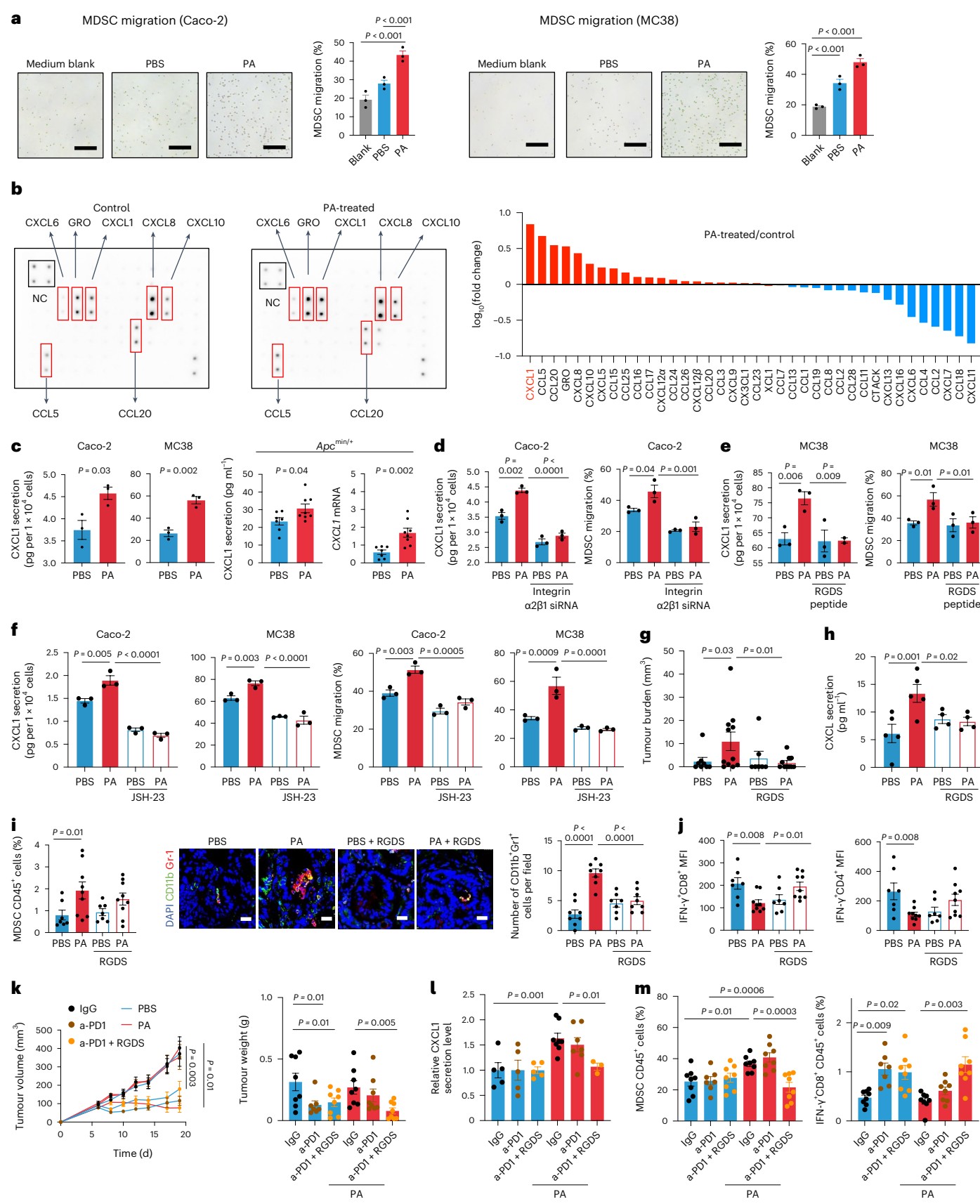

by integrin $\alpha_2\beta_1$ blockade (Extended Data Fig. 5b). The NF-κB inhibitor JSH-23 reversed *P. anaerobius*-induced phospho-p65 expression (Extended Data Fig. 5d), CXCL1 secretion and MDSC migration (Fig. 2f and Extended Data Fig. 5e), thus validating the role of the integrin $\alpha_2\beta_1$–NF-κB cascade in driving CXCL1 downstream of *P. anaerobius*. To corroborate our in vitro results, we administered RGDS peptide injections to AOM-induced CRC mice gavaged with *P. anaerobius* or PBS (Extended Data Fig. 5f). The RGDS peptide blocked the effect of *P. anaerobius* on CRC formation promotion (Fig. 2g) and serum CXCL1 induction (Fig. 2h). Flow cytometry and immunofluorescence staining revealed that *P. anaerobius* failed to promote the accumulation of MDSCs when mice were co-treated with RGDS peptide (Fig. 2i). Consequently, the reduction of IFN-γ⁺CD8⁺ T and $T_H2$ cells by *P. anaerobius* was reversed by RGDS (Fig. 2j and Extended Data Fig. 5g). Together, these findings confirmed that *P. anaerobius* induces MDSC infiltration and an immunosuppressive tumour microenvironment in CRC via the integrin $\alpha_2\beta_1$–NF-κB–CXCL1 axis.

### RGDS reverses *P. anaerobius*-driven anti-PD1 resistance

We further explored whether blockade of the integrin α2β1 receptor by RGDS peptide could reverse *P. anaerobius*-induced anti-PD1 resistance. As shown in Fig. 2k and Extended Data Fig. 5h, without intratumoral *P. anaerobius* injection, the RGDS peptide had no antitumor effect on anti-PD1 therapy in MC38 allografts. However, co-treatment with RGDS peptide rescued the efficacy of the monoclonal anti-PD1 in *P. anaerobius*-injected mice, as evidenced by reduced tumour growth and tumour volume compared with the *P. anaerobius* group ($P = 0.005$). Co-treatment with RGDS peptide also blocked the effect of intratumoral *P. anaerobius* on CXCL1 secretion (Fig. 2l), MDSC infiltration (Fig. 2m) and IFN-γ⁺CD8⁺ T cell suppression (Fig. 2m and Extended Data Fig. 5j), thereby reversing the anti-PD1 resistance phenotypes. Consistent with these data, the RGDS peptide inhibited integrin $\alpha_2\beta_1$–NF-κB signalling to the baseline in *P. anaerobius* injected intratumorally to mice (Extended Data Fig. 5k). Targeting the *P. anaerobius*–integrin α2β1 interplay thus reverses *P. anaerobius*-driven anti-PD1 resistance in CRC.

### *P. anaerobius* culture supernatant activates MDSC function

Myeloid-derived suppressor cells suppress T cell activation by expressing arginase (Arg1) and inducible nitric oxide synthase (iNOS)[20]. However, conditioned medium from CRC cells co-cultured with *P. anaerobius* failed to elicit changes in *Arg1* and *iNOS* mRNA expression in MDSCs (Extended Data Fig. 6a). We thus investigated whether *P. anaerobius* might directly activate MDSCs. To this end, we co-incubated primary MDSCs in *P. anaerobius*-conditioned medium (PA-CM) and evaluated Arg1 and iNOS expression (Extended Data Fig. 6b). Incubation with PA-CM significantly increased the levels of mRNA ($P = 0.01$) and protein ($P < 0.001$) expression of both Arg1 and iNOS by MDSCs compared with the broth control and *E. coli*-conditioned medium (EC-CM; Fig. 3a). We then performed a T cell suppression assay to validate that MDSC function was boosted by PA-CM. Primary MDSCs were first treated with PA-CM, EC-CM or broth control for 24 h and then co-cultured with primary CD8⁺ or CD4⁺ T cells (Fig. 3b). Compared with MDSCs treated with either broth or EC-CM, MDSCs treated with PA-CM suppressed CD4⁺ and CD8⁺ T cell proliferation more effectively, as demonstrated by flow cytometry assays (Fig. 3c). Consistent with this, the numbers of activated CD8⁺ and CD4⁺ T cells (IFN-γ⁺ or granzyme B (GzmB)⁺) were significantly reduced after co-culture with MDSCs activated with PA-CM (Fig. 3d). Direct co-culture of PA-CM with CD8⁺ or CD4⁺ T cells had no effect on their function (Extended Data Fig. 6c), suggesting that the *P. anaerobius* secretome stimulates the MDSC function to suppress T cell activation in CRC.

To characterize PA-CM responsible for the activation of MDSCs, we centrifuged PA-CM into six different fractions based on molecular cut-offs (<3, 3–10, 10–30, 30–50, 50–100 and >100 kDa; Extended Data Fig. 6d). Among these, the 50–100 kDa fraction had the strongest

induction effect on mRNA and protein expression (Fig. 3e,f) of Arg1 and iNOS in MDSCs. Consistent with these data, MDSCs treated with the 50–100 kDa fraction significantly suppressed anti-CD3 and anti-CD28-stimulated CD8⁺ and CD4⁺ T cell proliferation as well as the proportion of IFN-γ⁺ and GzmB⁺ CD8⁺ and CD4⁺ T cells (Fig. 3g,h). Collectively, these results demonstrate that *P.*anaerobius-secreted products with a molecular weight of 50–100 kDa are responsible for MDSC activation in CRC.

### *P. anaerobius* protein lytC_22 induces MDSC activation

To identify the nature of secreted products involved in stimulation of MDSCs, we first inactivated proteins in PA-CM by heating or treatment with proteinase K (PK). Both heat and PK completely abolished the effect of PA-CM on Arg1 and iNOS in MDSCs (Fig. 4a,b). In agreement with this, MDSCs treated with heat- or PK-inactivated PA-CM failed to elicit suppression of CD8⁺ or CD4⁺ T cell function (Fig. 4c), suggesting that *P. anaerobius*-secreted proteins enhance the immunosuppressive function of MDSCs.

We therefore performed mass spectrometry to characterize the 50–100 kDa proteome of *P. anaerobius* supernatant. Among the identified proteins, *N*-acetylmuramoyl-L-alanine amidase (lytC_22) was specifically enriched in the 50–100 kDa fraction (Extended Data Fig. 7a). To evaluate the function of lytC_22, we expressed recombinant glutathione *S*-transferase (GST)-tagged lytC_22 in *E. coli* and purified lytC_22. Recombinant lytC_22 elevated the expression levels of Arg1 and iNOS, both mRNA and protein, in a dose-dependent manner in MDSCs (Fig. 4d). In addition, MDSCs treated with lytC_22 had a significantly stronger inhibitory effect on CD4⁺ and CD8⁺ T cell proliferation (Fig. 4e) and activation (Fig. 4f) than untreated MDSCs. LytC_22 also induced MDSC migration in a transwell assay (Extended Data Fig. 7b). Collectively, these results point to lytC_22 being the functional protein generated from *P. anaerobius* that promotes the immunosuppressive activity of MDSCs.

### *P. anaerobius* lytC_22 activates the MDSC Slamf4 receptor

We next sought to identify the corresponding receptor responsible for lytC_22-induced MDSC activation. Toll-like receptors (TLRs)[21] and the signalling lymphocytic activation molecule gene family (SLAMF)[22] are the main sensors of MDSCs in response to microbial challenges. We surveyed the mRNA expression levels of these receptors and found that only the Slamf4 receptor was significantly upregulated in MDSCs treated with *P. anaerobius* supernatant (Fig. 5a). In addition, scRNA-seq data from AOM and DSS model mice revealed that Cd244a was enhanced in the MDSC population after *P. anaerobius* treatment (Extended Data Fig. 7c). Treatment with lytC_22 also elevated the mRNA and protein expression levels of Slamf4 (Fig. 5b and Extended Data Fig. 7d). Slamf4 is a transmembrane receptor and its expression on MDSCs correlates with a high level of Arg1 and immunosuppressive activity of these cells[23]. In line with this, neutralizing antibodies to Slamf4 abrogated the induction of Arg1 and iNOS in MSDCs by recombinant lytC_22 (Fig. 5c) and also impaired the capacity of lytC_22-treated MDSCs to suppress CD8⁺ T cell function (Fig. 5d) and proliferation (Fig. 5e), which suggests that Slamf4 mediates MDSC activation in response to *P. anaerobius* lytC_22.

To determine whether there is direct interplay between lytC_22 and Slamf4, we first co-immunoprecipitated MDSC proteins that interact with recombinant lytC_22 (Fig. 5f). GST–lytC_22 specifically pulled down Slamf4 from primary MDSC lysates (Fig. 5f). Its direct interaction with lytC_22 was verified by GST pulldown using purified human SLAMF4 (hSLAMF4) protein (Fig. 5g). We confirmed the direct interplay between SLAMF4 and lytC_22, with an apparent dissociation constant of 40.12 nM, using a microscale thermophoresis (MST) assay (Fig. 5h). Based on in silico prediction of its binding sites with Slamf4 (Fig. 6a), we generated mutant lytC_22 (E570A, D653A, D725A, Y661A, D712A, R660A and K669A). Compared with wild-type lytC_22, the mutant lytC_22 exhibited impaired binding to Slamf4 (Fig. 6b) and

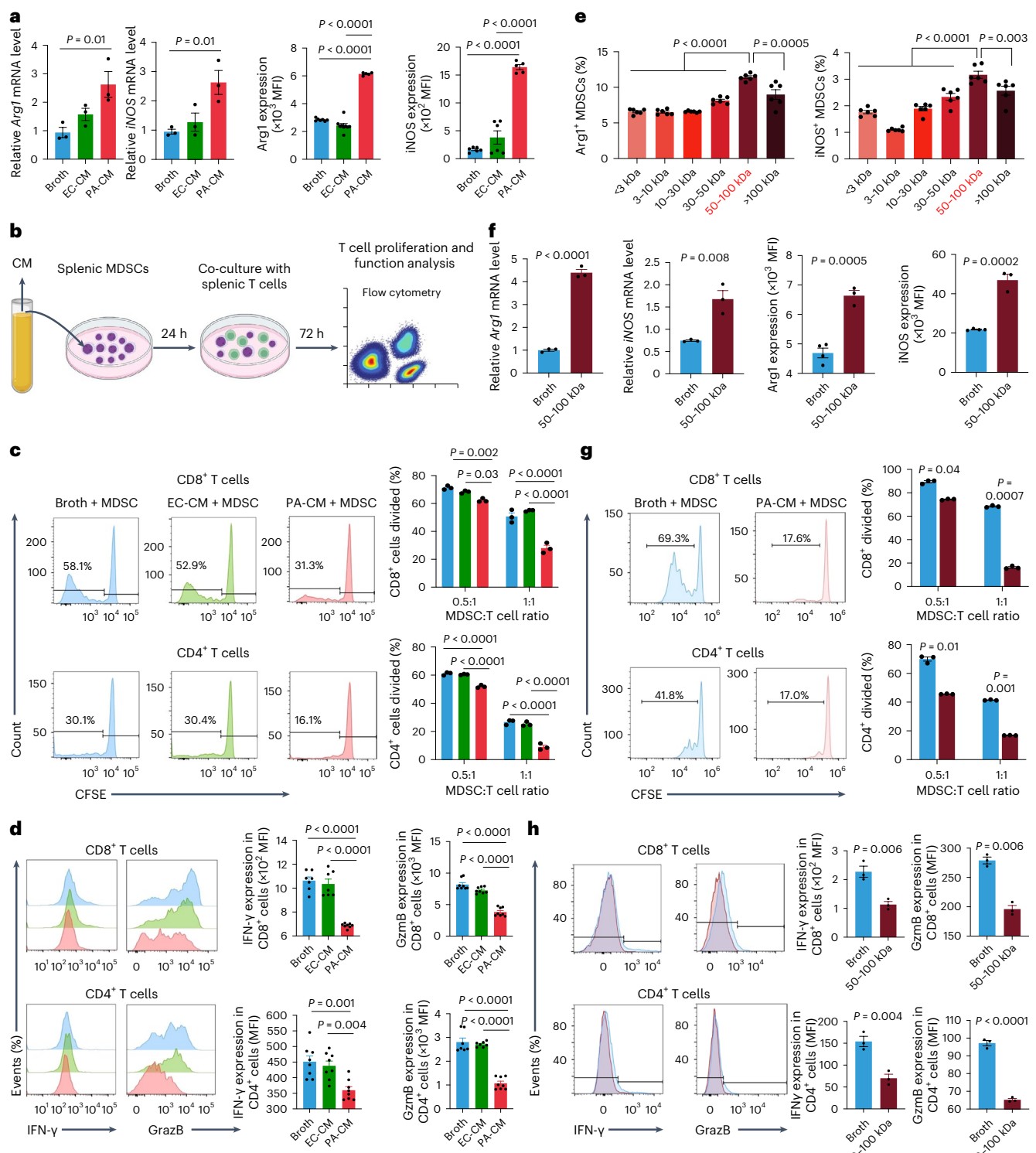

**Fig. 3 | *P. anaerobius*-derived products directly modulate MDSCs. a**, MDSCs treated with 10% PA-CM or EC-CM were harvested after treatment for 24 h. Levels of *Arg1* and *iNOS* mRNA expression and MFI in MDSCs. **b**, Schematic of the in vitro T cell suppression assay. **c**, Carboxyfluorescein succinimidyl ester (CFSE)-labelled T cells were co-cultured at different ratios with MDSCs that had been pretreated with control broth, EC-CM or PA-CM for 72 h, and the proliferation (right) of CD4⁺ and CD8⁺ T cells was determined by flow cytometry (left). **d**, T cells were co-cultured with MDSCs at a 1:1 ratio for 72 h and the MFI of IFN-γ and GzmB in CD4⁺ and CD8⁺ T cells was determined; *n* = 8 biologically independent samples. **e**, Different molecular fractions from PA-CM were used to treat MDSCs. The percentage of Arg1- (left) and iNOS-expressing (right) MDSCs was examined using flow cytometry; *n* = 6 biologically independent samples. The

red font indicates that 50–100 kDa fraction from PA-CM had the strongest effect on activating MDSCs. **f**, MDSCs were treated with the 50–100 kDa fraction from PA-CM or control broth. The levels of mRNA expression and MFI of Arg1 and iNOS of MDSCs were evaluated. **g,h**, T cells were co-cultured with MDSCs pretreated with the 50–100 kDa fraction from PA-CM for 72 h. CD4⁺ and CD8⁺ T cell proliferation (**g**) as well as the IFN-γ and GzmB MFI (**h**) were determined. **a,c,f–h**, *n* = 3 biologically independent samples. **a,c–h**, Data are the mean ± s.e.m. *P* values were calculated using a one-way ANOVA, followed by Tukey's post-hoc test (**a,c,e**); two-way ANOVA, followed by Bonferroni's post-hoc test (**c,g**) or unpaired two-tailed Student's *t*-test (**f,h**). Percentages in the flow plots are percentages of subsequent cell divisions in total CD8⁺ or CD4⁺ T cells (**c,g**).

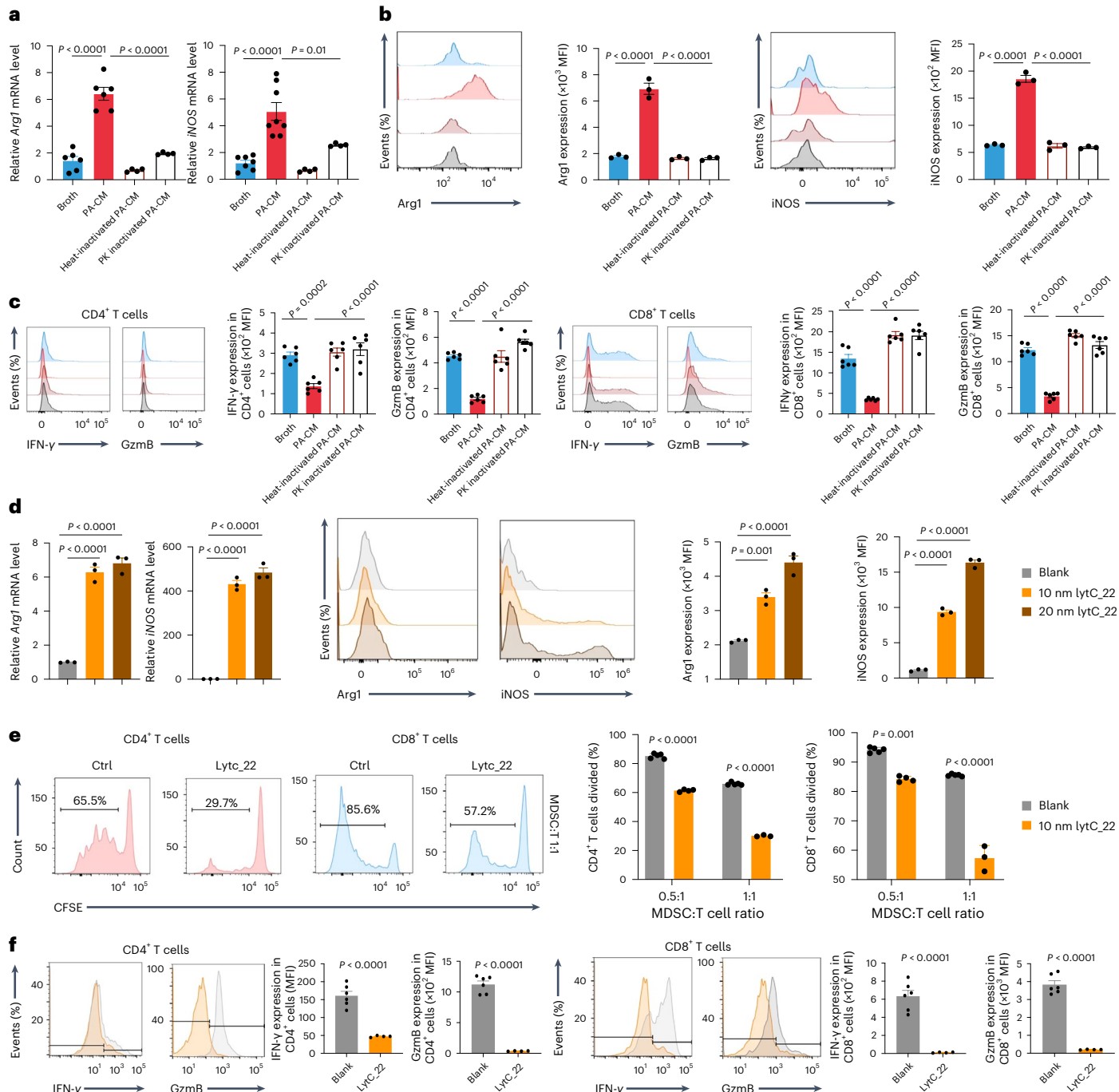

**Fig. 4 | The *P. anaerobius*-secreted protein lytC_22 induces MDSCs activation.**
**a**–**c**, MDSCs were treated with broth, PA-CM, heat-treated PA-CM or PK-treated PA-CM. **a**, Levels of *Arg1* and *iNOS* mRNA, detected by quantitative PCR, in MDSCs. *n* = 6 (broth, PA-CM); *n* = 4 (heat-inactivated PA-CM, PK-inactivated PA-CM). **b**, MFI of Arg1 and iNOS of MDSCs, determined by flow cytometry. **c**, The function of CD4⁺ and CD8⁺ T cells (IFN-γ and GzmB expression) was measured using flow cytometry after co-culture with MDSCs; *n* = 6 biologically independent samples. **d**, The Arg1 and iNOS levels of MDSCs treated with recombinant lytC_22 (10 and 20 nM) were determined using quantitative PCR (left) and flow cytometry (right). **b**,**d**, *n* = 3 biologically independent samples. **e**, CFSE-labelled T cells

were co-cultured at different ratios with MDSCs, pretreated with or without (control, Ctrl) lytC_22 (10 nM) for 72 h, and CD4⁺ and CD8⁺ T cell proliferation was determined by flow cytometry; *n* = 5 (solvent blank); *n* = 4 (lytC_22). Percentages in the flow plots are the percentages of subsequent cell divisions in total CD8⁺ and CD4⁺ T cells. **f**, T cells were co-cultured at a 1:1 ratio with MDSCs, pretreated with or without lytC_22 (10 nM) for 72 h and the MFI of IFN-γ and GzmB in the CD4⁺ and CD8⁺ T cell subpopulations was determined; *n* = 6 (solvent blank) and 4 (lytC_22). **a**–**f**, Data are the mean ± s.e.m. *P* values were calculated using a one-way ANOVA, followed by Tukey's post-hoc test (**a**–**d**); two-way ANOVA, followed by Bonferroni's post-hoc test (**e**) or an unpaired two-tailed Student's *t*-test (**f**).

accordingly was incapable of inducing Arg1 and iNOS expression (Fig. 6c) or immunosuppressive function (Fig. 6d and Extended Data Fig. 7e) of MDSCs in vitro. Collectively, these data indicate that *P. anaerobius*-derived lytC_22 directly interacts with Slamf4 receptor of MDSCs to drive MDSC activation.

To validate the function of lytC_22 in vivo, we established MC38 allografts and treated established tumours with monoclonal anti-PD1 in the presence or absence of recombinant lytC_22 or *P. anaerobius* by intratumoral injection. The efficacy of anti-PD1 therapy was reduced by co-treatment with either *P. anaerobius* or lytC_22, as evidenced

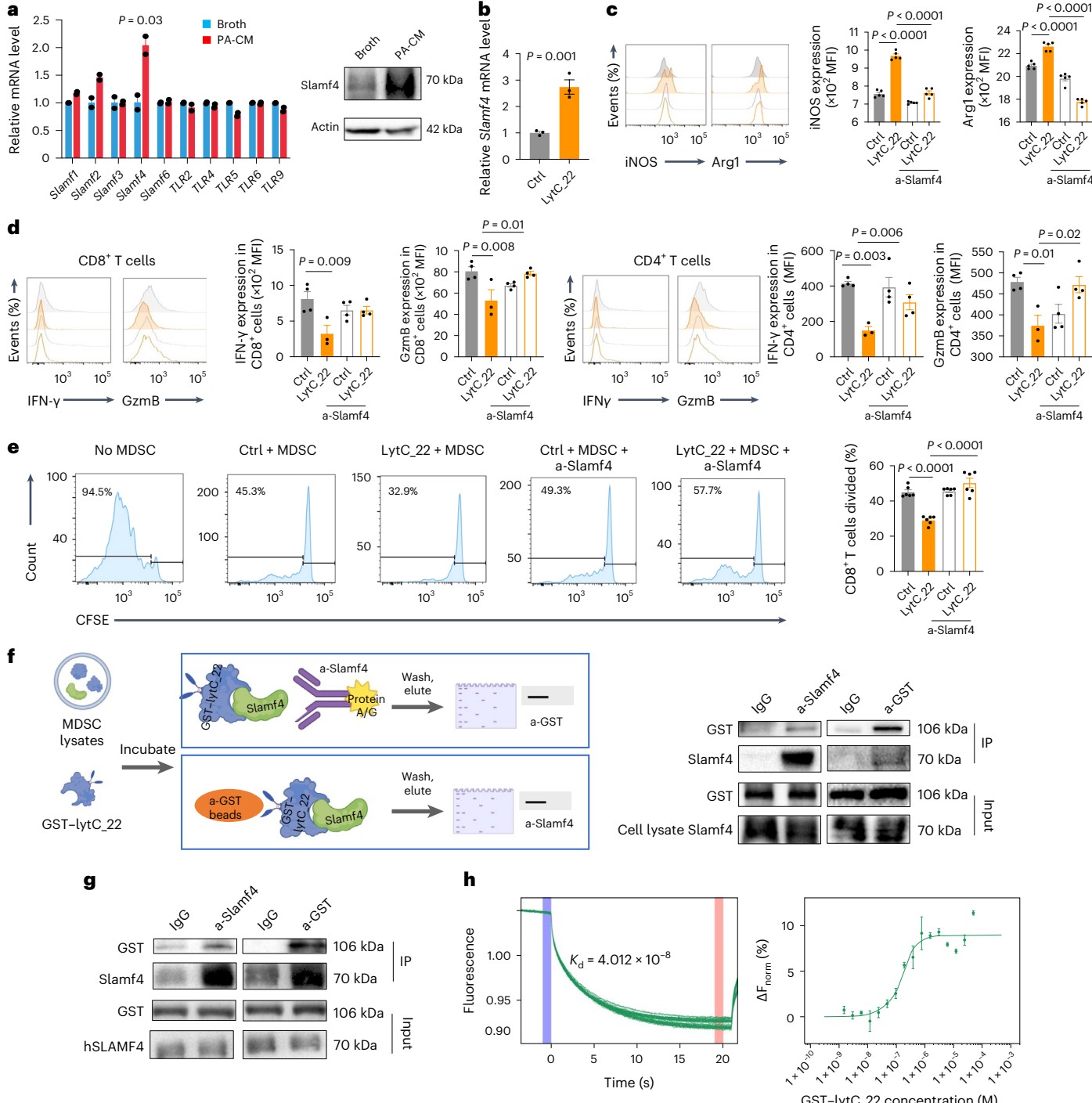

**Fig. 5 | *P. anaerobius* induces MDSC activation via the interaction between lytC_22 and Slamf4. a**, Levels of TLR and SLAMF mRNA expression in MDSCs treated with PA-CM or control broth (left); $n = 2$ biologically independent samples. Western blot showing the levels of Slamf4 protein in MDSCs treated with PA-CM or broth (right). **b**, *Slamf4* mRNA in MDSCs after recombinant lytC_22 treatment; $n = 3$ biologically independent samples. **c**–**e**, MDSCs were pre-incubated with 20 μg ml⁻¹ anti-Slamf4 (a-Slamf4), followed by lytC_22 treatment. **c**, MFI of Arg1 and iNOS in MDSCs; $n = 5$ biologically independent samples. **d**, IFN-γ and GzmB expression in CD4⁺ and CD8⁺ T cell populations after co-culture with MDSCs; $n = 3$ (lytC_22) and 4 (Ctrl, Ctrl + a-Slamf4 and lytC_22 + a-Slamf4) biologically independent samples. **e**, CD8⁺ T cell proliferation after co-culture

with MDSCs at a 1:1 ratio; $n = 6$ biologically independent samples. Percentages in the flow plots are the percentages of subsequent cell divisions in total CD8⁺ cells. **f**, GST-tag pulldown assay showing the interaction between GST–lytC_22 and endogenous Slamf4 in MDSCs. **g**, GST-tag pulldown of GST–lytC_22 with recombinant hSLAMF4. **f**,**g**, $n = 3$ independent experiments, with similar results; a-GST, anti-GST and IP, immunoprecipitation. **h**, MST assay for direct binding between lytC_22 and hSLAMF4. The dissociation constant ($K_d$) is provided. The blue colour band indicates the time for the instrument to settle. The pink color indicates the MST on time chosen in MO.Affinity Analysis software. **a**–**e**,**h**, Data are the mean ± s.e.m. *P* values were calculated using an unpaired two-tailed Student's *t*-test (**a**,**b**) or one-way ANOVA, followed by Tukey's post-hoc test (**c**–**e**).

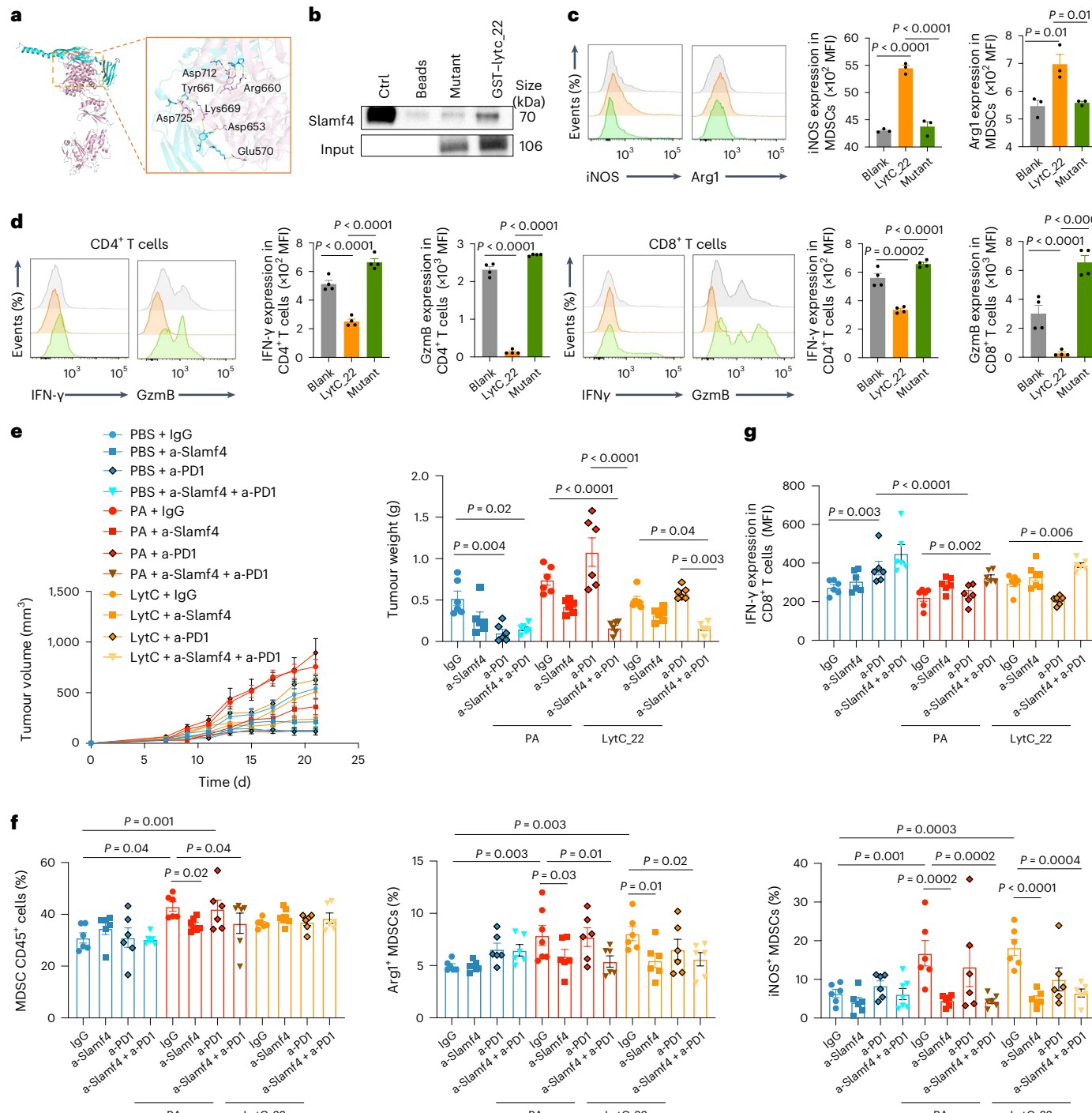

**Fig. 6 | Blocking the lytC_22–Slamf4 interaction can reverse the effect of lytC_22 and _P. anaerobius_ on MDSC modulation. a**, Schematic of the lytC_22–Slamf4 binding sites and their corresponding prediction result (pink, lytC_22; cyan, Slamf4 and yellow dash, hydrogen bond). LytC_22 binds to Slamf4 protein with a binding energy of −416.43 kcal mol⁻¹. LytC_22 is attached to Slamf4 with seven key residues: Glu570, Asp653, Asp725, Tyr661, Asp712, Arg660 and Lys669. **b**, LytC_22 mutant protein was purified following substitution of key binding residues with alanine. GST-tag pulldown of the GST–lytC_22 mutant with recombinant hSLAMF4. **c**, MDSCs were treated with lytC_22 or its mutant. Flow cytometry was used to determine the MFI of Arg1 and iNOS in MDSCs; $n = 3$ biologically independent samples. **d**, Following MDSC pre-treatment with lytC_22 or lytC_22 mutant protein, the expression of IFN-γ and GzmB in CD4⁺ and CD8⁺ T cells was assessed after co-culture with MDSCs; $n = 4$ biologically independent samples. **e–g**, Anti-Slamf4 reversed lytC_22-induced anti-PD1 resistance in the MC38 allograft mouse model; $n = 6$ mice per group. **e**, Tumour growth (left) and tumour weight (left). **f**, Percentage of MDSCs (left) as well as Arg1⁺ (middle) and iNOS⁺(right) MDSCs in tumours from mice in the different treatment groups. **g**, Percentage of IFN-γ⁺CD8⁺ T cells in the tumours. **c–g**, Data are the mean ± s.e.m. $P$ values were calculated using a one-way ANOVA, followed by Tukey's post-hoc test (**c,d**) or a one-way ANOVA, followed by Fisher's least significant difference test (**e–g**).

by significantly increased tumour volume (Fig. 6e) and tumour weight (Fig. 6e) compared with the anti-PD1 only treatment group. *P. anaerobius*- or lytC_22-induced anti-PD1 resistance was largely reversed by treatment with neutralizing anti-Slamf4 ($P < 0.01$; Fig. 6e and Extended Data Fig. 8a). Concordantly, *P. anaerobius* increased the proportion of total ($P < 0.001$), Arg1+ ($P = 0.003$) and iNOS+ ($P = 0.001$) MDSCs in MC38 allografts (Fig. 6f), which abrogated the induction of IFN-γ+ and GzmB+ CD8+ T cells by anti-PD1 treatment (Fig. 6g and Extended Data Fig. 8b). LytC_22 alone resulted in increased Arg1+ ($P = 0.003$) and iNOS+ ($P < 0.001$) MDSCs (Fig. 6f) together with a reduction in IFN-γ+ and GzmB+ CD8+ T cells compared with the PBS group after anti-PD1 treatment (Fig. 6g and Extended Data Fig. 8b). Importantly, anti-Slamf4 reversed lytC_22- or *P. anaerobius*-mediated MDSC activation (Fig. 6f) and T cell suppression (Fig. 6g and Extended Data Fig. 8b), as determined by flow cytometry, thereby restoring anti-PD1 efficacy. Collectively, our results suggest that *P. anaerobius*-derived lytC_22 protein promotes the immunosuppressive function of MDSCs and anti-PD1 resistance via the Slamf4 receptor of MDSC. Slamf4 blockade improves anti-PD1 efficacy in the context of lytC_22 treatment.

To evaluate whether the integrin $\alpha_2\beta_1$–CXCL1 and lytC_22–Slamf4 axes function independently of each other, we determined the effect of anti-Slamf4 on *P. anaerobius*-induced CXCL1 by CRC in vitro as well as in MC38 allografts. Anti-Slamf4 had no effect on *P. anaerobius*-induced CXCL1 in either model (Extended Data Fig. 8c–e). Furthermore, lytC_22 administration alone had no effect on CXCL1 (Extended Data Fig. 8c). Similarly, blockade of integrin $\alpha_2\beta_1$–CXCL1 by RGDS peptide did not modulate the production of lytC_22 by *P. anaerobius* co-cultured with CRC cells (Extended Data Fig. 8f,g). These data imply that *P. anaerobius* promotes MDSCs in CRC through two independent mechanisms.

## Discussion

In this study we uncovered an unappreciated role for intratumoral *P. anaerobius* in modulating ICB therapy efficacy in CRC. We demonstrated that *P. anaerobius*-derived proteins mediate a three-way crosstalk between *P. anaerobius*, CRC cells and MDSCs, driving sequential MDSC recruitment and activation (Extended Data Fig. 9). As a consequence, *P. anaerobius*-infected CRC harbours an immunosuppressive tumour microenvironment that dampens the T cell-dependent antitumor response following ICB therapy.

Myeloid-derived suppressor cells are major immunosuppressive cells in the tumour microenvironment. Accumulating evidence has associated MDSCs with cancer progression and resistance to ICB therapy[17,24]. Nevertheless, whether intratumoral bacteria play a role in enriching MDSCs in CRC remains unclear. Here we showed that *P. anaerobius* colonization increases the intratumoral MDSC abundances in CRC tumorigenesis in both *Apc*min/+ and AOM and DSS CRC mouse models. Increased MDSCs in turn create a permissive tumour microenvironment via antagonistic actions on antitumor CD8+ and CD4+ T cells. Accordingly, *P. anaerobius* treatment causes non-response to anti-PD1 therapy in MC38 allografts and AOM and DSS-driven CRC, which is associated with elevated MDSC infiltration and impaired cytotoxic T cell function. In agreement with our identification, several pathogenic bacteria were recently shown to skew the density and composition of tumour-infiltrating immune cells in CRC[25,26], thus impairing antitumor immunosurveillance. For instance, an increased abundance of Prevotellaceae correlates with exhausted intratumoral CD8+ T cells[27], whereas *Streptococcus gallolyticus* enriches the CD11b+ myeloid populations in AOM and DSS-induced CRC[28]. These findings encouraged us to explore the molecular basis of the *P. anaerobius*–MDSCs interplay in CRC.

Intratumoral accumulation of MDSCs consists of two discrete steps[29] involving the recruitment and expansion of immature myeloid cells, followed by activation to acquire immunosuppressive properties. We revealed that *P. anaerobius* facilitates both processes via distinct mechanisms. First, *P. anaerobius* promotes the CRC cell-mediated

recruitment of MDSCs. Co-culture of *P. anaerobius* with CRC cells stimulates the secretion of CXCL1, a chemokine reported to participate in the chemotaxis of CXCR2+ MDSCs[30]. Knockdown of CXCL1 and blockade of CXCR2 both abrogated *P. anaerobius*-induced MDSC migration. Corroborating these in vitro results, *P. anaerobius*-infected mice had elevated CXCL1 in CRC tissues as well as in circulation. Mechanistically, direct interaction of *P. anaerobius* with CRC cells via PCWBR2–integrin $\alpha_2\beta_1$ signalling[7] promotes NF-κB activation and nuclear translocation, which in turn transcriptionally activates CXCL1 expression and secretion. Together, these observations suggest that the interplay between intratumoral *P. anaerobius* and CRC cells promotes the CXCL1–CXCR2 axis to attract MDSCs to CRC tumours where they accumulate .

Beyond its effect on MDSC recruitment by interacting with CRC cells, we reveal that *P. anaerobius* directly promotes the activation of MDSCs via its secreted protein lytC_22. LytC_22 boosts the T cell suppressive function of MDSCs by inducing the expression of ARG1 and iNOS. Moreover, intratumoral injection of recombinant lytC_22 into CRC tumours promotes anti-PD1 therapy resistance in vivo by inducing intratumoral ARG1+ and iNOS+ MDSCs, thereby causing exacerbated T cell suppression. Co-immunoprecipitation assays and ligand–target interaction analyses by MST demonstrated that lytC_22 functions as a ligand for Slamf4 receptor expressed on MDSCs. Notably, antibody-directed inhibition of Slamf4 receptor reversed lytC_22-induced MDSC activation, inferring the role of lytC_22–Slamf4 in MDSC activation following their recruitment to CRC tumours. LytC proteins have been shown to dampen host immunity to facilitate *Streptococcus pneumoniae* infection and invasive disease[31], which is in line with our data showing the immunomodulatory effect of *P. anaerobius* lytC_22. On the other hand, Slamf4 receptor (also known as CD244) positivity in MDSCs correlates with their inhibitory function in antitumor immunity[23], supporting the idea that the lytC_22–Slamf4 interaction functionally activates MDSCs to impair T cells in the CRC tumour immune microenvironment. Several metabolites such as trimethylamine N-oxide and short-chain fatty acids secreted by intratumoral bacteria have been shown to modulate CD8+ T cell immunity[32–35]. The pathogenic bacterium *F. nucleatum* secretes Fap2 protein, which engages the inhibitory receptor TIGIT to arrest natural killer cell activation within the tumour microenvironment[36]. Our work therefore identifies an interaction between *P. anaerobius* and MDSCs, and expands our knowledge on the interplay between intratumoral microbiota and host immunity in modulating tumorigenesis.

Considering the pivotal role of *P. anaerobius*-mediated MDSC activation in CRC, we thus sought to devise therapeutic strategies to overcome *P. anaerobius*-driven immune suppression and anti-PD1 resistance. To this end, we first targeted the *P. anaerobius*–CRC cell interplay using RGDS peptide, an integrin $\alpha_2\beta_1$ inhibitor. The RGDS peptide reversed *P. anaerobius*-induced CXCL1 secretion by CRC cells and attenuated chemotaxis of MDSCs both in vitro and in vivo. Furthermore, RGDS peptide treatment of *P. anaerobius*-infected CRC reactivated intratumoral CD8+ T cells to restore the therapeutic efficacy of anti-PD1. We also pursued an alternative strategy targeting *P. anaerobius*–MDSCs via lytC_22–Slamf4 to suppress MDSC activation. As expected, blockade of Slamf4 using neutralizing antibody reversed lytC_22-induced MDSC activation, T cell suppression and anti-PD1 therapy resistance. These results suggest integrin $\alpha_2\beta_1$ and Slamf4 as potential therapeutic targets for overcoming ICB resistance in patients with CRC who have enrichment of *P. anaerobius* in their gut microbiota.

In summary, our study identified *P. anaerobius* as an intratumoral pathogenic bacterium that promotes immunosuppression and non-response to ICB therapy in CRC. We revealed a two-step underlining mechanism whereby *P. anaerobius* first interacts with CRC cells to promote CXCL1-driven MDSC chemotaxis into CRC and then by direct *P. anaerobius*–MDSC interplay via lytC_22–Slamf4 to endow MDSCs with an immunosuppressive function. This ultimately culminates in T cell suppression and ICB resistance. Blocking the interaction

between *P. anaerobius* and host tumour cells to impair MDSC trafficking and activation is a promising strategy to improve the efficacy of immunotherapy in CRC.

## Methods

### Bacterial strains and culture conditions

*P. anaerobius* strain VPI 4330 (catalogue number 27337) and *E. coli* strain MG1655 (catalogue number 700926) were purchased from the American Type Culture Collection (ATCC). *P. anaerobius* was cultured in Wilkins–Chalgren anaerobe broth (Thermo Fisher Scientific, CM0643) at 37 °C under anaerobic conditions (80% $N_2$, 10% $H_2$ and 10% $CO_2$). *E. coli* strain MG1655 was cultured overnight at 37 °C in Luria–Bertani medium with shaking (220 rpm min$^{-1}$).

### Mouse models

The Animal Experimentation Ethics Committee of the Chinese University of Hong Kong approved all of the animal experiments conducted in this study. Mice were kept in specific pathogen-free facilities, following a cycle of 12 h light and 12 h dark. Food and water were provided ad libitum. No statistical methods were used to pre-determine sample size but our sample sizes are similar to those in previous reports[7]. The mice were randomly assigned to different groups and no animals were excluded from the analyses. Data collection and analysis were not conducted in a blinded manner.

For the *Apc*$^{min/+}$ mice model, male C57BL/6 *Apc*$^{min/+}$ mice (six weeks old) were administered an antibiotic cocktail in their drinking water (0.2 g l$^{-1}$ ampicillin, neomycin and metronidazole, and 0.1 g l$^{-1}$ vancomycin) for two weeks to deplete their gut microbiota. One week after the antibiotic treatment, the mice were orally administrated with $1 \times 10^8$ c.f.u. *P. anaerobius*, *E. coli* MG1655 or PBS daily for seven weeks.

For the AOM-induced CRC mouse model, male conventional C57BL/6 wild-type mice were subjected to intraperitoneal injections of AOM (10 mg kg$^{-1}$) once a week for six weeks, followed by the same treatment as the *Apc*$^{min/+}$ mice model for 24 weeks. For integrin $\alpha_2\beta_1$ blockade, RGDS peptide (200 μg per mouse) was given to two additional groups of mice (one group treated with PBS and the other treated with *P. anaerobius*) three times a week by intraperitoneal injection for 24 weeks.

To study the tumour-promoting effect of *P. anaerobius* in an AOM and DSS-induced CRC model, male conventional C57BL/6 wild-type mice were injected with AOM (10 mg kg$^{-1}$) on day 0. After 7 d, the mice were provided with drinking water containing 1% DSS (MPBio) for 7 d, followed by drinking water for 14 d and then exposed to two more treatment cycles with 1% (wt/vol) DSS. On days 14, 35 and 56, we administered *P. anaerobius* ($1 \times 10^8$ c.f.u.), *E. coli* ($1 \times 10^8$ c.f.u.) or PBS to the mice daily for two weeks after suspending DSS treatment.

To study the effect of *P. anaerobius* on anti-PD1 therapy using an AOM and DSS-induced CRC model, male conventional C57BL/6 wild-type mice were injected with AOM (10 mg kg$^{-1}$) on days 0 and 19. The animals were given 1% DSS solubilized in water 3 times for 5 d each time, starting on days 7, 19 and 38, followed by regular water for 7 d. After drinking an antibiotic cocktail for two weeks, the mice were given *P. anaerobius* ($1 \times 10^8$ c.f.u.) or PBS orally daily until the end of the experiment. Monoclonal anti-mPD1 (100 μg) or an isotype control monoclonal antibody was administered by intraperitoneal injection twice a week for a total of three weeks after beginning *P. anaerobius* treatment for 10 d.

For the allograft experiments, MC38 cells ($5 \times 10^5$) were subcutaneously injected into the right flank of male C57BL/6 mice. After 6 d, *P. anaerobius* ($1 \times 10^7$ c.f.u. per 50 μl), an equal volume of PBS and RGDS peptide (200 μg per 50 μl) were administered by multipoint intratumoral injection every 3 d until the end of the experiments. The mice were intraperitoneally injected with isotype control or anti-mouse PD1 (BioXcell) twice a week (100 μg per injection) after the first bacterial treatment. For the MDSC depletion, mice were treated with anti-mouse Ly6G or control rat IgG (250 μg per injection; BioXcell) via

intraperitoneal injections every 48 h for 14 d after the first bacterial treatment. To validate the effect of recombinant lytC_22 protein in vivo, lytC_22 (25 μg protein in 50 μl PBS) was given by multipoint intratumoral injection every 3 d once tumours had become measurable. For Slamf4 receptor blockade, the mice were intravenously injected, via the tail vein, three times a week (total of six doses) with anti-CD244 (eBioscience) or isotype antibody (both at a dose of 50 μg antibody in 100 μl PBS) starting after the first lytC_22 treatment.

### *P. anaerobius* quantification in fecal and mucosal samples from naive and AOM and DSS-induced CRC mice

Six-week-old male conventional C57BL/6 wild-type mice were divided into two groups. One group was injected with AOM (10 mg kg$^{-1}$) on days 0 and 19, and 1% DSS solubilized in water was given to the animals three times for 5 d each time, starting on days 7, 19 and 38, followed by regular water for 7 d. The mice in the other group were fed in parallel under normal conditions. After tumour formation had occurred, the mice were orally gavaged with *P. anaerobius* ($1 \times 10^8$ c.f.u.) daily for two weeks. The mice were sacrificed at specific time points (days 0–5 and 7) following *P. anaerobius* gavage to obtain stool samples and colon tissue. Stool DNA was extracted from collected samples using a Quick-DNA fecal/soil microbe miniprep kit (Zymo Research). Amplification and detection of *P. anaerobius* DNA in the stool samples were performed using a Universal SYBR Green Master reaction on the QuantStudio 7 flex system (Thermo Fisher Scientific). Fluorescence in situ hybridization was employed to assess the abundance of *P. anaerobius* in colon tissue samples.

### Cell culture

The colon cancer cell lines HCT116, Caco-2 and MC38 were obtained from the ATCC. The normal colon epithelial cell line NCM460 was obtained from INCELL. The cells were cultured in DMEM medium supplemented with 10% fetal bovine serum (FBS), 100 U penicillin and 100 μg ml$^{-1}$ streptomycin. The cells were cultured at 37 °C in a cell culture incubator with 5% $CO_2$. The bacterial co-culture assay was performed according to previously reported protocols[7]. For the inhibitor-treatment experiments, cells were pretreated with 30 μM NF-κB inhibitor JSH-23 (MCE) or 100 μM integrin $\alpha_2\beta_1$ inhibitor RGDS peptide (MCE) for 24 h and then treated with *P. anaerobius*.

### Conditional culture medium preparation

Cancer cells or blank medium were incubated with *P. anaerobius* for 4 h under anaerobic conditions. The medium containing bacteria was then replaced with DMEM medium supplemented with 10% FBS, 1% penicillin–streptomycin and 200 mg ml$^{-1}$ gentamycin for 2 h, which was used to kill all the extracellular bacteria. The medium containing the killed bacteria was replaced with full DMEM medium. After culturing for 24 h, the culture medium was collected, necrotic cell debris was removed by centrifugation at 2,000*g* for 10 min and conditioned supernatant was obtained by filtration through a 0.45 μm membrane filter (Millipore).

### CXCL1 detection

CXCL1 protein in cancer cell culture medium and serum was detected using ELISA assay kits (Abcam) according to the manufacturer's instructions. To determine the level of CXCL1 in tumour tissue, whole protein was isolated from tumour tissue using Cytobuster (Novagen) and total 50 μg protein was used to detect CXCL1 using an ELISA kit (Abcam).

### Integrin $\alpha_2\beta_1$ and CXCL1 knockdown

Cells were transfected with siRNA targeting integrin $\alpha_2\beta_1$, CXCL1 or CXCL3, or non-targeting RNA using Lipofectamine 2000 (Thermo Fisher Scientific) according to the manufacturer's instructions. Integrin $\alpha_2\beta_1$ were transiently silenced using a pool of two different siRNAs— that is m siRNA to *ITGA2* (Ambion, S7537) and *ITGB1* (GenePharma). *CXCL1* and *CXCL3* were silenced by designed siRNA (Ambion). A silencer

negative transcription control was introduced into each experiment. Following 6 h of transfection, the medium was replaced with full DMEM medium supplemented with serum and antibiotic. *CXCL1, CXCL3* and integrin $\alpha_2\beta_1$ knockdown was validated by ELISA and western blotting.

## ScRNA-seq

Tumours isolated from AOM and DSS-induced mice were minced and digested in RPMI1640 medium containing 1 mg ml$^{-1}$ collagenase type IV (Roche) and 0.5 mg ml$^{-1}$ DNase I for 20 min at 37 °C with gentle rocking. The cell suspension was filtered through a 70 μm cell strainer. Single cells were stained with BV421-labelled anti-CD45.2 (BioLegend) and 7AAD viability staining solution (BioLegend) for 30 min at 4 °C, washed and resuspended in PBS (2% FBS) buffer for flow cytometric sorting using a FACS Aria II cell sorter (BD Biosciences). Sorted CD45$^+$7AAD$^-$ cells were diluted with trypan blue and counted using a haemocytometer. Tumours from three different mice were pooled per sample. Single-cell capturing for scRNA-seq and library preparation were done by the SBS core laboratories in CUHK. Briefly, a single-cell suspension of 500–1,000 cells μl$^{-1}$ in FACS buffer was added to a real-time PCR master mix to sample up to 8,000 cells; these were then combined with Single Cell 3′ gel beads and dispense oil onto a single cell 3′ chip (10x Genomics). RNA transcripts from individual cells were uniquely barcoded and reverse transcribed in droplets. Complementary DNA molecules were pre-amplified and pooled, followed by library construction. All libraries were quantified using a Qubit fluorometer and real-time PCR was performed on a LightCycler 96 system (Roche Life Science). The size profiles of pre-amplified cDNA and sequencing libraries were examined using the Agilent high sensitivity D5000 and D1000 ScreenTape systems, respectively. All single-cell libraries were sequenced on a NextSeq 500 system (Illumina) using a NextSeq 500 high output v2 kit (Illumina).

## Analysis of scRNA-seq data

Raw data (raw reads) of FASTQ files were assembled from the Raw BCL files using Illumina's bcl2fastq converter. Pre-processing of the sequencing results were conducted using the 10x Genomics Cell Ranger pipeline (v6.0.0) with default settings and analysis was performed in R. The dataset was filtered to exclude low-quality cells using the following criteria: (1) <5% or >95% unique molecular identifier or gene count, or (2) proportion of mitochondrial genes > 20%. All of the downstream analyses were based on the clean data with high quality. All downstream single-cell analyses were then performed using Cell Ranger and Seurat[33,34], unless mentioned otherwise. Briefly, for each gene and each cell barcode (filtered by Cell Ranger), unique molecule identifiers were counted to construct digital expression matrices. Secondary filtration performed using Seurat, where a gene was deemed to be expressed if it showed expression in more than three cells. Each cell was mandated to have a minimum of 200 expressed genes. Cells expressing fewer than 200 genes or multi-cells were excluded from futher analysis. Cell Ranger count takes FASTQ files and performs alignment, filtering, barcode counting and unique molecular identifier counting. It uses the Chromium cellular barcodes to generate feature barcode matrices by Cell Ranger count or Cell Ranger aggr and reruns the dimensionality reduction, clustering and gene expression algorithms using Cell Ranger default parameter settings.

## Blocking of Slamf4 on MDSCs

For blocking of lytC_22–Slamf4 interaction, primary MDSCs were pre-incubated with purified monoclonal anti-mouse Slamf4 (Invitrogen, eBio244F4) at 20 μg ml$^{-1}$ for 1 h on ice before the addition of lytC_22 protein.

## Human chemokine array

Chemokines were quantified using a human chemokine array kit (Abcam, ab169812) according to the manufacturer's instructions.

Briefly, array membranes were blocked in blocking buffer at room temperature for 1 h and then incubated overnight at 4 °C with 1.5 ml of conditioned culture medium from Caco-2 cells pre-incubated with or without *P. anaerobius*. The next day, the membranes were washed and incubated overnight with 1 ml primary biotin-conjugated antibody mixture at 4 °C. After washing, the membranes were incubated with 2 ml horseradish peroxidase-conjugated streptavidin at room temperature for 2 h. The membrane spots were exposed to ECL plus western blotting detection reagents. Positive controls on each membrane were used to normalize spot intensities across different groups.

## Isolation of MDSCs and MDSC migration assay

We collected MDSCs from the splenocytes of tumour-bearing mice using an MDSC isolation kit. To observe MDSC migration, $1 \times 10^5$ freshly isolated mouse MDSCs were added to transwell inserts with DMEM containing 10% FBS, DMEM from tumour cells infected with bacteria or DMEM containing different concentrations of conditioned medium from *P. anaerobius* were put in the bottom chamber of the transwell. Migrated MDSCs were counted under microscopy. To block the CXCR2 receptor, freshly isolated MDSCs were pre-incubated with 10 μM SB265610 (CXCR2 antagonist) for 1 h at 37 °C and then seeded in transwell inserts. For the drug treatment assay, MDSCs were cultured in RPMI1640 medium containing 10% FBS, 10 ng ml$^{-1}$ IL-6 and 10 ng ml$^{-1}$ GM-CSF to sustain its activity.

## Isolation of colonic lamina propria cells and flow cytometry

Colonic lamina propria cells were isolated from mice as described previously. Briefly, colon tissue was minced and incubated in a digestion buffer consisting of PBS, 1% BSA, 1 mg ml$^{-1}$ collagenase type IV and 0.5 mg ml$^{-1}$ DNase I for 30 min at 37 °C. The cell suspensions were filtered through a 70 μm strainer and then centrifuged at 750$g$ for 10 min. For surface marker staining, lamina propria cells were stained with fluorescent dye-conjugated antibodies in cell staining buffer for 20 min. For intracellular staining, the cells were stimulated for 4 h in RPMI complete medium with 30 ng ml$^{-1}$ phorbol 12-myristate 13-acetate (Abcam) and 1 μg ml$^{-1}$ ml ionomycin (STEMCELL Technologies) in the presence of 2.5 μg ml$^{-1}$ monomycin (BioLegend). The cells were then stained with cell surface markers. Samples for flow cytometry analysis were fixed with cell fixation buffer (BioLegend), permeabilized with FOXP3 Perm buffer (eBioscience) according to the manufacturer's recommendations and stained with intracellular antibody. Positive cells were acquired and quantified using the FlowJo software. Samples for single-cell sequencing were sorted in collaboration with the LKSIHS core facility.

## Evaluation of MDSC immunosuppressive activity

T cells were collected from the splenocytes of non-tumour-bearing mice using a T cell isolation kit (STEMCELL Technologies), followed by CFSE staining. An EasySep mouse MDSC (CD11b$^+$Gr-1$^+$) isolation kit (STEMCELL Technologies, 19867) was used to collect MDSCs from the spleen of tumour-bearing mice on days 17–19 after subcutaneous inoculation with $5 \times 10^5$ cells MC38 cells. CFSE-labelled T cells ($1 \times 10^5$ cells) and $1 \times 10^5$ cells, the half ($5 \times 10^4$) of MDSCs were seeded in a 96-well plate. T cell proliferation was induced with Dynabeads mouse T-activator CD3/CD28 (Thermo Fisher Scientific, 11452D) for 3 d, followed by flow cytometry analysis using the FlowJo software to evaluate the CFSE dilution rate and the percentage of functional cells in the T cell subpopulations.

## Immunofluorescence staining

Antigen retrieval was performed by incubation of the slides in Tris–HCl buffer at 100 °C for 20 min, followed by two washes with PBST (PBS containing 0.1% Tween 20). The slides were incubated with 1% BSA for 1 h to block non-specific binding of immunoglobulin and then overnight incubation with primary antibody to CD11b

(dilution, 1:200; Abcam, ab133357) and Gr-1 (dilution, 1:50; BioLegend, 108448) at 4 °C. Following two washes with PBST, the specimens were incubated with Alexa Fluor 488 secondary antibody (dilution, 1:500; Jackson Immuno, 111-545-003 (X2)). The accuracy of automated measurements was confirmed through independent evaluation by two pathologists. Cells stained with the indicated antibodies were counted at ×63 magnification in at least ten fields per section.

## Quantification of the *P. anaerobius* relative abundance in the TCGA colon adenocarcinoma cohort

We downloaded the unaligned RNA-seq data, in FASTQ format, of colon adenocarcinoma from the TCGA database[37]. The pathseq[38] pipeline was used to map the unaligned RNA-seq of TCGA colon adenocarcinoma samples to gut-related bacteria after removing human reads and low-quality reads. The pre-built host genome was obtained from the GATK Resource Bundle FTP server in /bundle/pathseq/. The microbe references used here included 1,520 cultivated bacterial genomes[39] and colon cancer-related bacteria that were identified through extensive and statistically rigorous validation. The normalized score generated by pathseq was used as the relative transcriptome abundance for each species. We calculated the mean expression level of marker genes as the signature score for each immune cell type. For MSDC cells, the markers CD33, ITGAM and ARG1 were utilized, whereas effector T cells were characterized using CD44, CD8A and IFN-γ.

## Fluorescence in situ hybridization

A *P. anaerobius* Alexa Fluor 488-conjugated specific probe (5'-ATATCTACGATGCCGTAAATATA-3') was labelled with Spectrum-Green (Thermo Fisher Scientific). Fresh colon tissues were embedded in paraffin and cut into 5 μm sections. The colon tissue sections were then subjected to hybridization using hybridization buffer (0.9 M NaCl, 20 mM Tris–HCl pH 7.3 and 0.01% SDS) mixed with oligonucleotide probe (1:50). After incubation for 48 h at 37 °C in a dark humid chamber, each slide was rinsed with sterile double-distilled water for 5 min, air-dried in the dark and finally mounted using ProLong gold antifade mountant with 4,6-diamidino-2-phenylindole (DAPI; Thermo Fisher Scientific).

## Prediction of lytC_22–Slamf4 binding sites and construction of lytC_22 mutant protein

The protein structures of lytC_22 (UniProt identifier: A0A379CGB7) and CD244 (UniProt identifier: P02679) were retrieved from the UniProt database (https://www.uniprot.org/). The HDOCK software was employed to predict the protein–protein binding conformation, with selection based on the scoring function and identification of the most negative energy conformation. Molecular dynamics simulations were performed using Gromacs 2020, applying the AMBER99SB-ILDN force field for proteins and employing the TIP3P dominant water model. To neutralize system charge, sodium or chloride ions were introduced according to the docking results. The system was then subjected to a 100 ns molecular dynamics simulation under the isothermal-isobaric ensemble.

Throughout the molecular dynamics simulation, key properties of interest such as system energy, the root mean square deviation of protein structure fluctuations and root mean square fluctuation analysis of amino acid residues were monitored over time to investigate changes in system-related characteristics. Following optimization of the binding conformation, the Pymol 2.1 software was utilized to visualize and map the protein interaction sites.

To obtain the mutant protein that is unable to bind with Slamf4, after determining which key residues on lytC_22 bind to Slamf4, alanine residues were introduced to replace these binding residues. LytC_22 mutant was then purified as described earlier.

## Silver staining and protein identification

The 50–100 kDa fraction from PA-CM was separated by 10% SDS–PAGE. Silver staining was performed following the instructions provided by the manufacturer (Thermo Fisher Scientific). After staining, specific bands located between 50 and 100 kDa were excised into 1.5 ml plastic tubes for in-gel digestion and mass spectrometry analysis by Shanghai Biotree Biotech.

## Expression, purification and in vitro activity of lytC_22

LytC_22 proteins were produced from *E. coli* BL21 (DE3) containing plasmids pGEX-4T1 for lytC_22. The cells were induced by incubation with 0.5 mM isopropyl-thio-β-D-galactopyranoside (Sangon Biotech) for 16 h at 25 °C. Next, the bacterial pellets were collected and resuspended in PBS containing 1 mg ml$^{-1}$ lysosome. After three freeze–thaw cycles, the samples were centrifuged at 4 °C and 20,000$g$ for 15 min and the supernatant containing the GST–lytC_22 protein was collected. GST–lytC_22 was further purified using a GST-tag protein purification kit (Beyotime). The GST tag was removed through incubation with 80 U thrombin protease (Cytiva) for 16 h at 25 °C, followed by incubated with glutathione Sepharose resins for 1 h at 4 °C. The thrombin was then removed by passing through a HiTrap Benzamidine FF column (Cytiva). Fractions containing the target protein were concentrated and further purified by centrifugation at 4,000$g$ and 4 °C for 1 h using 50 kDa MWCO Amicon Ultra-4 centrifugal filters. To detect the effect of lytC_22 on MDSCs, 10 or 20 nM purified lytC_22 was added into RPMI 1640 medium when culturing primary MDSCs. After treatment for 24 h, the MDSCs were collected for further experiments.

## RNA extraction and quantitative PCR

Total RNA was extracted from cells and flash-frozen colon tissues using TRIzol reagent (Thermo Fisher Scientific). Complementary DNA was synthesized from 1 μg of total RNA using a PrimeScript RT–PCR kit (Takara). Quantitative real-time PCR was performed using a LightCycler 480 real-time PCR system (Roche Applied Sciences). The target mRNA level was normalized to that of β-actin and calculated using the $2^{-\Delta\Delta Ct}$ method.

## GST pulldown assay

Primary MDSCs were lysed using IP lysis buffer (1% NP-40, 150 mM NaCl, 5 mM MgCl$_2$ and 25 mM HEPES pH 7.5) for 1 h and total protein from MDSCs was acquired by centrifugation at 20,000$g$ and 4 °C for 15 min. For the GST pulldown assay, GST–lytC_22 and MDSC cell lysate or recombinant Slamf4 protein (ABclonal, RP00167) were incubated overnight at 4 °C with anti-GST (dilution, 1:200; Cell Signaling Technology, 2624S) or anti-Slamf4 (dilution, 1:100; Cell Signaling Technology, 54560S). Protein A/G beads were added to the samples and incubated at 4 °C for 2 h. The beads were then washed five times with IP lysis buffer and boiled in SDS–PAGE gel loading buffer. The eluted proteins were separated by SDS–PAGE and analysed by western blotting.

## Western blots

Total protein was separated by 10% SDS–PAGE and then transferred onto polyvinylidene difluoride membranes for 2 h. After blocking with 5% non-fat milk for 1 h at room temperature, the blots were incubated with primary antibodies overnight at 4 °C and then washed. Next, the blots were incubated with secondary antibodies for 1 h at room temperature. The protein bands were exposed to ECL plus western blotting detection reagents. The following primary antibodies were used at a dilution of 1:1,000: anti-integrin α2 (Abcam, ab133557; dilution,), anti-integrin β1 (Abcam, ab179471), anti-β-actin (Cell Signaling Technology, 4970S), anti-phospho-NF-κB p65 (Cell Signaling Technology, 3033S), anti-NF-κB p65 (Cell Signaling Technology, 8242S), anti-lamin A/C (Cell Signaling Technology, 4777) and anti-Slamf4 (Cell Signaling Technology, 54560). Anti-GST tag (Cell Signaling Technology,

2625) was used at a dilution of 1:5,000. Quantification was performed using Image Lab 6.1 (Bio-Rad).

## MST assay

Analyses of direct binding of between Slamf4 and lytC_22 were performed using a Monolith NT.115 instrument (NanoTemper Technologies). Recombinant Slamf4 was fluorescently labelled in advance. To measure binding between Slamf4 and GST–lytC_22, gradient concentrations of GST–lytC_22 (11, 5.5, 2.75, 1.375, 0.6875 and 0.34375 µg µl$^{-1}$) were incubated with 0.2 µg µl$^{-1}$ pre-labelled recombinant Slamf4 at a ratio of 1:1 (vol/vol) for 30 min at room temperature. For measurement, the incubated mixture was loaded into capillaries (NanoTemper Technologies) and loaded onto a Monolith NT.115 instrument. A cap scan of all the capillaries was performed and the MST experiment was conducted using auto LED/excitation power and medium MST power. Data were analysed using the NT Analyses 1.5.41 software.

## Statistical analyses

A Mann–Whitney $U$-test or Student's $t$-test was used to analyse the differences between two sample groups. A one-way ANOVA was used to analyse the differences between multiple groups. Data are presented as the mean ± s.e.m. Pilot experiments were used to estimate the sample size. Statistical analysis was performed using the GraphPad Prism version 9 software (GraphPad Software). Comparisons with $P < 0.05$ were considered to be statistically significant.

## Reporting summary

Further information on research design is available in the Nature Portfolio Reporting Summary linked to this article.

## Data availability

All data are available in the main text, extended data or supplementary materials. The RNA-seq data have been deposited at the NCBI under the accession code PRJNA1070509. Source data are provided with this paper.

## Code availability

All customized scripts used in this study were incorporated and uploaded at https://github.com/YanqiangDing/CRC_PD1_scRNA

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

## Acknowledgements

This project was supported by the National Key R&D Program of China (grant no. 2020YFA0509200/2020YFA0509203), RGC Research Impact Fund Hong Kong (grant no. R4032-21F), Research Talent Hub-Innovation and Technology Fund Hong Kong (grant no. ITS/177/21FP), Shenzhen–Hong Kong–Macao Science and Technology Program (Category C) Shenzhen (grant no. SGDX20210823103535016), RGC Theme-based Research Scheme (grant no. T12-703/19-R), HMRF Hong Kong (grant nos. 7181256 and 9202626), RGC–CRF Hong Kong (grant no. C4039-19G) and Vice-Chancellor's Discretionary Fund CUHK. The scRNA-seq was performed at the Single Cell Omics Core, School of Biomedical Sciences, Faculty of Medicine, The Chinese University of Hong Kong.

## Author contributions

Y.L. performed most of the experiments and drafted the paper. C.C.W. supervised the study and revised the paper. Y.D. analysed the scRNA-seq data. D.H. analysed TCGA and metagenomic data. H.H. and M.G. provided technical support for gene-editing in bacteria. J.W. and H.C.-H.L. assisted with the mouse experiments. A.H.-K.C. analysed the immunobiological staining data. J.Y. designed and supervised the study and revised the paper.

## Competing interests

The authors declare no competing interests.

## Additional information

**Extended data** is available for this paper at https://doi.org/10.1038/s41564-024-01695-w.

**Correspondence and requests for materials** should be addressed to Jun Yu.

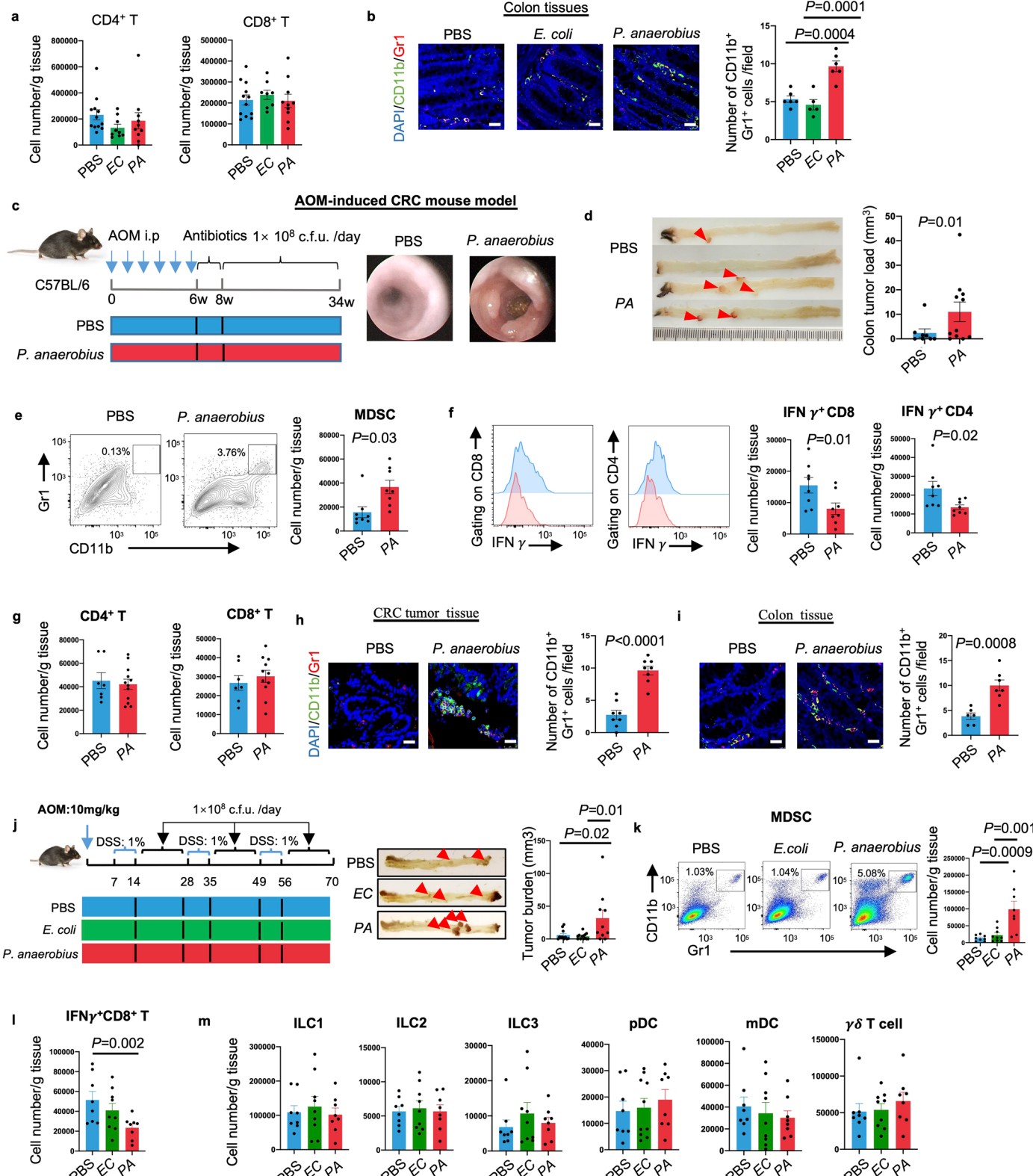

**Extended Data Fig. 1 | See next page for caption.**

**Extended Data Fig. 1 | _P. anaerobius_ promotes an immune suppressive tumour microenvironment in CRC progression. a**. Absolute number of CD4$^+$T cells and CD8$^+$T cells in colonic lamina propria (LP) cells from _Apc_ $^{min/+}$ mice. PBS (n = 12), _E.coli_ (n = 9) and _P. anaerobius_ (n = 10). **b**. Representative immunofluorescent staining picture and quantification of CD11b$^+$Gr-1$^+$cells in colon from _Apc_ $^{min/+}$ mice. Scale bar: 25μm. **c**. Experimental design for _P. anaerobius_ treatment in AOM-induced CRC mouse model. Representative images of mouse colonoscopy at week 33 (right). **d**. Representative images of colon at sacrifice (left). Tumour number and volume. PBS (n = 8), _P. anaerobius_ (n = 11). **e**. Absolute number of MDSCs, (**f**) IFNγ$^+$CD8$^+$T cells and IFNγ$^+$CD4$^+$T cells in colonic LP. 8 mice were used in each group, including PBS (n = 8), _P. anaerobius_ (n = 8). **g**. Absolute number of CD4$^+$T cells and CD8$^+$T cells in colonic LP cells. PBS (n = 7) and _P. anaerobius_ (n = 11). **h-i**. Representative immunofluorescent staining picture and quantification of CD11b$^+$Gr-1$^+$cells in tumour (**h**) and colon (**i**). Scale bar: 25 $\mu$m. **j**. Experimental design for _P. anaerobius_ treatment in AOM/DSS-induced CRC mouse model (Right). Representative images of colon at sacrifice (middle). Tumour burden(right). PBS (n = 11), _E.coli_ (n = 12) and _P. anaerobius_ (n = 10). **k**. Percentage of MDSCs, (**l**) IFNγ$^+$CD8$^+$T cells, and (**m**) Innate lymphoid cells (ILCs), plasmacytoid dendritic cells (pDCs), myeloid dendritic cells (mDCs), γδ T cells in the colonic lamina propria (LP). **k-m** PBS (n = 8), _E.coli_ (n = 9) and _P. anaerobius_ (n = 8). Data are presented as mean ± SEM. P values were calculated by one-way ANOVA followed by Tukey's post-hoc test (**a-b, j-l**) and unpaired two-tailed Student's t-test (**d-i**).

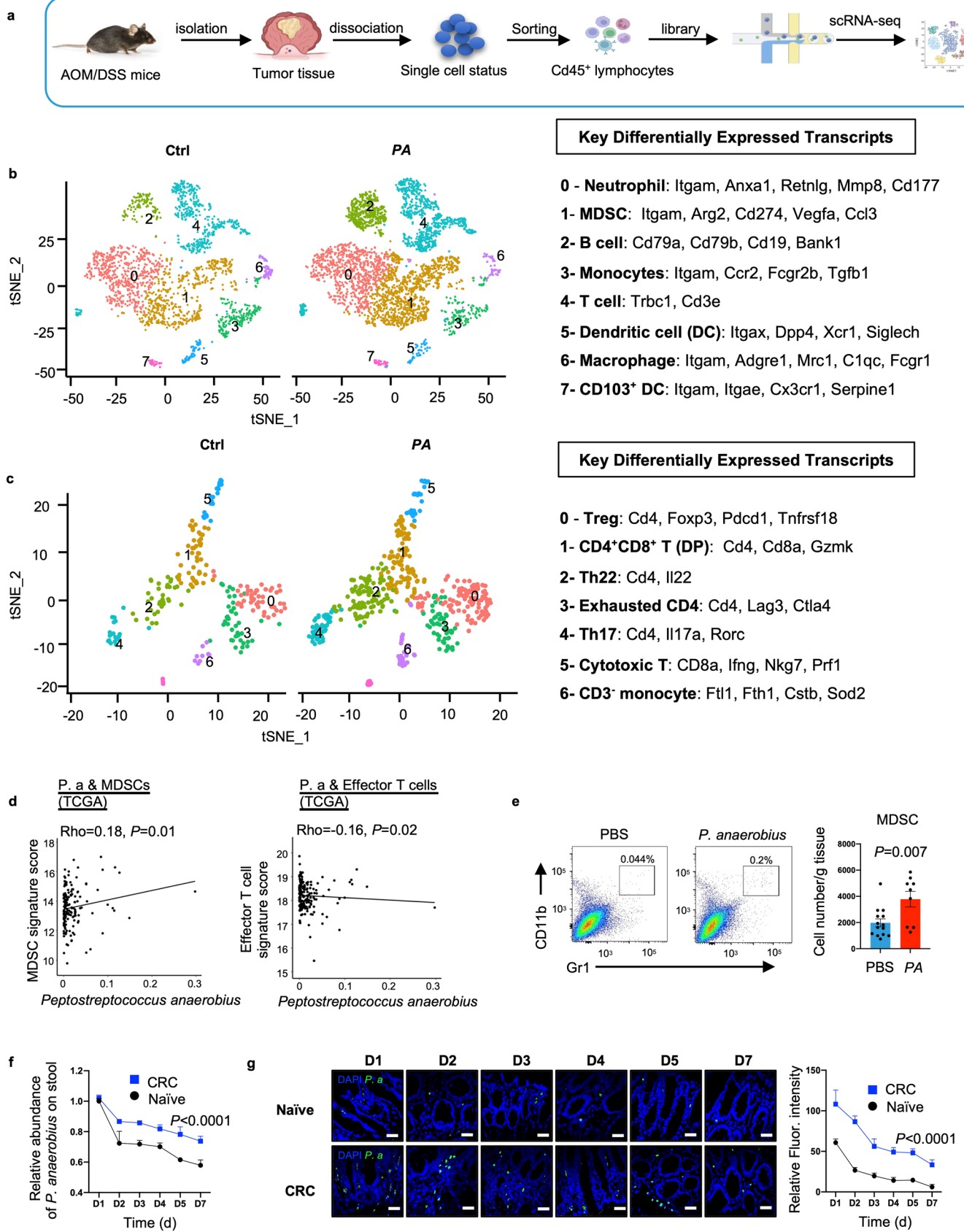

**Key Differentially Expressed Transcripts**

0 - **Neutrophil**: Itgam, Anxa1, Retnlg, Mmp8, Cd177

1- **MDSC**: Itgam, Arg2, Cd274, Vegfa, Ccl3

2- **B cell**: Cd79a, Cd79b, Cd19, Bank1

3- **Monocytes**: Itgam, Ccr2, Fcgr2b, Tgfb1

4- **T cell**: Trbc1, Cd3e

5- **Dendritic cell (DC)**: Itgax, Dpp4, Xcr1, Siglech

6- **Macrophage**: Itgam, Adgre1, Mrc1, C1qc, Fcgr1

7- **CD103$^+$ DC**: Itgam, Itgae, Cx3cr1, Serpine1

**Key Differentially Expressed Transcripts**

0 - **Treg**: Cd4, Foxp3, Pdcd1, Tnfrsf18

1- **CD4$^+$CD8$^+$ T (DP)**: Cd4, Cd8a, Gzmk

2- **Th22**: Cd4, Il22

3- **Exhausted CD4**: Cd4, Lag3, Ctla4

4- **Th17**: Cd4, Il17a, Rorc

5- **Cytotoxic T**: CD8a, Ifng, Nkg7, Prf1

6- **CD3$^-$ monocyte**: Ftl1, Fth1, Cstb, Sod2

**Extended Data Fig. 2 | See next page for caption.**

**Extended Data Fig. 2 | scRNA-seq revealed that *P. anaerobius* promotes an immune suppressive tumour microenvironment in CRC. a**. Schematic diagram of experimental design for single-cell RNA sequencing (scRNA-seq). CD45⁺ infiltrating immune cells from colon tumours of control- and *P. anaerobius-treated* AOM/DSS mice were subjected to scRNA-seq (n = 3 per group). **b**. tSNE plot embedding of CD45⁺ leukocyte in tumours treated with *P. anaerobius* or not (Ctrl, control). Key differentially expressed transcripts that define each cell cluster were listed on left. **c**. tSNE plot of T lymphocytes in tumours treated with *P. anaerobius* or not. Sub-cluster identification within T lymphocytes with key differentially transcripts and associated signatures were listed on left. **d**. Data from TCGA database showed the correlation between the abundance of *P. anaerobius* and MDSCs, effector T cells. **e**. Naive C57BL/6 mice were colonized with PBS and *P. anaerobiu* for 4 weeks, Percentage of MDSCs in the colonic lamina propria (LP) were determined by flow cytometry analysis. 9-14 mice were used in each group, including PBS (n = 14) and *P. anaerobius* (n = 9). **f**. Stool abundance of *P. anaerobius* was determined by q-PCR. 3-5 mice were used in each group, including D1: CRC (n = 5) and Naïve (n = 5). D2-D5: CRC (n = 4) and Naïve (n = 4). D6: CRC (n = 3) and Naïve (n = 3). **g**. Colonic abundance of *P. anaerobius* was determined by Fluorescence In Situ Hybridization (FISH). Scale bar: 25 $\mu$m. Data are presented as mean ± SEM. P values were calculated by one-way ANOVA followed by spearman correlation (**d**), unpaired two-tailed Student's t-test (**e**) and two-way ANOVA followed by Bonferroni's post-hoc test (**f, g**).

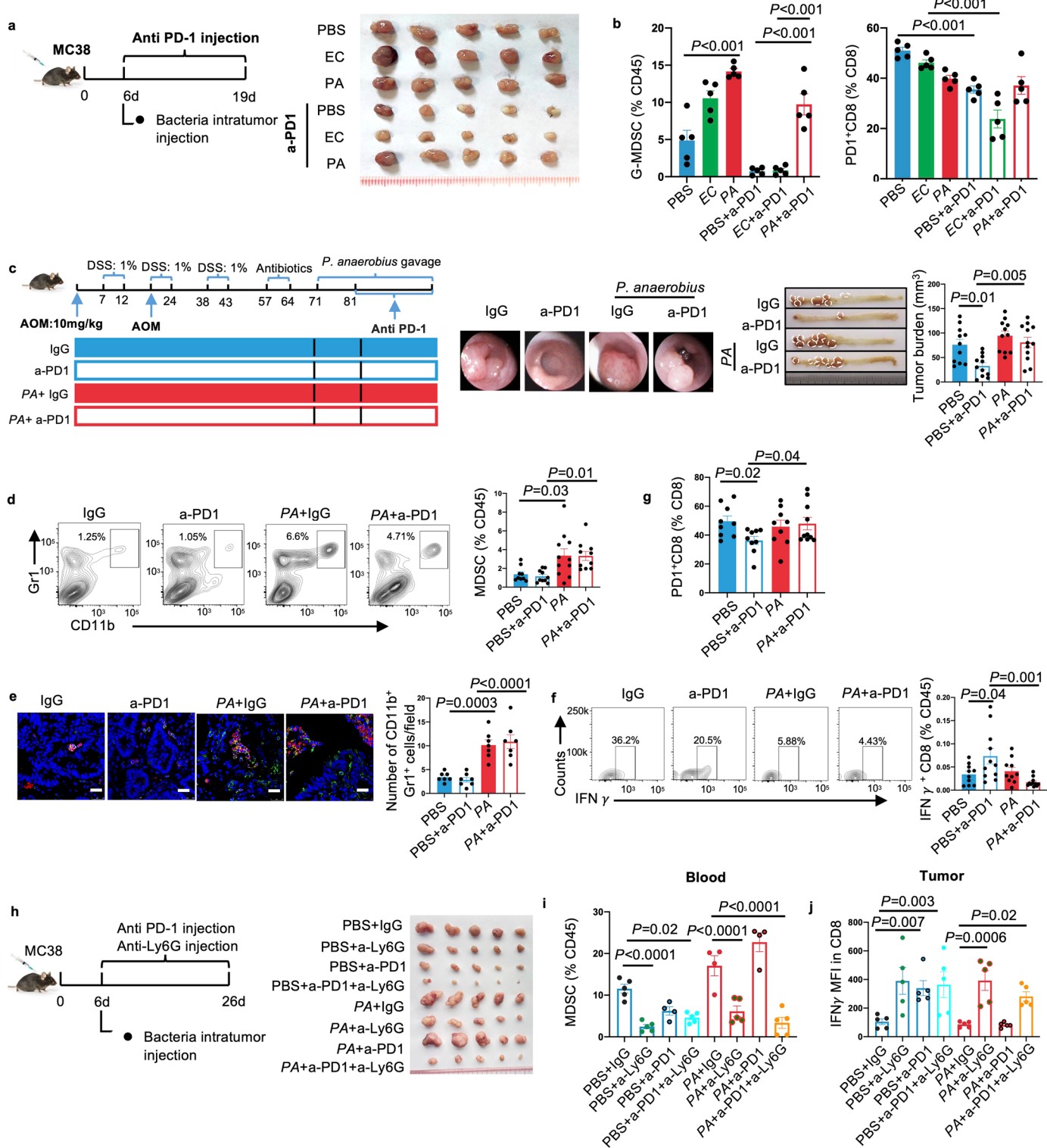

**Extended Data Fig. 3 | See next page for caption.**

**Extended Data Fig. 3 | *P. anaerobius* induced anti-PD1 immunotherapy resistance by modulating MDSCs. a**. Schematic diagram for the MC38 allograft model (Right). Representative images of tumours after harvesting (left). **b**. Percentage of G-MDSCs (left), PD1⁺CD8⁺T cells (right) in MC38 allografts. 5 mice were used in each group. **c**. Schematic diagram of AOM/DSS induced CRC mouse model. Representative images of mouse colonoscopy at day 100 (left), images of colon and tumour numbers (middle), quantification of tumour burden (right) in each group. PBS (n = 11), PBS+ a-PD1 (n = 11), PA (n = 11), PA+ a-PD1 (n = 12). **d**. Representative tumour flow plots and percentage of MDSCs in tumours from AOM/DSS mice model. PBS (n = 9), PBS+ a-PD1 (n = 10), PA (n = 10), PA+ a-PD1 (n = 11). **e**. Immunofluorescence quantification of CD11b⁺Gr-1⁺ cells in tumour from AOM/DSS mice model. Scale bar: 25 μm. **f**. Representative tumour flow plots and percentage of IFNγ⁺CD8⁺T cells in tumours from AOM/DSS mice

model. PBS (n = 10), PBS+ a-PD1 (n = 11), PA (n = 10), PA+ a-PD1 (n = 11). **g**. Percentage of PD1⁺CD8⁺T cells in AOM/DSS mice model. PBS (n = 9), PBS+ a-PD1 (n = 9), PA (n = 9), PA+ a-PD1 (n = 10). **h**. Schematic diagram of MDSC depletion using anti-Ly6G antibody in MC38 allograft model (left), and representative image of tumours after at sacrifice (right). 5×10⁵ MC38 cells were subcutaneously injected into C57BL/6 J mice (n = 5). After 6 days, mice were injected with *P. anaerobius* intratumorally and followed by PD1 mAb treatment. 250ug anti-Ly6G was intraperitoneally injected every 48 hrs to deplete MDSCs in mice. **i**. Percentage of MDSCs in blood from different groups of mice were determined by flow cytometry. **j**. MFI of IFNγ⁺CD8⁺T cells in tumour were determined by flow cytometry. **h-j** 5 mice were used in each group. Data are presented as mean ± SEM. P values were calculated by one-way ANOVA followed by Tukey's post-hoc test(**b-g, i-j**).

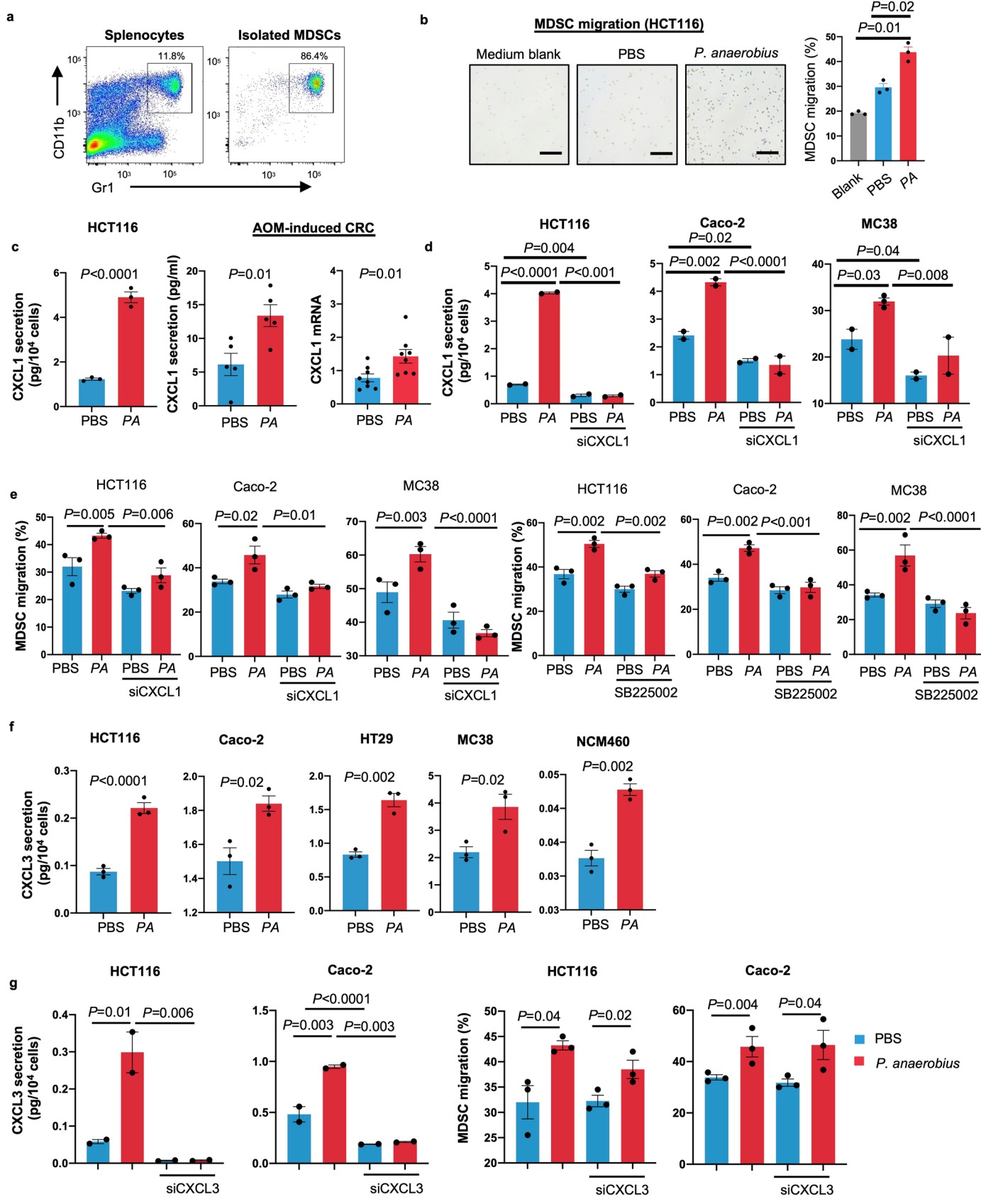

**Extended Data Fig. 4 | See next page for caption.**

**Extended Data Fig. 4 | *P. anaerobius* induced CXCL1 secretion in tumour cells to promote MDSC migration. a**. The CD11b⁺Gr-1⁺ cell content of the isolated fraction from spleen of MC38-bearing C57BL/6 mice. **b**. Representative images of MDSCs migrated to the lower chamber and quantification of MDSC migration towards conditioned medium from HCT116 cells pre-incubated with vehicle or *P. anaerobius*. Scale bar: 100 $\mu$m. **c**. Quantification of CXCL1 secretion from HT29, induced by *P. anaerobius* using ELISA (n = 3 in each group) and Quantification of serum CXCL1 (pg/ml) in AOM-injected mice (n = 5 in each group). mRNA expression of CXCL1 in the colon of AOM-injected mice (n = 8 in each group). **d**. CXCL1 secretion from HCT116, Caco-2 and MC38 cell lines pretreated with *P. anaerobius* or PBS with or without CXCL1 knockdown (n = 2 biologically independent samples). **e**. *P. anaerobius*-promoted MDSC migration in conditioned medium of HCT116 cells was abolished by CXCL1 knockdown (left), or SB225002 (CXCR2 antagonist) treatment (right) (n = 3 biologically independent samples). **f**. Quantification of CXCL3 secretion from HCT116, Caco-2, HT29, MC38, NCM460 induced by *P. anaerobius* using ELISA (n = 3 in each group). **g**. CXCL3 secretion (n = 2 in each group) and MDSC migration (n = 3 in each group) rate induced by HCT116 and Caco-2 pretreated with *P. anaerobius* or PBS with or without CXCL3 knockdown. Data are presented as mean ± SEM. P values were calculated by one-way ANOVA followed by Tukey's post-hoc test (**b, d, e, g**), unpaired two-tailed Student's t-test (**c, f**).

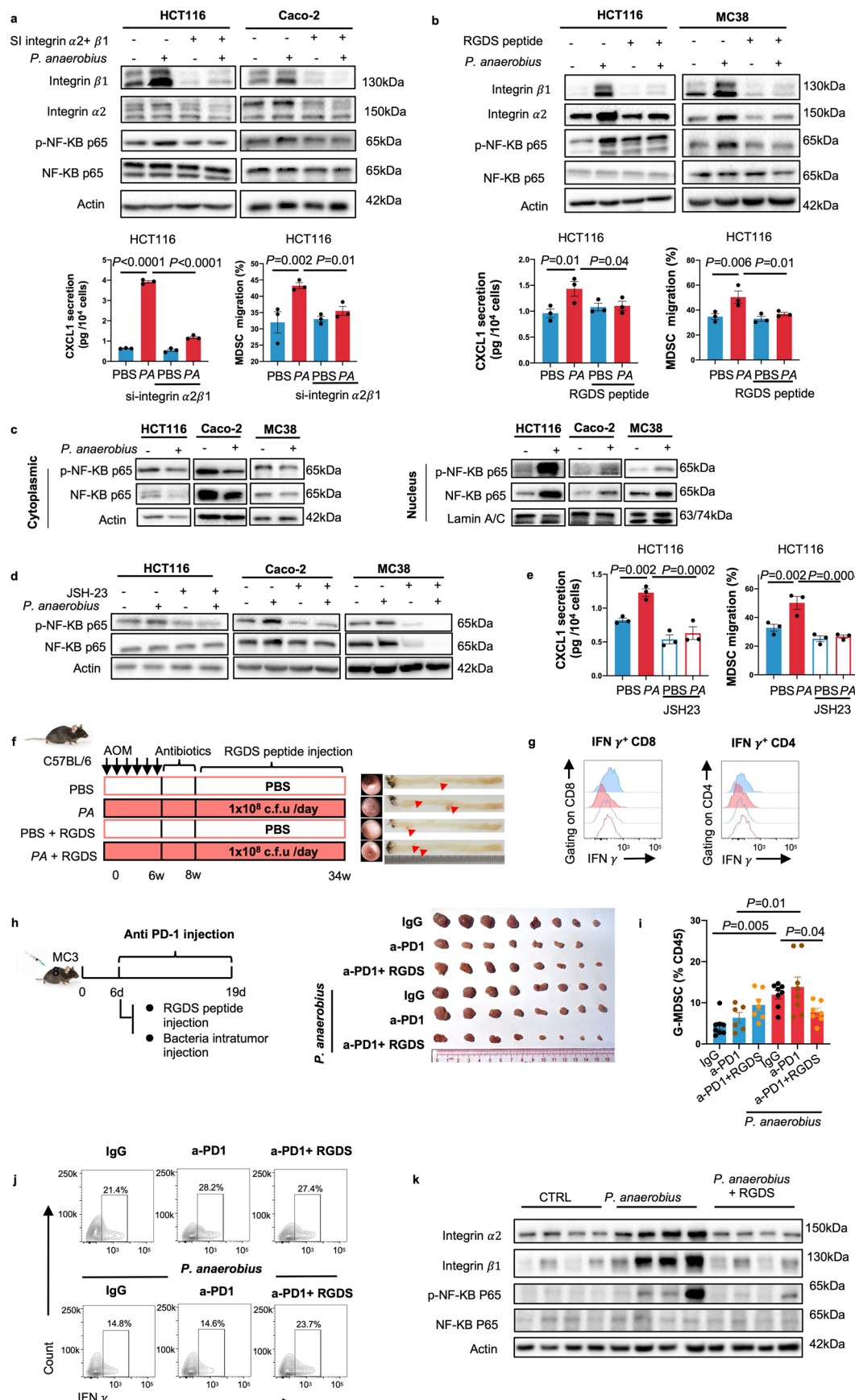

**Extended Data Fig. 5 | See next page for caption.**

**Extended Data Fig. 5 | *P. anaerobius* activates the NF-κB pathway though interactiong with integrin α2/β1 on cancer cells. a**. Validation of integrin α2/β1 knock down efficiency by western blot in HCT116 and Caco-2 cells (up) and Integrin α2β1 knockdown in HCT116 cells abrogated *P. anaerobius*-induced CXCL1 secretion and migration of MDSCs (down). **b**. Antibody blocking of integrin α2/β1 suppressed *P. anaerobius*-induced NF-κB activation in HCT116 and MC38 cells (up) and RGDS peptide treatment in HCT116 cells abrogated *P. anaerobius*-induced CXCL1 secretion and migration of MDSCs (down). **a-b** n = 3 biologically independent samples. **c**. *P. anaerobius* promoted the nuclear translocation of NF-κB p65 and phospho-p65 in CRC cells. **d**. Induction of NF-κB activation by *P. anaerobius* was abolished by JSH-23 in HCT116, Caco-2 and MC38 cells. Three independent experiments were repeated with similar results in **a–d**. **e**. Treatment with JSH-23 abolished *P. anaerobius*-induced CXCL1 secretion (left) and MDSCs migration (right) in conditioned medium from HCT116 cell (n = 3 biologically independent samples). **f**. Schematic diagram of RGDS treatment in mice. RGDS abrogated *P. anaerobius*-induced colorectal tumorigenesis. **g**. Representative flow plots of IFNγ⁺CD8⁺T cells and IFNγ⁺CD4⁺ T cells in colonic LP cells. **h**. Schematic diagram of MC38 allograft model. RGDS peptide reversed *P. anaerobius*-mediated anti-PD1 resistance in MC38 allografts. Representative images of tumours after sacrificing. **i**. Percentage of G-MDSCs in tumours were determined by flow cytometry analysis. 6-8 mice were used in each group including IgG (n = 8), a-PD1 (n = 6), a-PD1 + RGDS (n = 7), PA+ IgG (n = 8), PA+ a-PD1 (n = 8), PA+ a-PD1 + RGDS (n = 7). **j**. Representative tumour flow plots of IFNγ⁺CD8⁺T cells in tumours. **k**. Integrin α2/β1 blockade abolished the effect of *P. anaerobius* on NF-κB pathway in MC38 allograft tumours (n = 4 biologically independent samples). Data are presented as mean ± SEM. P values were calculated by one-way ANOVA followed by Tukey's post-hoc test (**e, i**).

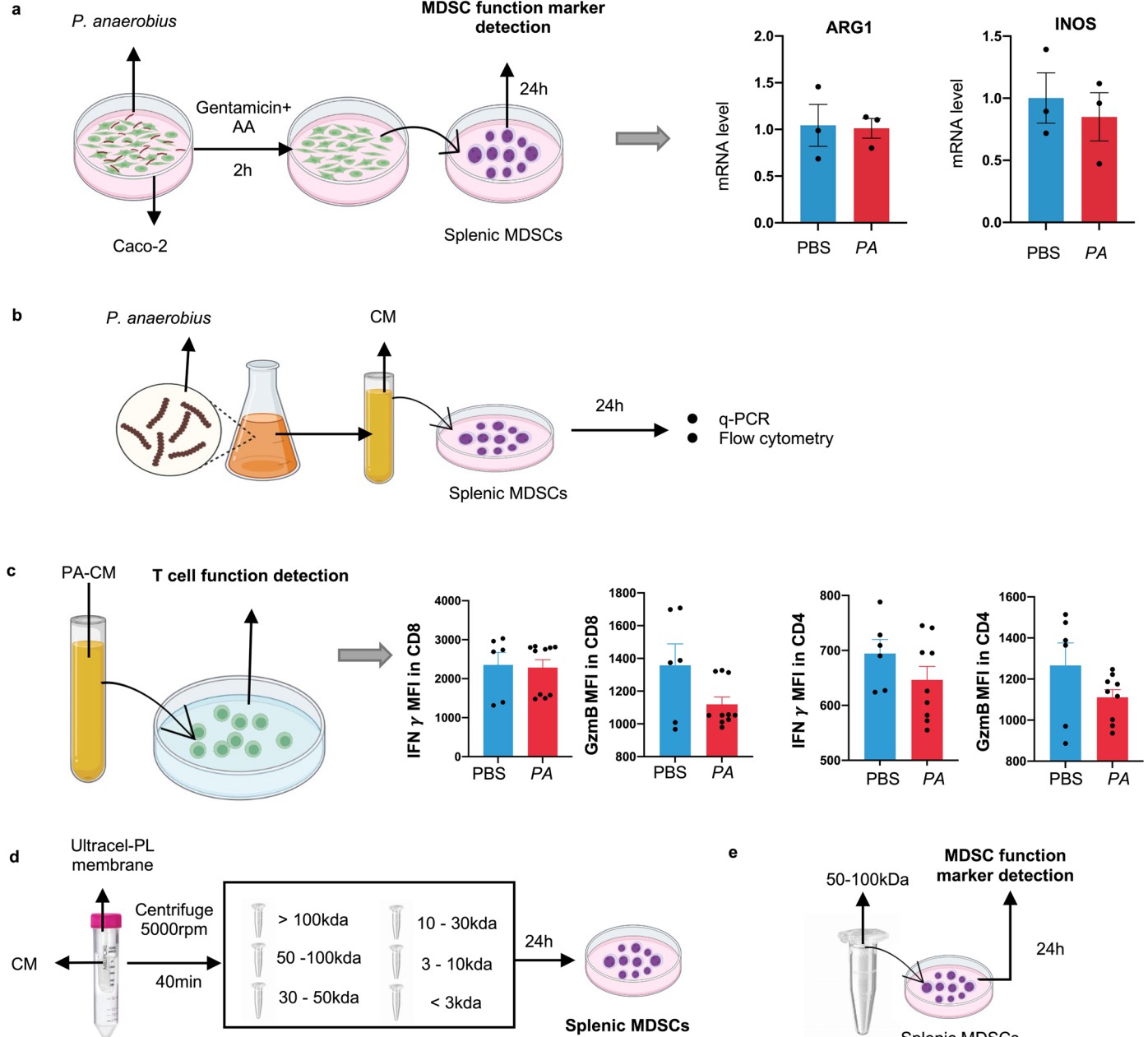

**Extended Data Fig. 6 | *P. anaerobius* produced metabolites to modulate MDSCs, further influencing T cell function. a**. Schematic of experiments design. MDSCs were treated with 10% culture medium from Caco-2 pre-incubated with *P. anaerobius* for 24 h. mRNA expressions level of Arg1 and iNOS were detected using RT-qPCR (n = 3 biologically independent samples). **b**. Schematic diagram of experimental design. MDSCs treated with 10% *P. anaerobius*-conditioned medium (Pa-CM) or *E. coli*-conditioned medium (Ec-CM) were harvested after 24 h treatment. **c**. Primary T cells were treated with 10% Pa. CM for 24 h. The percentage of functional CD4+ and CD8+ T cell was detected using FACS (n = 6-10 biologically independent samples). **d**. Different molecular fractions from Pa-CM were used to treat MDSCs. **e**. MDSCs were treated with 50-100 kDa fraction from Pa-CM or control broth. Data are presented as mean ± SEM. P values were calculated unpaired two-tailed Student's t-test.

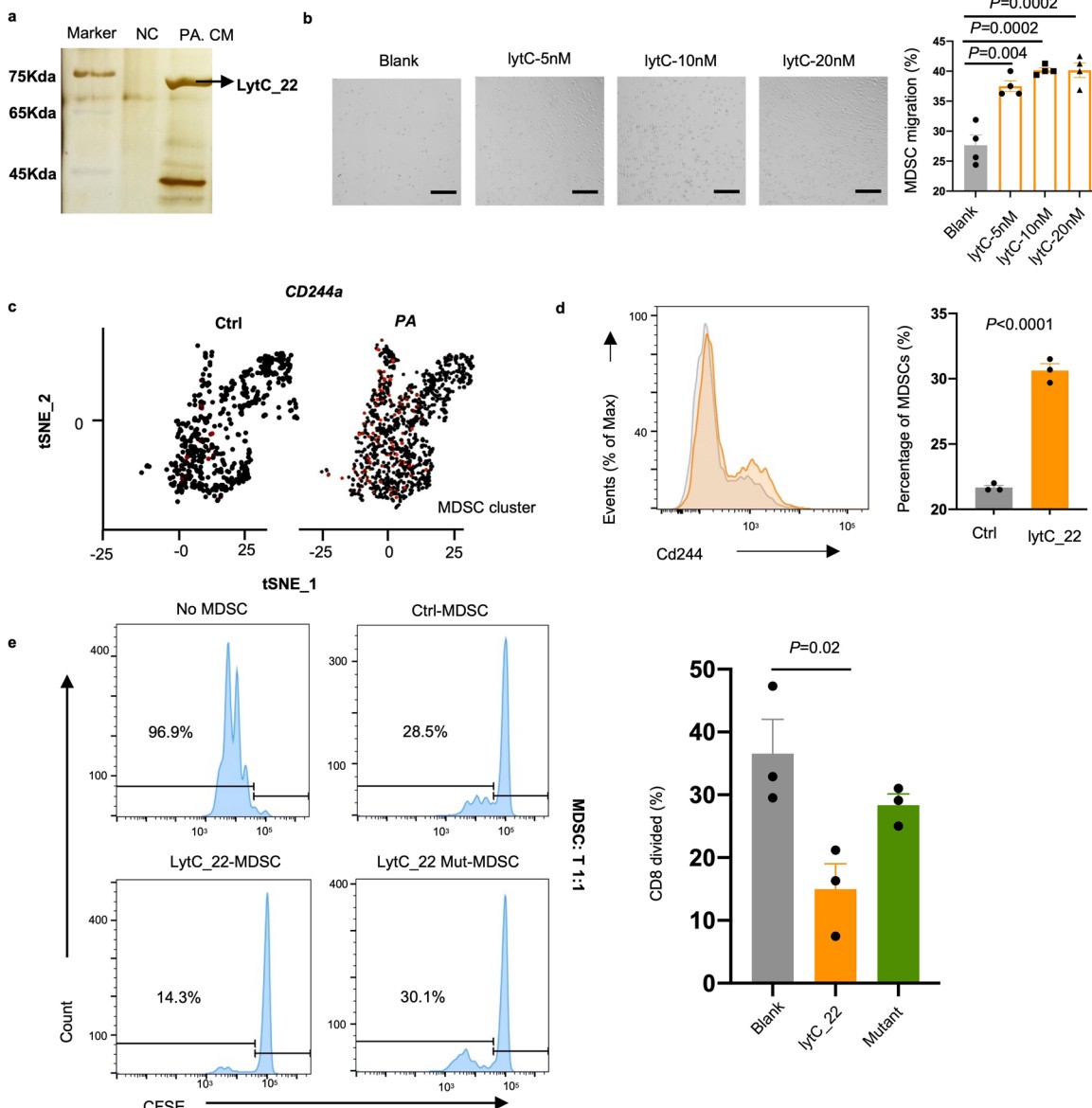

**Extended Data Fig. 7 | *P. anaerobius* activates MDSCs activation via lytC_22/Slamf4 interaction. a**. The 50–100 kDa fraction from Pa-CM was separated using sodium dodecyl sulfate-polyacrylamide gel electrophoresis. **b**. Representative images of MDSCs migrated to the lower chamber and quantification of MDSC migration towards different concentration of lytC_22 (n = 4 biologically independent samples). Scale bar: 100 μm. **c.** tSNE plots showing expression levels of *Cd244a* in MDSCs clusters from tumours treated with *P. anaerobius* or not

(Ctrl, control). **d**. Percentage of CD244+ MDSCs in MDSCs after LytC_22 treatment was determined by FACS analysis. **e**. MDSCs were pretreated with lytC_22 and lytC_22 mutant protein. CD8+ T cell proliferation after co-culture with MDSCs. **c-d** n = 3 biologically independent sample. Data are presented as mean ± SEM. P values were calculated by one-way ANOVA followed by Tukey's post-hoc test (**b, e**) and unpaired two-tailed Student's t-test (**d**).

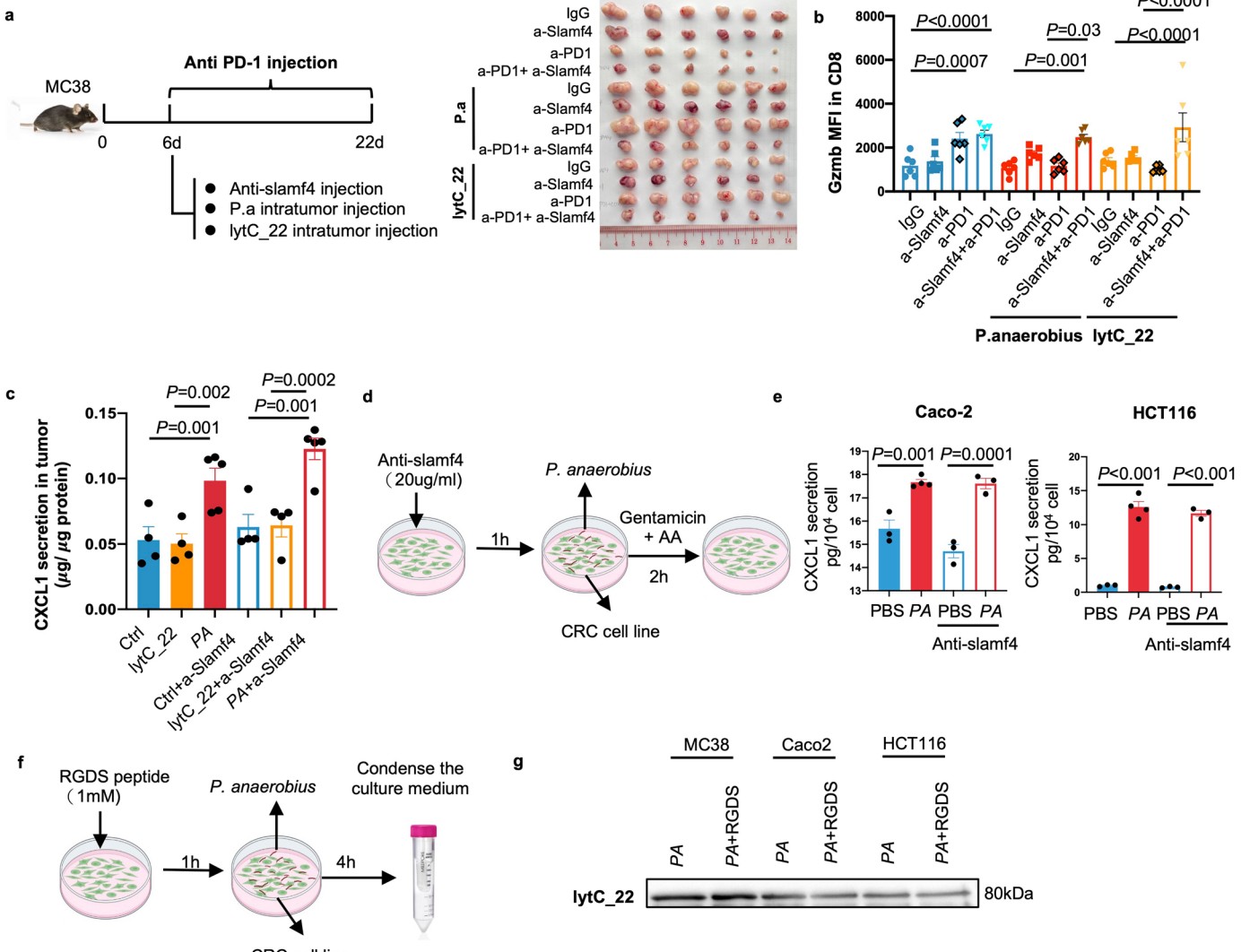

**Extended Data Fig. 8 | The integrin α2β1–CXCL1 and lytC_22–Slamf4 axis function independently. (a-c)** 5×10⁵ MC38 cells were subcutaneously injected into C57BL/6J mice (n = 6). After 6 days, mice were injected with *P. anaerobius* or lytC_22 intratumorally and followed by PD1 mAb treatment. 100ug anti-PD1 intraperitoneally and 50ug anti-Slamf4 antibody intravenously injected every 3 days. **a.** Schematic diagram of experimental design (left) and representative images of tumours (right). **b.** Gzmb⁺CD8⁺T cells in tumours from different groups. 6 mice were used in each group. **c.** CXCL1 level in tumour tissue were detected using ELISA. 4-5 mice were used in each group including n = 4 mice in Ctrl, lytC_22, Ctrl+a-Slamf4, lytC_22+a-Slamf4 and n = 5 mice in *PA*, *PA* + a-Slamf4 group. **d.** Schematic diagram of experimental design. CRC cell line (Caco-2, HCT116) pretreated with 20ug/ml anti-Slamf4 antibody, followed by

*P. anaerobius* infection. Then culture medium was collected. **e.** Quantification of secreted CXCL1 from human CRC cells Caco-2, HCT116 treated with vehicle or *P. anaerobius* after blocking Slamf4 receptor. PBS (n = 3), *PA* (n = 4), PBS+a-Slamf4 (n = 3), *PA* + a-Slamf4 (n = 3). **f.** Schematic diagram of experimental design. Human CRC cell line (Caco-2, HCT116) and mouse CRC cell line (MC38) pretreated with 1 mM RGDS peptide, followed by *P. anaerobius* infection. Then culture medium was collected and condensed using 50 kDa protein concentration column. **g.** Quantification of secreted lytC_22 from *P. anaerobius* was detected using western blot. Data are presented as mean ± SEM. P values were calculated by one-way ANOVA followed by Fisher's LSD test (**b**) and one-way ANOVA followed by Tukey's post-hoc test (**c, e**).

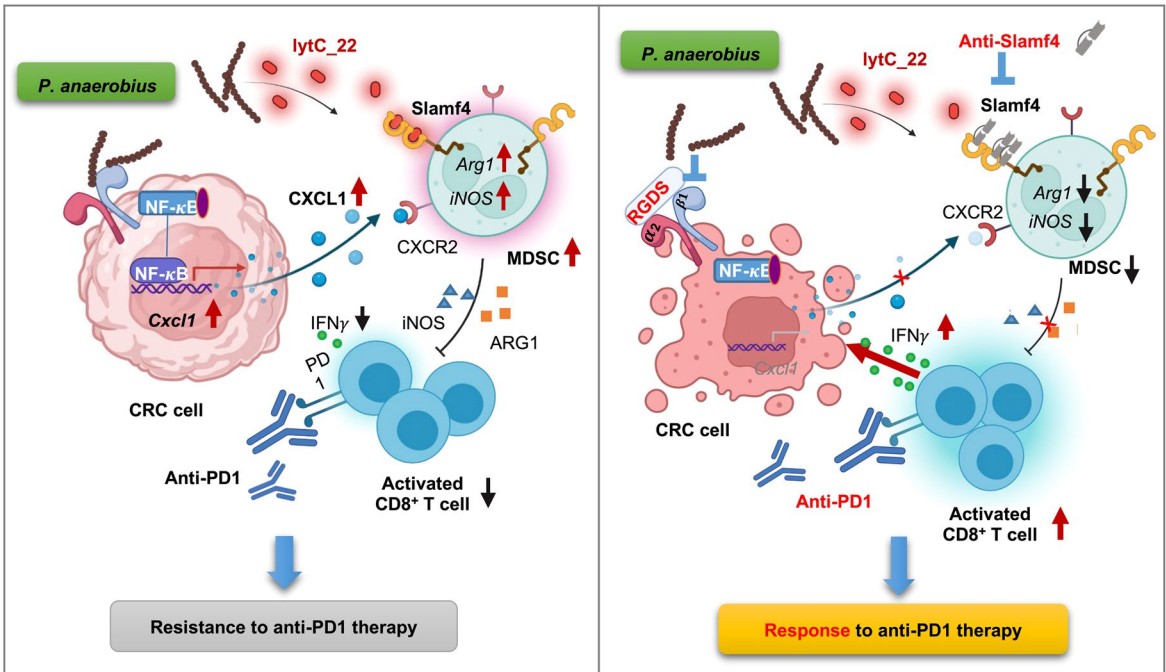

**Extended Data Fig. 9 | Schematic model of *P. anaerobius*-driven anti-PD1 resistance in CRC and potential therapeutic approaches.** *P. anaerobius* engaged with CRC cells via integrin α2/β1 receptor to promote nuclear translocation of NF-κB p65, which in turn stimulates release of CXCL1 to recruit CXCR2+ MDSCs to tumours. In addition, lytC_22 protein secreted from *P. anaerobius* directly activates Slamf4 receptor on MDSCs, leading to elevated arginase 1 (Arg1) and iNOS expression. *P. anaerobius* infection thus increased MDSCs accumulation and activation, leading to suppression of CD8+ T cells and non-response to anti-PD1 therapy. Blockade of integrin α2/β1 or Slamf4 could interrupt this process to impair MDSCs in tumour microenvironment and reactivate IFNγ expressing CD8+ T cells to overcome *P. anaerobius*-driven anti-PD1 resistance.

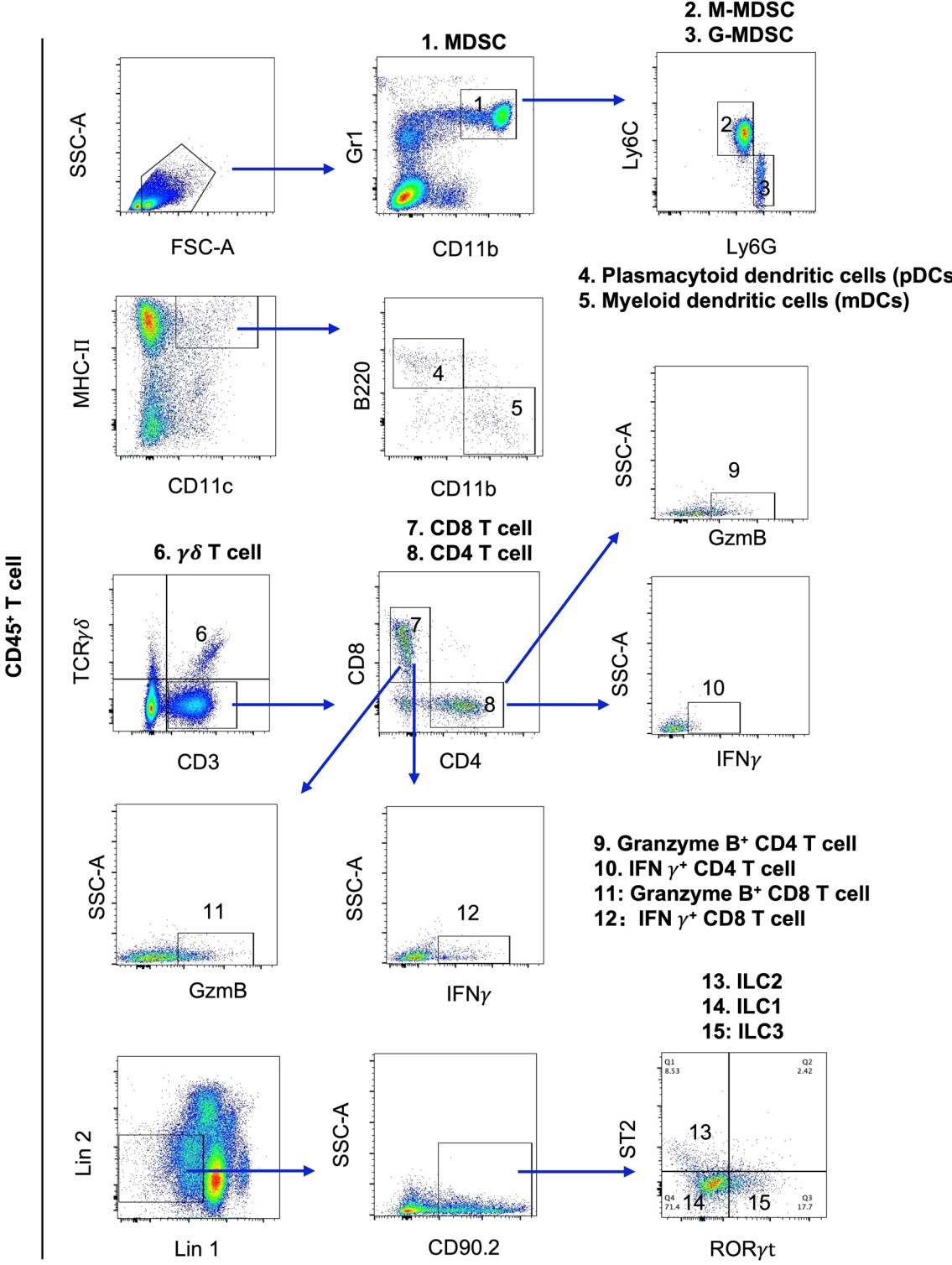

**Extended Data Fig. 10 | Gating strategy of lymphocytes isolated from mice tumour or colonic lamina propria.** Gating strategy of lymphocytes from mice, including MDSCs (CD11b⁺Gr1ʰⁱ), M-MDSC (CD11b⁺Gr1ʰⁱLy6CʰⁱLy6G⁻), G-MDSC (CD11b⁺Gr1ʰⁱLy6CⁱⁿᵗLy6G⁺), pDCs (CD11C⁺ MHC-Π⁺CD11b⁻B220⁺), mDCs (CD11C⁺MHC-Π⁺CD11b⁺B220⁻), γδ T cell (CD3⁺TCR γδ⁺), CD4⁺ T (CD3⁺CD4⁺), CD8⁺ T (CD3⁺CD8⁺), ILC1 (Lin1⁻Lin2⁻CD90.2⁺ST2⁻RORγt⁻), ILC2 (Lin1⁻Lin2⁻CD90.2⁺ST2⁺RORγt⁻), ILC3 (Lin1⁻Lin2⁻CD90.2⁺ST2⁻RORγt⁺). Lin1: CD11c, CD5; Lin2: CD3, B220, F4/80, CD11b.

# Reporting Summary

## Statistics

For all statistical analyses, confirm that the following items are present in the figure legend, table legend, main text, or Methods section.

| n/a | Confirmed | |
|---|---|---|
| ☐ | ☒ | The exact sample size (*n*) for each experimental group/condition, given as a discrete number and unit of measurement |
| ☐ | ☒ | A statement on whether measurements were taken from distinct samples or whether the same sample was measured repeatedly |
| ☐ | ☒ | The statistical test(s) used AND whether they are one- or two-sided *Only common tests should be described solely by name; describe more complex techniques in the Methods section.* |
| ☒ | ☐ | A description of all covariates tested |
| ☒ | ☐ | A description of any assumptions or corrections, such as tests of normality and adjustment for multiple comparisons |
| ☐ | ☒ | A full description of the statistical parameters including central tendency (e.g. means) or other basic estimates (e.g. regression coefficient) AND variation (e.g. standard deviation) or associated estimates of uncertainty (e.g. confidence intervals) |
| ☐ | ☒ | For null hypothesis testing, the test statistic (e.g. *F*, *t*, *r*) with confidence intervals, effect sizes, degrees of freedom and *P* value noted *Give P values as exact values whenever suitable.* |
| ☒ | ☐ | For Bayesian analysis, information on the choice of priors and Markov chain Monte Carlo settings |
| ☒ | ☐ | For hierarchical and complex designs, identification of the appropriate level for tests and full reporting of outcomes |
| ☒ | ☐ | Estimates of effect sizes (e.g. Cohen's *d*, Pearson's *r*), indicating how they were calculated |

*Our web collection on statistics for biologists contains articles on many of the points above.*

## Software and code

Policy information about availability of computer code

| Data collection | Image Lab software v6.1.0 (Bio-rad), QuantStudio Real Time PCR software v1.7.2 (Applied Biosystems), FACS Diva 6.1.3 (BD Biosciences) |
|---|---|
| Data analysis | FlowJo (v9), Image J 1.51, Prism v9, HDOCK software, Pymol 2.1 software |

For manuscripts utilizing custom algorithms or software that are central to the research but not yet described in published literature, software must be made available to editors and reviewers. We strongly encourage code deposition in a community repository (e.g. GitHub). See the Nature Portfolio guidelines for submitting code & software for further information.

## Data

Policy information about availability of data

All manuscripts must include a data availability statement. This statement should provide the following information, where applicable:
- Accession codes, unique identifiers, or web links for publicly available datasets
- A description of any restrictions on data availability
- For clinical datasets or third party data, please ensure that the statement adheres to our policy

The datasets generated during and/or analysed during the current study are available from the corresponding authors upon request. All related raw data has been deposited to the public datasets with accession code.

## Human research participants

Policy information about studies involving human research participants and Sex and Gender in Research.

| | |
|---|---|
| Reporting on sex and gender | N/A |
| Population characteristics | N/A |
| Recruitment | N/A |
| Ethics oversight | N/A |

Note that full information on the approval of the study protocol must also be provided in the manuscript.

# Field-specific reporting

Please select the one below that is the best fit for your research. If you are not sure, read the appropriate sections before making your selection.

☒ Life sciences       ☐ Behavioural & social sciences       ☐ Ecological, evolutionary & environmental sciences

For a reference copy of the document with all sections, see nature.com/documents/nr-reporting-summary-flat.pdf

# Life sciences study design

All studies must disclose on these points even when the disclosure is negative.

| | |
|---|---|
| Sample size | No statistical methods were used to predetermine sample size, but our sample sizes are similar to those in previous reports |
| Data exclusions | No exclusion of data was performed |
| Replication | All experiments were repeated 3 three times as three independent experiments unless otherwise stated. |
| Randomization | All samples and animals were allocated in random. |
| Blinding | blinding was not possible in experiments since the same investigator performed the experiment and data analysis. |

# Reporting for specific materials, systems and methods

We require information from authors about some types of materials, experimental systems and methods used in many studies. Here, indicate whether each material, system or method listed is relevant to your study. If you are not sure if a list item applies to your research, read the appropriate section before selecting a response.

## Materials & experimental systems

| n/a | Involved in the study |
|---|---|
| ☐ | ☒ Antibodies |
| ☐ | ☒ Eukaryotic cell lines |
| ☒ | ☐ Palaeontology and archaeology |
| ☐ | ☒ Animals and other organisms |
| ☒ | ☐ Clinical data |
| ☒ | ☐ Dual use research of concern |

## Methods

| n/a | Involved in the study |
|---|---|
| ☒ | ☐ ChIP-seq |
| ☐ | ☒ Flow cytometry |
| ☒ | ☐ MRI-based neuroimaging |

## Antibodies

| | |
|---|---|
| Antibodies used | Intagrin α2 (Abcam, ab133557), Intagrin β1 (Abcam, ab179471), NF-KB P65 (Cell Signaling Technology, 8242S), Phospho-NF-κB p65 (Ser536) (Cell Signaling Technology, 3033S), Slamf4 (Cell Signaling Technology, 54560S), anti-GST (Cell Signaling Technology, 2624S), |
| Validation | All antibodies are obtained from commercial sources, and vendors have shown validation on their websites. |

# Eukaryotic cell lines

Policy information about cell lines and Sex and Gender in Research

| | |
|---|---|
| Cell line source(s) | Colon cancer cell lines HT-29 , Caco-2, HCT116, MC38 were obtained from ATCC. HT29 was isolated from a female patient. Caco-2 was isolated from a male patient. HCT116 was isolated from a male patient. MC38 was isolated from C57BL/6 mice. Colon normal immortalized epithelial cell line NCM460 was obtained from INCELL. NCM460  was isolated from the normal colon of a Hispanic male. |
| Authentication | The cell lines were bought from ATCC and INCELL with authentication (STR profiling). |
| Mycoplasma contamination | None of the cell lines were mycoplasma contamination |
| Commonly misidentified lines (See ICLAC register) | No commonly misidentified cell lines were used in this study. |

# Animals and other research organisms

Policy information about studies involving animals; ARRIVE guidelines recommended for reporting animal research, and Sex and Gender in Research

| | |
|---|---|
| Laboratory animals | Description of research mice used for experiments can be found in the relevant figure legends and Methods. Mice were kept in specific pathogen-free facilities, following a 12-hour light/12-hour dark cycle, and had ad libitum access to food and water. Food and water were provided ad libitum. |
| Wild animals | This study did not involve wide animals. |
| Reporting on sex | Findings only applied to male mice. |
| Field-collected samples | This study did not involve field-collected samples. |
| Ethics oversight | All animal experiments were approved by the Animal Experimentation Ethics Committee of the Chinese University of Hong Kong |

Note that full information on the approval of the study protocol must also be provided in the manuscript.

# Flow Cytometry

## Plots

Confirm that:

☒ The axis labels state the marker and fluorochrome used (e.g. CD4-FITC).

☒ The axis scales are clearly visible. Include numbers along axes only for bottom left plot of group (a 'group' is an analysis of identical markers).

☒ All plots are contour plots with outliers or pseudocolor plots.

☒ A numerical value for number of cells or percentage (with statistics) is provided.

## Methodology

| | |
|---|---|
| Sample preparation | Sample preparation was listed in Methods. colons were minced into pieces and incubated in a digestion buffer consisting of PBS, 1% BSA, 1 mg/mL collagenase type IV and 0.5mg/ml DNase I for 30 min at 37?. The cell suspensions were filtered through a 70μm strainer and then centrifuged at 2000 rpm, 10min. For surface marker staining, LP cells were stained with fluorescent-conjugated antibodies in cell staining buffer for 20 min. For intracellular staining, cells were stimulated for 4 h in RPMI complete medium with 30 ng/ml phorbol 12-myristate 13-acetate (Abcam) and 1μg/ml ml ionomycin (STEMCELL Technologies) in the presence of 2.5μg/ml monomycin (BioLegend). Then, cells were stained with cell surface markers. Samples for flowcytometry analysis were fixed with the cell fixation buffer (BioLegend), permeabilized with FOXP3 Perm buffer (eBioscience) according to the manufacturer's recommendations and stained with intracellular antibody. |
| Instrument | BD FACSAria Fusion |
| Software | BD FACSAria Fusion, Flowjo v9 |
| Cell population abundance | Tumor infiltrating CD45 positive cells from each mice colon (50000-100000 cells) were used to further identify different kinds of immune cells by its surface markers. |

Gating strategy

Gating strategy of lymphocytes from mice, including MDSCs (CD11b+Gr1hi), M-MDSC (CD11b+Gr1hiLy6ChiLy6G-), G-MDSC (CD11b+Gr1hiLy6CintLy6G+), pDCs (CD11C+ MHC-Π+CD11b-B220+), mDCs (CD11C+MHC-Π+CD11b+B220-), gd T cell (CD3 +TCR gd+), CD4+ T (CD3+CD4+), CD8+ T (CD3+CD8+), ILC1 (Lin1-Lin2-CD90.2+ST2-RORgt-), ILC2 (Lin1-Lin2-CD90.2+ST2 +RORrt-), ILC3 (Lin1-Lin2-CD90.2+ST2-RORrt+). Lin1: CD11c, CD5; Lin2: CD3, B220, F4/80, CD11b.

☒ Tick this box to confirm that a figure exemplifying the gating strategy is provided in the Supplementary Information.

