## [Peer Review File · Nature Microbiology]

Peer Review Information

Journal: Nature Microbiology

Manuscript Title: Peptostreptococcus anaerobius mediates anti-PD1 therapy resistance and exacerbates colorectal cancer via myeloid-derived suppressor cells in mice

Corresponding author name(s): Professor Jun Yu

Reviewer Comments & Decisions:

Decision Letter, initial version:

Message: 3rd March 2023

Dear Professor Yu,

Thank you for your patience while your manuscript "Peptostreptococcus anaerobius provokes anti-PD-1 resistance in colorectal cancer by promoting MDSCs recruitment and activation" was under peer-review at Nature Microbiology. It has now been seen by 3 referees, whose expertise and comments you will find at the of this email. You will see from their comments below that while they find your work of interest, some important points are raised. We are very interested in the possibility of publishing your study in Nature Microbiology, but would like to consider your response to these concerns in the form of a revised manuscript before we make a final decision on publication.

In particular, you will see that several referees ask for further validation of the interaction between LytC_22 and Slamf-4, its effects upon MDSC activation and recruitment and to determine whether these activities are independent. Referee #3 also questioned whether the resistance to anti-PD-1 therapy was simply due to *P. anaerobius* driving more aggressive tumorigenesis rather than a direct immunosuppressive effect of the bacterium and asked for experiments using alternative chemotherapies as a control. Another important concern raised by referee #2 was the relevance of your findings to human CRC, whether *P. anaerobius* can be detected in human tumour samples and if this is associated with decreased anti-PD-1 treatment efficacy. There were also requests for further controls and flow cytometric analyses to validate whether MDSCs are recruited to tumours and what happens to other cell populations. We feel that these are critical points which would need to be addressed for us to further consider a revised manuscript. The remaining issues are outlined in the referees' reports are clear and should be straightforward to address. Should further experimental data allow you to address these criticisms, we would be happy to look at a revised manuscript.

We are committed to providing a fair and constructive peer-review process. Please do not hesitate to contact us if there are specific requests from the reviewers that you believe are technically impossible or unlikely to yield a meaningful outcome. If you would like to discuss your plan for revision and submit a point-by-point revision plan as to how you would respond to the referees' comments, I would be happy to read and discuss this with you.

2We strongly support public availability of data. Please place the data used in your paper into a public data repository, if one exists, or alternatively, present the data as Source Data or Supplementary Information. If data can only be shared on request, please explain why in your Data Availability Statement, and also in the correspondence with your editor. For some data types, deposition in a public repository is mandatory - more information on our data deposition policies and available repositories can be found at <https://www.nature.com/nature-research/editorial-policies/reporting-standards#availability-of-data>.

Please include a data availability statement as a separate section after Methods but before references, under the heading "Data Availability". This section should inform readers about the availability of the data used to support the conclusions of your study. This information includes accession codes to public repositories (data banks for protein, DNA or RNA sequences, microarray, proteomics data etc...), references to source data published alongside the paper, unique identifiers such as URLs to data repository entries, or data set DOIs, and any other statement about data availability. At a minimum, you should include the following statement: "The data that support the findings of this study are available from the corresponding author upon request", mentioning any restrictions on availability. If DOIs are provided, we also strongly encourage including these in the Reference list (authors, title, publisher (repository name), identifier, year). For more guidance on how to write this section please see: <http://www.nature.com/authors/policies/data/data-availability-statements-data-citations.pdf>

- * If you have not done so already we suggest that you begin to revise your manuscript so that it conforms to our Article format instructions at <http://www.nature.com/nmicrobiol/info/final-submission>. Refer also to any guidelines provided in this letter.

When submitting the revised version of your manuscript, please pay close attention to our [href="https://www.nature.com/nature-portfolio/editorial-policies/image-integrity">Digital Image Integrity Guidelines](https://www.nature.com/nature-portfolio/editorial-policies/image-integrity). and to the following points below:

- that unprocessed scans are clearly labelled and match the gels and western blots presented in figures.

2- that control panels for gels and western blots are appropriately described as loading on sample processing controls
- all images in the paper are checked for duplication of panels and for splicing of gel lanes.

Note: This url links to your confidential homepage and associated information about manuscripts you may have submitted or be reviewing for us. If you wish to forward this e-mail to co-authors, please delete this link to your homepage first.

Nature Microbiology is committed to improving transparency in authorship. As part of our efforts in this direction, we are now requesting that all authors identified as 'corresponding author' on published papers create and link their Open Researcher and Contributor Identifier (ORCID) with their account on the Manuscript Tracking System (MTS), prior to acceptance. This applies to primary research papers only. ORCID helps the scientific community achieve unambiguous attribution of all scholarly contributions. You can create and link your ORCID from the home page of the MTS by clicking on 'Modify my Springer Nature account'. For more information please visit please visit www.springernature.com/orcid.

If you wish to submit a suitably revised manuscript we would hope to receive it within 6 months. If you cannot send it within this time, please let us know. We will be happy to consider your revision, even if a similar study has been accepted for publication at Nature Microbiology or published elsewhere (up to a maximum of 6 months).

Yours sincerely,

Reviewer Expertise:

Referee #1: cancer immunology, microbiome
Referee #2: cancer immunology, checkpoint inhibition therapy
Referee #3: microbiota, myeloid immune interactions, cancer

Reviewer Comments:

Reviewer #1 (Remarks to the Author):

In the manuscript "Peptostreptococcus anaerobius provokes anti-PD-1 resistance in colorectal cancer by promoting MDSCs recruitment and activation", Liu et al. identify the mechanism by which *P. anaerobius* interacts with both colorectal cancer cells and MDSCs to suppress anti-tumor immunity. The authors nicely identify this is facilitated by several host factors including integrin alpha2/beta1 and Slamf4 along with *P. anaerobius*-derived LytC_22 protein. While the experiments are very thorough, there are some questions regarding the relative contribution of these host proteins as well as the impact of *P. anaerobius* in naïve mice.

In figure 1, the authors establish that *P. anaerobius* suppresses the presence of tissue IFN γ -expressing CD4 and CD8 cells. Are the total numbers of CD4 and CD8 cells also reduced? This will help understand if these populations fail to be recruited or if they are recruited, but become suppressed at the tissue site.

Similarly, to above do these cells express any exhaustion markers, particularly PD1 during *P. anaerobius* gavage (during AOM/DSS and/or MC38 injections)? The authors use PD1 blockade, but did not report the levels of these markers on the proposed target immune population.

There is an overall question about the impact of *P. anaerobius* in the gut prior to tumorigenesis. Do naïve mice given *P. anaerobius* also display increased MDSC recruitment or does this require a transformed epithelium in the intestine to facilitate this response? Furthermore, are AOM/DSS-treated or MC-38-injected mice more prone to harboring *P. anaerobius* compared to naïve mice, potentially due to a weakened immune state? Measuring *P. anaerobius* CFUs post oral gavage in naïve vs. tumor bearing mice will help address the physiological relevance of this interaction.

In Fig. 3, the authors show that immune suppression by *P. anaerobius* is mediated by interactions with tumor cell integrin alpha2/beta1 and subsequent CXCL1 release in CRC cell lines. They also go on to show that *P. anaerobius* can directly activate MDSCs via production of LytC_22 and Slamf4. Does LytC_22/Slamf4 blockade impact CXCL1 release and/or conversely, does integrin alpha2/beta1-CXCL1 blockade impact LytC_22 production? This will help to understand if these two events function independently or in concert with each other to suppress immune surveillance.

Does *P. anaerobius* lacking LytC_22 fail to induce MDSC recruitment and enhanced tumorigenesis?

Reviewer #2 (Remarks to the Author):

The manuscript by Liu and colleagues reports on the mechanism by which a specific bacteria species, *Peptostreptococcus anaerobius*, inhibits anti-tumor immunity and anti-PD-1 efficacy in a set of colorectal cancer mouse models. In the APC/min mouse model, oral gavage of *P. anaerobius* increased the number of polyps, which was associated with

4

increased myeloid cells phenotypically resembling MDSCs and also decreased T cells. In the transplantable MC38 model, they switch to intratumoral administration of *P. anaerobius*, and found decreased efficacy of systemic anti-PD-1. In the AOM/DSS colitis model, they found that oral gavage of *P. anaerobius* increased tumor formation, decreased anti-PD-1 efficacy, and increased MDSC accumulation. Mechanistically, they provide evidence that *P. anaerobius* induces CXCL1 from tumor cells, which recruits MDSCs, and also engages MDSCs via secreted LytC_22 through interaction with Slamf4. Taken together, these are potentially important results that also point towards potential therapeutic opportunities for reversal of this immune regulatory process in vivo. Nonetheless, there are several details that deserve further elaboration and clarification.

Specific comments:

1. The change from oral gavage to intratumoral bacterial administration was not spelled out clearly. I assume that oral gavage was not immune modulatory in the subcutaneous MC38 model. Since it was indeed immune modulatory in the spontaneous colonic tumor models, these results suggest that this phenomenon is mediated by tumor-associated bacteria rather than a systemic immune effect driven by altered composition of the gut microbiota. These points need to be developed and explained further throughout the manuscript: the title, abstract, and discussion would need to describe this as tumor-associated bacteria, and the background section of the paper should also describe prior literature on the effect of intratumoral bacteria. In the Results section, whenever the bacteria are delivered intratumorally this needs to be mentioned.
2. Page 6: regarding MDSCs, the markers used to define them by FACS and IHC should be described in the text. Because there are various phenotypes described in different models and studies, the reader should be able to make a rapid interpretation.
3. The effects of LytC_22 recombinant protein on MDSCs needs an appropriate control, especially since the protein was made in bacteria that could contribute relevant contaminants such as LPS. The best control would be a mutant protein predicted not to bind Slamf4.
4. Since the overall model here involves intratumoral (or tumor-associated) bacteria, the results raise the question of whether *P. anaerobius* genomic sequences can be found within human tumors, and whether this is associated with an altered tumor microenvironment (increased MDSCs and decreased CD8 T cells), or better yet with decreased anti-PD-1 efficacy in the clinic.

Reviewer #3 (Remarks to the Author):

In this paper Liu et al. have analyzed the effect of a bacterium *Peptostreptococcus anaerobius* in the resistance to anti-PD1 treatment in different mouse models of colorectal cancer. They have first shown that this bacterium induces the recruitment of MDSC in both APCmin/+ mice and AOM/DSS mice. This leads to an increase in tumor burden in the colon of these mice. Then they have shown that this bacterium by recruiting MDSC interferes with efficacy of anti-PD1 treatment. They found another activity of *Peptostreptococcus anaerobius* that is to induce the antitumor properties of MDSC via the release of a protein called lytc22.

Although I found this manuscript interesting and the concept novel, there are several

5technical issues that do not support the conclusions of the authors and as is the manuscript has little advancement over previous observations from the same authors (Ref 5).

Indeed, the authors do not distinguish between aggressiveness of the tumor after intratumoral infection with *Peptostreptococcus anaerobius* and response to treatment.

Fig. 1 In their previous publication (Long et al. ref 5) they have already described that *Peptostreptococcus anaerobius* injection induces the recruitment of all of the granulocytic population and not only of MDSC. Here, they have focused on MDSC for its link to anti-PD1 treatment efficacy. Here they show in Fig. 1 that indeed i.t. injection of *Peptostreptococcus anaerobius* induces the recruitment of MDSC (already known), it favors CRC development (already known). No mention to other cell population like ILC or the different DC subtypes. Also they evaluate % and not absolute numbers of these populations. Thus it is not clear whether there is a reduction of other cell types or an expansion of these. The authors should show absolute numbers and evaluate other cell populations that are important in the gut such as ILC (1, 2 or 3), DC subtypes, gd T cells.

Fig. 2 they show that anti PD1 treatment is ineffective when *Peptostreptococcus anaerobius* is injected i.t. They claim that this is dependent on the MDSC. However, in Fig. 2c they show that also *E. coli* induces a recruitment of these cells albeit not at the same level. Thus, why is *E. coli* not interfering with anti-PD1 treatment? The authors have not shown that a similar effect is observed after for instance a chemotherapeutic treatment such as 5-FU. I would bet a similar effect is observed also in this case and is related primarily to the more aggressive nature of the tumors. In Fig. 2I a control is missing which is *Peptostreptococcus anaerobius* + anti-Ly6G. Indeed the authors cannot say if the observed effect would be independent on anti-PD1 treatment, for the aggressiveness of the tumors.

Fig. 3 here they show that PA treatment of tumor cells leads to the production of chemokines involved in MDSC recruitment. However, in Fig. 3a-b a control again is missing, which is medium + PA treated with antibiotics without the tumor cells. In Fig. 3c the authors switch to Caco-2 cells that they have not analyzed earlier. Then in Fig. 3d, they show the level of CXCL1 by different cell lines. Except for MC38, the amount of CXCL-1 is so tiny (few picograms in most of the cell lines) that one wonders how physiologically relevant this may be.

Fig. 4 is in the identification of a possible mechanism but the inhibitors used may affect so many different pathways that is difficult to evaluate which one is involved, and the findings are all association and causally related. In Fig. 4F again an important control is missing which is PA alone.

In Fig. 4J the difference between i.t. injection of PBS or PA is not significant anymore. Why is that? What type of statistics is used? The differences are so tiny that one wonders the significance.

Fig. 5 the control with *E. coli* is lost.

From Fig. 5 onwards, the authors identified first the supernatant and then the Lytc_22 as a possible mediator of the activity of PA, but have not demonstrated that this protein is also involved in the recruitment of MDSC. Does this mean that MDSC recruitment is not

important anymore? Indeed, in fig 7 they treated mice with Lytc_22 without PA and again see the effect of interference with anti-PD1 treatment.

Fig. 7I: They have not carried out an important experiment of anti-slamf4 and PA treatment.

In conclusions, I think that the authors lack important controls of the experiments and have not really demonstrated that what observed is related to an activity of PA against anti-PD1 treatment but mostly related to PA increased aggressiveness on tumor development.

Author Rebuttal to Initial comments

Response to the comments of referees in relation to the manuscript:

NMICROBIOL-23010097A: “*Peptostreptococcus anaerobius* provokes anti-PD-1 resistance in colorectal cancer by promoting MDSCs recruitment and activation”

Response to Reviewer #1:

*In the manuscript “*Peptostreptococcus anaerobius* provokes anti-PD-1 resistance in colorectal cancer by promoting MDSCs recruitment and activation”, Liu et al. identify the mechanism by which *P. anaerobius* interacts with both colorectal cancer cells and MDSCs to suppress anti-tumor immunity. The authors nicely identify this is facilitated by several host factors including integrin alpha2/beta1 and Slamf4 along with *P. anaerobius*-derived lytC_22 protein. While the experiments are very thorough, there are some questions regarding the relative contribution of these host proteins as well as the impact of *P. anaerobius* in naïve mice.*

*1. In figure 1, the authors establish that *P. anaerobius* suppresses the presence of tissue IFN γ -expressing CD4 and CD8 cells. Are the total numbers of CD4 and CD8 cells also reduced? This will help understand if these populations fail to be recruited or if they are recruited, but become suppressed at the tissue site.*

Response: Total CD4 and CD8 cells in the colon were not altered by *P. anaerobius* (S Figure 1A and 1C). These data are now presented in the text as follows:

Results (p.5, line 91-92):

Functional CD8⁺ T cells (IFN γ ⁺CD8⁺) and Th2 cells (IFN γ ⁺CD4⁺) were significantly decreased in lamina propria of *P. anaerobius*-treated mice compared to controls (both P<0.05) (Figure 1D), whereas no differences were observed in total CD8⁺ and CD4⁺ T cells (S Figure 1A).

Results (p.5, line 103-104):

No change was found in the infiltration of total CD8⁺ and CD4⁺ T cells (S Figure 1C).

2. Similarly, to above do these cells express any exhaustion markers, particularly PD1 during *P. anaerobius* gavage (during AOM/DSS and/or MC38 injections)? The authors use PD1 blockade, but did not report the levels of these markers on the proposed target immune population.

Response: We have now determined proportion of PD1⁺CD8⁺ cells in MC38 allografts and AOM/DSS mouse models (**S Figure 2A and 2B**), which showing that anti-PD1 suppressed PD1⁺CD8⁺ T cells. *P. anaerobius* alone had no effect on PD1, but it abolished reduction of PD1 by anti-PD1 in AOM/DSS model. These are now described in the revised text as follows:

Results (p.8, line 154-155):

Anti-PD-1 also suppressed PD1⁺CD8⁺ T cells, while *P. anaerobius* had no such effect (**S Figure 2A**).

Results (p.8, line 171-p.9, line 172):

Moreover, *P. anaerobius* reversed the effect of anti-PD-1 on decreasing PD1⁺CD8⁺ T cells (**S Figure 2B**).

3a. There is an overall question about the impact of *P. anaerobius* in the gut prior to tumorigenesis. Do naïve mice given *P. anaerobius* also display increased MDSC recruitment or does this require a transformed epithelium in the intestine to facilitate this response?

Response: To address this, we have now gavaged naïve mice with *P. anaerobius* for 1 month and analyzed MDSCs in colonic lamina propria by flow cytometry. *P. anaerobius* also promoted MDSC recruitment in these naïve mice (**S Figure 1L**). These findings are now described in the revised text as follows:

Results (p.7, line 136-139):

To assess if *P. anaerobius*-induced MDSC recruitment is tumor-specific, we gavaged naïve mice with *P. anaerobius*. Flow cytometry showed *P. anaerobius* also increased MDSCs in naïve mice (**S Figure 1L**), suggesting that *P. anaerobius* recruits MDSCs independently from tumorigenesis.

3b. Furthermore, are AOM/DSS-treated or MC-38-injected mice more prone to harboring *P. anaerobius* compared to naïve mice, potentially due to a weakened immune state? Measuring *P. anaerobius* CFUs post oral gavage in naïve vs. tumor bearing mice will help address the physiological relevance of this interaction.

Response: We have now measured stool *P. anaerobius* abundance by qPCR in AOM/ DSS-induced CRC mouse model and in naïve mice gavaged *P. anaerobius* for 2 weeks. We detected increased abundance of *P. anaerobius* in stool of AOM/DSS-treated mice as compared to naïve mice (**S Figure 1M**). Fluorescence in situ hybridization (FISH) also confirmed increased *P.*

anaerobius colonization in the colon of AOM/DSS-treated mice compared to naïve mice (**S Figure 1N**), indicating that CRC-bearing mice are more prone to *P. anaerobius*. This information has been added to the text as follows:

Results (p.7, line 139-140):

Nevertheless, the presence of CRC increased stool *P. anaerobius* abundance and its colonization in the colon (**S Figure 1M and 1N**).

4. In Fig. 3, the authors show that immune suppression by *P. anaerobius* is mediated by interactions with tumor cell integrin alpha2/beta1 and subsequent CXCL1 release in CRC cell lines. They also go on to show that *P. anaerobius* can directly activate MDSCs via production of LytC_22 and Slamf4. Does LytC_22/Slamf4 blockade impact CXCL1 release and/or conversely, does integrin alpha2/beta1-CXCL1 blockade impact LytC_22 production? This will help to understand if these two events function independently or in concert with each other to suppress immune surveillance.

Response: To ask if LytC_22/Slamf4 blockade impact CXCL1 release, we established MC38 allografts and injected them with PBS, lytC_22, or *P. anaerobius*, followed by anti-Slamf4 antibody treatment. Unlike *P. anaerobius*, lytC_22 did not induce CXCL1 secretion. In addition, anti-Slamf4 had no effect on *P. anaerobius*-induced CXCL1 *in vivo* (**S Figure 7B**). Anti-Slamf4 also failed to reverse *P. anaerobius*-induced CXCL1 secretion in Caco-2 and HCT116 *in vitro* (**S Figure 7C and 7D**).

We next investigated if blockade of integrin alpha2/beta1 impacts lytC_22 by addition of RGDS peptide to CRC cells co-cultured with *P. anaerobius*. Western blot indicated that RGDS peptide had no effect on production of lytC_22 by *P. anaerobius* (**S Figure 7E and 7F**). These results are now described as follows:

Results (p.17, line 357-364):

To evaluate if integrin $\alpha 2\beta 1$ -CXCL1 and lytC_22-Slamf4 axes function independently of each other, we determined effect of anti-Slamf4 on *P. anaerobius*-induced CXCL1 by CRC *in vitro* and in MC38 allografts. In both models, anti-Slamf4 had no effect on *P. anaerobius*-induced CXCL1 (**S Figure 7B-7D**). LytC_22 administration alone also had no effect on CXCL1 (**S Figure 7B**). Similarly, blockade of integrin $\alpha 2\beta 1$ -CXCL1 by RGDS peptide did not modulate production of lytC_22 by *P. anaerobius* co-cultured with CRC cells (**S Figure 7E and 7F**). These data imply that *P. anaerobius* promotes MDSCs in CRC through two independent mechanisms.

5. Does *P. anaerobius* lacking *LytC_22* fail to induce MDSC recruitment and enhanced tumorigenesis?

Response: Despite our best efforts to genetically manipulate *P. anaerobius* based on a variety of techniques, including electroporation or transformation, we failed to generate an isolate with *LytC_22* knockout. This may be due to difficulties in introducing foreign DNA to gram +ve bacteria, the strict anaerobic nature of *P. anaerobius*, or that *LytC_22* might be essential for viability.

Response to Reviewer #2:

*The manuscript by Liu and colleagues reports on the mechanism by which a specific bacteria species, *Peptostreptococcus anaerobius*, inhibits anti-tumor immunity and anti-PD-1 efficacy in a set of colorectal cancer mouse models. In the APC/min mouse model, oral gavage of *P. anaerobius* increased the number of polyps, which was associated with increased myeloid cells phenotypically resembling MDSCs and also decreased T cells. In the transplantable MC38 model, they switch to intratumoral administration of *P. anaerobius*, and found decreased efficacy of systemic anti-PD-1. In the AOM/DSS colitis model, they found that oral gavage of *P. anaerobius* increased tumor formation, decreased anti-PD-1 efficacy, and increased MDSC accumulation. Mechanistically, they provide evidence that *P. anaerobius* induces CXCL1 from tumor cells, which recruits MDSCs, and also engages MDSCs via secreted *lytC_22* through interaction with *Slamf4*. Taken together, these are potentially important results that also point towards potential therapeutic opportunities for reversal of this immune regulatory process in vivo. Nonetheless, there are several details that deserve further elaboration and clarification.*

1. The change from oral gavage to intratumoral bacterial administration was not spelled out clearly. I assume that oral gavage was not immune modulatory in the subcutaneous MC38 model. Since it was indeed immune modulatory in the spontaneous colonic tumor models, these results suggest that this phenomenon is mediated by tumor-associated bacteria rather than a systemic immune effect driven by altered composition of the gut microbiota. These points need to be developed and explained further throughout the manuscript: the title, abstract, and discussion would need to describe this as tumor-associated bacteria, and the background section of the paper should also describe prior literature on the effect of intratumoral bacteria. In the Results section, whenever the bacteria are delivered intratumorally this needs to be mentioned.

Response: We have now re-written our manuscript to focus to intratumoral bacteria, and added the description of intratumoral injection where possible, as follows:

Introduction (p.3, line 41-47):

Accumulating evidence has implicated the existence of intratumoral microbiome in multiple cancers¹. For instance, CRC-associated microbes play important roles in colorectal cancer (CRC) development through modulating tumor cell proliferation and tumor immune microenvironment^{1, 2}. Spatial profiling revealed that pathogen-infected tumor cells in CRC patients displayed gene signatures of enhanced cell migration and interferon response³. Studies with individual intratumoral pathogens also confirmed the ability to foster an immunosuppressive niche and drive colorectal tumorigenesis^{4, 5}.

Results (p.7, line 144-145):

Given the enrichment of MDSCs by *P. anaerobius*, we explored whether intratumoral *P. anaerobius* could affect anti-PD-1 efficacy.

Results (p.12, line 240-250):

We further explored whether blocking integrin $\alpha_2\beta_1$ receptor by RGDS peptide could reverse *P. anaerobius*-induced anti-PD-1 resistance. As shown in **Figure 4J and 4K**, without intratumoral *P. anaerobius* injection, RGDS peptide had no antitumor effect on anti-PD-1 in MC38 allografts. Whereas co-treatment of RGDS peptide rescued anti-PD-1 mAb efficacy in *P. anaerobius*-injected mice, as evidenced by reduced tumor growth and tumor volume as compared to *P. anaerobius* group ($P=0.005$). Co-treatment with RGDS peptide also blocked the effect of intratumoral *P. anaerobius* on CXCL1 secretion (**Figure 4L**), MDSC infiltration (**Figure 4M**), and IFN- γ^+ CD8 $^+$ T cell suppression (**Figure 4N**), thereby reversing anti-PD-1 resistance phenotypes. Consistently, RGDS peptide inhibited integrin $\alpha_2\beta_1$ /NF- κ B signaling to the baseline in *P. anaerobius* injected intratumorally to mice (**S Figure 4C**).

2. Page 6: regarding MDSCs, the markers used to define them by FACS and IHC should be described in the text. Because there are various phenotypes described in different models and studies, the reader should be able to make a rapid interpretation.

Response: We have defined CD11b $^+$ Gr-1 $^+$ cells as MDSCs by flow cytometry (Cd45 $^+$ Cd11b $^+$ Gr-1 $^{\text{high}}$) and immunofluorescence (CD11b $^+$ Gr-1 $^+$), and these definitions have been added to **p.5, line 88 and line 93**.

3. The effects of *LytC_22* recombinant protein on MDSCs needs an appropriate control, especially since the protein was made in bacteria that could contribute relevant contaminants such as LPS. The best control would be a mutant protein predicted not to bind Slamf4.

Response: To include a protein control for *LytC_22*, we have now performed molecular docking to identify binding between *LytC_22* and Slamf4, and identified seven key amino acid residues (E570, D653, D725, Y661, D712, R660, K669) on *LytC_22* (**Figure 7I**). We have generated a mutant *LytC_22* by replacing all seven residues with Ala. Mutant *LytC_22* showed weaker binding to Slamf4 (**Figure 7J**) and was impaired in inducing Arg1 and iNOS expression in MDSCs *in vitro* (**Figure 7K**). Consequently, MDSCs treated with *LytC_22* mutant failed to suppress T cell proliferation and function compared to blank control (**Figure 7L and S Figure 7A**). This information has been added as follows:

Results (p.16, line 328-333):

Based on *in-silico* prediction of its binding sites with Slamf4 (**Figure 7I**), we generated mutant *lytC_22* (E570A, D653A, D725A, Y661A, D712A, R660A, and K669A). Compared to wildtype

12lytC_22, mutant lytC_22 exhibited impaired binding to Slamf4 (**Figure 7J**), and accordingly was incapable of inducing Arg1 and iNOS expression (**Figure 7K**) and immunosuppressive function (**Figure 7L and S Figure 6D**) of MDSCs *in vitro*.

4. Since the overall model here involves intratumoral (or tumor-associated) bacteria, the results raise the question of whether *P. anaerobius* genomic sequences can be found within human tumors, and whether this is associated with an altered tumor microenvironment (increased MDSCs and decreased CD8 T cells), or better yet with decreased anti-PD-1 efficacy in the clinic.

Response: We have now analyzed TCGA CRC dataset and found that the abundance of *P. anaerobius* positively associated with MDSCs ($P = 0.01$) whilst being negatively correlated with effector T cells ($P = 0.02$) (**S Figure 1K**). This is now added as follows:

Results (p.7, line 132-134):

Intratumoral *P. anaerobius* abundance in TCGA CRC cohort was found to be positively correlated with MDSCs ($P=0.01$), whilst being negatively correlated with effector T cells ($P=0.02$) (**S Figure 1K**).

Response to Reviewer #3:

*In this paper Liu et al. have analyzed the effect of a bacterium *Peptostreptococcus anaerobius* in the resistance to anti-PD1 treatment in different mouse models of colorectal cancer. They have first shown that this bacterium induces the recruitment of MDSC in both APC^{min}/+ mice and AOM/DSS mice. This leads to an increase in tumor burden in the colon of these mice. Then they have shown that this bacterium by recruiting MDSC interferes with efficacy of anti-PD1 treatment. They found another activity of *Peptostreptococcus anaerobius* that is to induce the antitumor properties of MDSC via the release of a protein called lyc22. Although I found this manuscript interesting and the concept novel, there are several technical issues that do not support the conclusions of the authors and as is the manuscript has little advancement over previous observations from the same authors (Ref 5). Indeed, the authors do not distinguish between aggressiveness of the tumor after intratumoral infection with *Peptostreptococcus anaerobius* and response to treatment.*

*1. Fig. 1 In their previous publication (long et al. ref 5) they have already described that *Peptostreptococcus anaerobius* injection induces the recruitment of all of the granulocytic population and not only of MDSC. Here, they have focused on MDSC for its link to anti-PD1 treatment efficacy. Here they show in Fig. 1 that indeed i.t. injection of *Peptostreptococcus anaerobius* induces the recruitment of MDSC (already known), it favors CRC development (already known). No mention to other cell population like ILC or the different DC subtypes. Also, they evaluate % and not absolute numbers of these populations. Thus, it is not clear whether there is a reduction of other cell types or an expansion of these. The authors should show absolute numbers and evaluate other cell populations that are important in the gut such as ILC (1, 2 or 3), DC subtypes, gd T cells.*

Response: We have now calculated the absolute number of MDSCs and IFN γ ⁺CD8⁺ T cells (S

Figure 1F and 1G) in the AOM/DSS model, and the data were consistent with previous data. We have performed flow cytometry to detect ILC (types 1-3), plasmacytoid dendritic cells (pDCs), myeloid dendritic cells (mDCs), and $\gamma\delta$ T cells in colonic lamina propria from AOM/DSS-induced CRC mouse model, and no significant changes were found in these cell populations after *P. anaerobius* treatment (S **Figure 1H**). These results are described as follows:

Results (p.6, line 110-115):

We also validated these observations in AOM/DSS-induced CRC mouse model. *P. anaerobius* promoted tumor growth (S **Figure 1E**), increased the number of intratumoral MDSCs and decreased IFN γ ⁺CD8⁺ T cells compared to PBS and *E. coli* (S **Figure 1F and 1G**). However, no

significant changes were observed in the abundance of lymphoid cell (ILC) type 1, 2, 3, plasmacytoid dendritic cells (pDCs), myeloid dendritic cells (mDCs) and $\gamma\delta$ T cells in colonic lamina propria (S Figure 1H).

2a. Fig. 2 they show that anti PD1 treatment is ineffective when *Peptostreptococcus anaerobius* is injected i.t. They claim that this is dependent on the MDSC. However, in Fig. 2c they show that also *E. coli* induces a recruitment of these cells albeit not at the same level. Thus, why is *E. coli* not interfering with anti-PD1 treatment?

Response: Based on our data from Figure 2C, there was no significant change on the percentage of MDSCs between PBS and *E. coli* group.

2b. The authors have not shown that a similar effect is observed after for instance a chemotherapeutic treatment such as 5-FU. I would bet a similar effect is observed also in this case and is related primarily to the more aggressive nature of the tumors.

Response: To assess if *P. anaerobius* impacts chemotherapy, we generated AOM/DSS-induced CRC mice. Colonoscopy was performed to monitor tumorigenesis, after which mice were randomized and treated with PBS, *E. coli*, or *P. anaerobius*. After 10 days, 5-FU (50mg/kg) or Oxaliplatin (7.5 mg/kg) was given twice a week i.p. for 3 weeks. As shown below (panel A), *P. anaerobius* elicited chemotherapy resistance in CRC. *In vitro* studies demonstrated that *P. anaerobius* promoted CRC stemness as evidenced by elevated CD133 and LGR5 expression (panel B). Distinct mechanisms thus underlying chemoresistance and immunotherapy resistance induced by *P. anaerobius*. These data have not been included in the current revised manuscript, as they are part of a separate investigation.

2c. In Fig. 2I a control is missing which is *Peptostreptococcus anaerobius* + anti-Ly6G. Indeed, the authors cannot say if the observed effect would be independent on anti-PD1 treatment, for the aggressiveness of the tumors.

Response: We have now repeated the experiment in **Figure 2I** with *P. anaerobius*+anti-Ly6G and PBS+anti-Ly6G groups being included. In this MC38 allograft model, *P. anaerobius* had no significant effect on tumor growth or aggressiveness, irrespective of anti-Ly6G treatment (**Figure 2I** and **2J**).

3a. Fig. 3 here they show that PA treatment of tumor cells leads to the production of chemokines involved in MDSC recruitment. However, in Fig. 3a-b a control again is missing, which is

medium + PA treated with antibiotics without the tumor cells. In Fig. 3 c the authors switch to Caco-2 cells that they have not analyzed earlier.

Response: We have now repeated experiments in **Figure 3A and 3B** using Caco-2 and MC38 cells with the inclusion of control group (DMEM medium+*P. anaerobius* with antibiotics). Consistently, *P. anaerobius* significantly increased MDSC migration compared to this control.

Results (p.9, line 185-188):

CM derived from CRC cells (HCT 116, Caco2 and MC38) pre-incubated with *P. anaerobius* exhibited enhanced ability to induce MDSCs chemotaxis as compared to CM from CRC cells pre-incubated with PBS and blank control (DMEM medium+*P. anaerobius* with antibiotics) (**Figure 3B and S Figure 3B**).

3b. Then in Fig. 3d, they show the level of CXCL1 by different cell lines. Except for MC38, the amount of CXCL-1 is so tiny (few picograms in most of the cell lines) that one wonders how physiologically relevant this may be.

Response: CXCL1 levels appeared low as we normalized their secretion to absolute cell number. This corresponds to 300-500pg/mL CXCL1 in the conditioned medium of HCT116 and Caco-2 cells (6.0×10^5 cells per well), which compares favorably to the typical serum CXCL1 levels of $\sim 100\text{pg/mL}^1$. This information has been added in the Figure 3 legend.

4. Fig. 4 is in the identification of a possible mechanism but the inhibitors used may affect so many different pathways that is difficult to evaluate which one is involved, and the findings are all association and causally related. In Fig. 4F again an important control is missing which is PA alone.

Response: We have now presented all the data for PBS control, PA alone, RGDS peptide, and PA+RGDS together in **Figure 4F**, showing that RGDS peptide specifically inhibit the growth of PA-treated tumors.

5. In Fig. 4J the difference between i.t. injection of PBS or PA is not significant anymore. Why is that? What type of statistics is used? The differences are so tiny that one wonders the significance.

Response: In MC38 allografts, intratumoral injection of *P. anaerobius* was performed in established tumors for 2 weeks (average tumor size 100mm^3), in contrast to *Apc*^{Min/+} and AOM/DSS-induced CRC where mice were given prolonged treatments for >2 months during tumor initiation and progression. The short duration of exposure might account for the lack of effect of *P. anaerobius* in MC38 allografts. Two-way ANOVA was employed to analyze the statistical significance between groups in **Figure 4J** and description for statistical testing has been added to **Figure legends**.

6. Fig. 5 the control with *E. coli* is lost.

Response: We have repeated the experiments and added *E. coli* group as the control (**Figure 5A-5D**). *E. coli* had no effect on MDSC migration, Arg1/iNOS expression, and function. This is now described in the revised text as follows:

Results (p.12, line 259-p13, line 266):

PA-CM significantly increased mRNA ($P<0.05$) and protein ($P<0.001$) expression of Arg1 and iNOS on MDSCs as compared to broth control and *E. coli* CM (**Figure 5A**). To validate that MDSCs function was boosted by PA-CM, we performed T cell suppression assay (**Figure 5B**). Primary MDSCs were first treated with PA-CM, *E. coli* CM or broth control for 24h, and then co-cultured with primary CD8⁺ T or CD4⁺ T cells (**Figure 5B**). Compared to broth and *E. coli* CM treated MDSCs, MDSCs treated with PA-CM more effectively suppressed CD4⁺ and CD8⁺ T cell proliferation as evidenced by flow cytometry assay (**Figure 5C**).

7a. From Fig. 5 onwards, the authors identified first the supernatant and then the *Lytic_22* as a possible mediator of the activity of PA, but have not demonstrated that this protein is also involved in the recruitment of MDSC. Does this mean that MDSC recruitment is not important anymore? Indeed, in fig 7 they treated mice with *Lytic_22* without PA and again see the effect of interference with anti-PD1 treatment.

Response: We have now determined whether *lytc_22* could promote MDSC migration. Transwell assay showed that *lytc_22* (5-20nM) moderately induced MDSC migration *in vitro* ($P<0.001$) (**S Figure 6A**). Moreover, blockade of *lytc_22* receptor *slamf4* using anti-*slamf4* in *P. anaerobius*-treated MC38 allograft impaired MDSC recruitment (**Figure 7O**). These data suggest that *lytc_22* partly contribute to MDSC recruitment to promote anti-PD-1 resistance. This is now described in the revised text as follows:

Results (p.14, line 300-line 301):

LytC_22 also induced MDSC migration in transwell assay (**S Figure 6A**).

8. Fig. 7I: They have not carried out an important experiment of anti-*slamf4* and PA treatment.

Response: We have now repeated MC38 allografts model in **Figure 7M-7N** and added anti-Slamf4+*P. anaerobius* and anti-Slamf4+*lytC_22* groups. Anti-Slamf4 abolished *lytC_22*- and *P. anaerobius*-induced Arg1⁺ MDSCs and iNOS⁺ MDSCs, confirming that blocking Slamf4 receptor reversed *lytC_22*/*P. anaerobius*-induced MDSCs activation (**Figure 7O**). Combination of anti-*slamf4*+anti-PD1 overcome anti-PD1 resistance mediated by *lytC_22* or *P. anaerobius* (**Figure 7N**). This has been added to the revised text as follows:

Results (p.16, line 337-p.17, line 355):

To validate the function of *lytC_22* *in vivo*, we established MC38 allografts and treated established tumors with anti-PD-1 mAb with or without recombinant *lytC_22* or *P. anaerobius* by intratumoral injection. Anti-PD-1 efficacy was reduced by *P. anaerobius* or *lytC_22* treatment with significantly increased tumor volume (Figure 7M) and tumor weight (Figure 7N) as compared to those treated with anti-PD-1 alone. *P. anaerobius* or *lytC_22*-induced anti-PD-1 resistance was largely reversed by anti-Slamf4 neutralizing antibody treatment ($P < 0.01$) (Figure 7M and 7N). Concordantly, *P. anaerobius* promoted total ($P < 0.001$), Arg1⁺ ($P = 0.003$), and iNOS⁺ ($P = 0.001$) MDSCs in MC38 allografts (Figure 7O); which abrogated the induction of IFN γ ⁺ and GrzB⁺ CD8⁺T cells by anti-PD-1 treatment (Figure 7P and S Figure 7A). *lytC_22* alone increased Arg1⁺ ($P = 0.003$) and iNOS⁺ ($P < 0.001$) MDSCs (Figure 7O), together with a reduction in IFN γ ⁺ and GrzB⁺ CD8⁺ T cells compared with PBS group after anti-PD-1 treatment (Figure 7P and S Figure 7A). Importantly, anti-Slamf4 neutralizing antibody reversed *lytC_22*- or *P. anaerobius*-mediated MDSCs activation (Figure 7O) and T cell suppression (Figure 7P and S Figure 7A) by flow cytometry, thereby restoring anti-PD-1 efficacy. Collectively, our results suggested that *P. anaerobius*-derived *lytC_22* protein promoted immunosuppressive function of MDSCs and anti-PD-1 resistance via Slamf4 receptor of MDSC. Slamf4 blockade improves anti-PD1 efficacy in the context of *lytC_22* treatment.

1. Lukaszewicz-Zajac M, Zajkowska M, Paczek S, et al. The Significance of CXCL1 and CXCR1 as Potential Biomarkers of Colorectal Cancer. *Biomedicines* 2023;11.

Decision Letter, first revision:

Message: 20th November 2023

Dear Jun,

Thank you for your patience while your manuscript "Peptostreptococcus anaerobius provokes anti-PD-1 resistance in colorectal cancer by promoting MDSCs recruitment and activation" was under peer-review at Nature Microbiology. It has now been seen by 2 referees, whose expertise and comments you will find at the of this email. As mentioned by Jess, we have asked referee #3 to comment on your rebuttal to the previous concerns from

19referee #2 as they were unable to look at the revision. Referee #3 confirmed that you satisfactorily addressed these concerns.

You will see from their comments below that while they find your work of interest, some important points are raised. We are very interested in the possibility of publishing your study in Nature Microbiology, but would like to consider your response to these concerns in the form of a revised manuscript before we make a final decision on publication.

In particular, you will see that referee #3 has some remaining concerns that will need to be addressed. The rest referees' reports are clear and the remaining issues should be straightforward to address.

If you have not done so already please begin to revise your manuscript so that it conforms to our Article format instructions at <http://www.nature.com/nmicrobiol/info/final-submission/>

The usual length limit for a Nature Microbiology Article is six display items (figures or tables) and 3,000 words. We have some flexibility, and can allow a revised manuscript at 3,500 words, but please consider this a firm upper limit. There is a trade-off of ~250 words per display item, so if you need more space, you could move a Figure or Table to Supplementary Information.

Some reduction could be achieved by focusing any introductory material and moving it to the start of your opening 'bold' paragraph, whose function is to outline the background to your work, describe in a sentence your new observations, and explain your main conclusions. The discussion should also be limited. Methods should be described in a separate section following the discussion, we do not place a word limit on Methods.

Nature Microbiology titles should give a sense of the main new findings of a manuscript, and should not contain punctuation. Please keep in mind that we strongly discourage active verbs in titles, and that they should ideally fit within 90 characters each (including spaces).

Please include a data availability statement as a separate section after Methods but before references, under the heading "Data Availability". This section should inform readers about the availability of the data used to support the conclusions of your study. This information

20includes accession codes to public repositories (data banks for protein, DNA or RNA sequences, microarray, proteomics data etc...), references to source data published alongside the paper, unique identifiers such as URLs to data repository entries, or data set DOIs, and any other statement about data availability. At a minimum, you should include the following statement: "The data that support the findings of this study are available from the corresponding author upon request", mentioning any restrictions on availability. If DOIs are provided, we also strongly encourage including these in the Reference list (authors, title, publisher (repository name), identifier, year). For more guidance on how to write this section please see:
<http://www.nature.com/authors/policies/data/data-availability-statements-data-citations.pdf>

To improve the accessibility of your paper to readers from other research areas, please pay particular attention to the wording of the paper's opening bold paragraph, which serves both as an introduction and as a brief, non-technical summary in about 150 words. If, however, you require one or two extra sentences to explain your work clearly, please include them even if the paragraph is over-length as a result. The opening paragraph should not contain references. Because scientists from other sub-disciplines will be interested in your results and their implications, it is important to explain essential but specialised terms concisely. We suggest you show your summary paragraph to colleagues in other fields to uncover any problematic concepts.

If your paper is accepted for publication, we will edit your display items electronically so they conform to our house style and will reproduce clearly in print. If necessary, we will re-size figures to fit single or double column width. If your figures contain several parts, the parts should form a neat rectangle when assembled. Choosing the right electronic format at this stage will speed up the processing of your paper and give the best possible results in print. We would like the figures to be supplied as vector files - EPS, PDF, AI or postscript (PS) file formats (not raster or bitmap files), preferably generated with vector-graphics software (Adobe Illustrator for example). Please try to ensure that all figures are non-flattened and fully editable. All images should be at least 300 dpi resolution (when figures are scaled to approximately the size that they are to be printed at) and in RGB colour format. Please do not submit Jpeg or flattened TIFF files. Please see also 'Guidelines for Electronic Submission of Figures' at the end of this letter for further detail.

Figure legends must provide a brief description of the figure and the symbols used, within 350 words, including definitions of any error bars employed in the figures.

When submitting the revised version of your manuscript, please pay close attention to our [href="https://www.nature.com/nature-research/editorial-policies/image-integrity"](https://www.nature.com/nature-research/editorial-policies/image-integrity) Digital Image Integrity Guidelines. and to the following points below:

Please include a statement before the acknowledgements naming the author to whom correspondence and requests for materials should be addressed.

Finally, we require authors to include a statement of their individual contributions to the paper -- such as experimental work, project planning, data analysis, etc. -- immediately after the acknowledgements. The statement should be short, and refer to authors by their initials. For details please see the Authorship section of our joint Editorial policies at http://www.nature.com/authors/editorial_policies/authorship.html

- * include a point-by-point response to any editorial suggestions and to our referees. Please include your response to the editorial suggestions in your cover letter, and please upload your response to the referees as a separate document.

- * ensure it complies with our format requirements for Letters as set out in our guide to authors at www.nature.com/nmicrobiol/info/gta/

- * state in a cover note the length of the text, methods and legends; the number of references; number and estimated final size of figures and tables

- * resubmit electronically if possible using the link below to access your home page:

*This url links to your confidential homepage and associated information about manuscripts you may have submitted or be reviewing for us. If you wish to forward this e-mail to co-authors, please delete this link to your homepage first.

Please ensure that all correspondence is marked with your Nature Microbiology reference number in the subject line.

Nature Microbiology is committed to improving transparency in authorship. As part of our efforts in this direction, we are now requesting that all authors identified as 'corresponding author' on published papers create and link their Open Researcher and Contributor Identifier (ORCID) with their account on the Manuscript Tracking System (MTS), prior to acceptance. This applies to primary research papers only. ORCID helps the scientific community achieve unambiguous attribution of all scholarly contributions. You can create and link your ORCID from the home page of the MTS by clicking on 'Modify my Springer Nature account'. For more information please visit please visit www.springernature.com/orcid.

We hope to receive your revised paper within three weeks. If you cannot send it within this

time, please let us know.

Yours sincerely,

Reviewers Comments:

Reviewer #1 (Remarks to the Author):

In the revised version of the manuscript "Peptostreptococcus anaerobius provokes anti-PD-1 resistance in colorectal cancer by promoting MDSCs recruitment and activation", the authors have performed nearly all of the requested experiments to address limitations in the original submission. The generation of *P. anaerobius* mutants that fail to bind to host integrins would provide further mechanistic insight from the bacterial standpoint. However, the author's new findings fill other critical gaps, thus strengthening the overall conclusions.

Reviewer #3 (Remarks to the Author):

I am partly satisfied with the revision.
I am still concerned about two points:

Fig 3 are cells still alive after PA treatment? Why do the authors need to normalize for the number of cells?

The authors should in each individual figure explain better where they used unpaired Student's t test and where anova. It looks to me that the authors used Student's t test everywhere (for example fig 4 L and M)

Author Rebuttal, first revision:

Response to the comments of referees in relation to the manuscript:

NMICROBIOL-23010097R2: "Peptostreptococcus anaerobius provokes anti-PD-1 resistance in colorectal cancer by promoting MDSCs recruitment and activation"

Reviewer #1:

In the revised version of the manuscript "Peptostreptococcus anaerobius provokes anti-PD-1 resistance in colorectal cancer by promoting MDSCs recruitment and activation", the authors

23have performed nearly all of the requested experiments to address limitations in the original submission. The generation of *P. anaerobius* mutants that fail to bind to host integrins would provide further mechanistic insight from the bacterial standpoint. However, the author's new findings fill other critical gaps, thus strengthening the overall conclusions.

Response: Many thanks for your positive comments.

Response to Reviewer #3:

1. Fig 3 are cells still alive after PA treatment? Why do the authors need to normalize for the number of cells?

Response: In Fig. 3, cells are still alive after PA treatment. We followed the protocol of our published paper (PMID: 28126350), *P. anaerobius* at this dosage (MOI:100) promoted CRC cell proliferation. We normalized for the number of cells to control for potential differences in cell growth. As shown below, even data prior to normalization consistently demonstrated that *P. anaerobius* increased CXCL1 in conditioned medium of CRC cells.

2. The authors should in each individual figure explain better where they used unpaired Student's t test and where anova. It looks to me that the authors used Student's t test everywhere (for example fig 4 L and M).

Response: We have added the details of statistical analysis in all figure legends. A Student's t-test was used to compare the differences between two sample groups (Fig 1G-J, Fig 3D, 3E, Fig 5G, 5I, Fig 6H, Fig 7A, 7B). One-way ANOVA was used to compare the difference between 3 or multiple groups (Fig 1B-1E, Fig 2B-2H, 2J, Fig 3B, 3F, 3J, Fig 4A-4I, 4K-4N, Fig 5A, 5D, 5F, Fig 6A-6C, 6E-6F, Fig 7C-7P).

Decision Letter, second revision:

Message Our ref: NMICROBIOL-23010097B

:

2420th December 2023

Dear Dr. Yu,

Thank you for submitting your revised manuscript "Peptostreptococcus anaerobius provokes anti-PD-1 resistance in colorectal cancer by promoting MDSCs recruitment and activation" (NMICROBIOL-23010097B). It has now been seen by the original referees and their comments are below. The reviewers find that the paper has improved in revision, and therefore we'll be happy in principle to publish it in Nature Microbiology, pending minor revisions to comply with our editorial and formatting guidelines.

We are now performing detailed checks on your paper and will send you a checklist detailing our editorial and formatting requirements after the break for the holidays, in the first weeks of January 2024. Please do not upload the final materials and make any revisions until you receive this additional information from us.

Thank you again for your interest in Nature Microbiology Please do not hesitate to contact me if you have any questions.

Reviewer #3 (Remarks to the Author):

I am happy with the revision

Final Decision Letter:

Message 4th April 2024

:

Dear Professor Yu,

I am pleased to accept your Article "Peptostreptococcus anaerobius mediates anti-PD1 therapy resistance and exacerbates colorectal cancer via myeloid-derived suppressor cells in mice" for publication in Nature Microbiology. Thank you for having chosen to submit your work to us and many congratulations.

25Once your paper is typeset, you will receive an email with a link to choose the appropriate publishing options for your paper and our Author Services team will be in touch regarding any additional information that may be required. Once your paper has been scheduled for online publication, the Nature press office will be in touch to confirm the details.

Please note that *Nature Microbiology* is a Transformative Journal (TJ). Authors may publish their research with us through the traditional subscription access route or make their paper immediately open access through payment of an article-processing charge (APC). Authors will not be required to make a final decision about access to their article until it has been accepted. Find out more about Transformative Journals
